# RETHINKING GNNS AND MISSING FEATURES: CHALLENGES, EVALUATION AND A ROBUST SOLUTION

## ABSTRACT

Handling missing node features is a key challenge for deploying Graph Neural Networks (GNNs) in real-world domains such as healthcare and sensor networks. Existing studies mostly address relatively benign scenarios, namely benchmark datasets with (a) high-dimensional but sparse node features and (b) incomplete data generated under *Missing Completely At Random (MCAR)* mechanisms. For (a), we theoretically prove that high sparsity substantially limits the information loss caused by missingness, making all models appear robust and preventing a meaningful comparison of their performance. To overcome this limitation, we introduce one synthetic and three real-world datasets with dense, semantically meaningful features. For (b), we move beyond MCAR and design evaluation protocols with more realistic missingness mechanisms. Moreover, we provide a theoretical background to state explicit assumptions on the missingness process and analyze their implications for different methods. Building on this analysis, we propose `GNNmim`, a simple yet effective baseline for node classification with incomplete feature data. Experiments show that `GNNmim` is competitive with respect to specialized architectures across diverse datasets and missingness regimes.

## 1 INTRODUCTION

Learning with missing features is a pervasive and often unavoidable challenge in many real-world machine learning applications, such as healthcare (Braem et al., 2024; Mirkes et al., 2016), IoT sensor networks (Faizin et al., 2019; Okafor & Delaney, 2021; Agbo et al., 2022), and recommender systems (Marlin & Zemel, 2009; He et al., 2017; Marlin et al., 2011). This issue naturally extends to Graph Neural Networks (GNNs), which are increasingly applied in domains where missing features are common. In this work, we focus specifically on the problem of *missing node feature data*, a setting that has received growing attention in the GNN literature (Um et al., 2023; Yun et al., 2024; Rossi et al., 2022; Guo et al., 2023; Taguchi et al., 2021; Errica & Niepert, 2024; Um et al., 2025)

A wide range of methods have been proposed, from simple mean imputation (You et al., 2020) to architectures that jointly impute and predict during training (Guo et al., 2023). These approaches are typically evaluated by synthetically removing features from widely used node classification benchmarks such as CORA, CITESEER, and PUBMED (Yang et al., 2016). However, despite the growing number of models, little attention has been paid to the validity of these evaluation protocols. We argue that two critical issues remained largely unaddressed: (i) the datasets used for evaluation, and (ii) the missingness mechanisms applied to generate incomplete features.

Regarding (i), existing evaluations rely on datasets with *extremely sparse* node features, typically bag-of-words representations where the vast majority of entries are zero. This raises a crucial question: *can robustness to missing features be meaningfully assessed when most features are already absent?* Our theoretical analysis shows that in highly sparse settings, the mutual information between features and labels is barely affected by additional missingness, except at extremely high missing rates. Empirically, we find that all the existing GNN-based methods maintain high performance across a wide range of missingness levels on these benchmarks, with performance degrading only when more than 90% of entries are removed. These results cast serious doubt on the ability of current benchmarks to meaningfully assess the robustness of the models.

To move beyond this limitation, we identify a set of datasets, one synthetic and three real-world, with dense, raw features that are naturally low-dimensional and semantically meaningful (e.g., physical measurements). These datasets offer a more realistic setting for studying GNNs under feature missingness. This focus on dataset quality aligns with recent calls for more careful benchmark design in graph machine learning (Bechler-Speicher et al., 2025; Coupette et al., 2025).

Regarding (ii), the design of the missingness mechanisms used during evaluation is overly simplistic. Most prior works consider only *Missing Completely At Random (MCAR)* mechanisms (Rubin, 1976; Little & Rubin, 2019), where feature deletion is independent of the data. In practice, however, missingness is often related to the feature values or prediction target (Carreras et al., 2021; Hazewinkel et al., 2022; Kopra et al., 2015). For example, a patient might be less likely to report their weight if it is above a certain threshold. This corresponds to a Missing Not At Random (MNAR) mechanism (Rubin, 1976), in which the probability of missingness depends on the unobserved feature value itself. A further limitation of existing evaluation protocols is the implicit assumption that the missingness mechanism remains identical across training and test data. In practice, however, this is often not the case: for example, training data may be historical and collected with obsolete sensors prone to failures, while test data come from newer sensors with little or no missingness. To overcome this limitation of the current evaluation procedure, we design more realistic evaluation protocols. These include new, more representative instances of MCAR and MNAR mechanisms, as well as train–test distribution shifts. Such conditions more accurately capture real-world deployment challenges, where both the causes and the distributions of missing data may vary across stages.

Finally, we introduce a simple yet effective GNN model, **GNNmim**, based on the Missing Indicator Method (MIM) (Van Ness et al., 2023). GNNmim augments the node feature matrix with a binary mask indicating which features are missing. The resulting representation is processed by a standard GNN without requiring any learned imputation. GNNmim does not rely on any assumption on the distribution of the missingness and, despite its simplicity, it is competitive with respect to several state-of-the-art methods showing robustness under a variety of missingness settings.

**Contributions.** To summarize, our main contributions are:

1. We provide a theoretical analysis showing that the impact of missing features depends strongly on feature sparsity, and derive an information-theoretic bound on the resulting loss.

2. We introduce one synthetic and three real-world datasets with dense, informative features, and show experimentally that models appearing robust on sparse benchmarks fail on these datasets.

3. We propose realistic evaluation protocols, including new, more representative instances of MCAR and MNAR mechanisms and train–test distribution shifts, and demonstrate that existing methods are not robust to all the possible settings.

4. We introduce GNNmim, a simple yet effective method, and show that it is competitive with respect to existing approaches across datasets, missingness types, and distribution shifts.

The core aim of this paper is to redefine how research on GNNs with missing features should move forward. We show that apparent progress in this area has been largely constrained by the evaluation itself: existing benchmarks rely on sparse, weakly informative features and overly benign missingness mechanisms, making current results difficult to interpret and obscuring the true robustness of existing methods. By introducing dense, semantically meaningful datasets, realistic missingness protocols, and a clear theoretical framing, we establish a foundation that enables more meaningful and reliable research directions. Within this improved evaluation setup, GNNmim is intentionally simple: once evaluation artifacts are removed, a lightweight, assumption-free model can outperform more complex approaches. Thus, GNNmim serves as an effective baseline that naturally arises from the identification and analysis of the limitations of the current evaluation setup. The broader contribution of this work lies in establishing a principled and realistic evaluation framework, with GNNmim serving as a clear baseline within it.

## 2 LEARNING FROM INCOMPLETE GRAPH DATA

We consider an attributed graph $G = (V, E, \mathbf{X}, \mathbf{Y})$, where $V = \{1, \ldots, n\}$ is the set of nodes, $E \subseteq V \times V$ is the set of edges represented by the adjacency matrix $\mathbf{A} \in \{0, 1\}^{n \times n}$, $\mathbf{X} \in \mathbb{R}^{n \times d}$ is

the node feature matrix with entry $X_{ij}$ denoting feature $j$ of node $i$, and $\mathbf{Y} \in \mathcal{Y}^n$ is the vector of node labels.

When data is incomplete, some entries of $\mathbf{X}$ are unobserved. Let $\mathbf{M} \in \{0,1\}^{n \times d}$ be the missingness indicator matrix that has $M_{ij} = 1$ if $x_{ij}$ is missing and 0 otherwise. In our setting, the missingness indicator matrix $\mathbf{M}$ is directly and deterministically constructed from the observed dataset. Missing values are explicitly marked in the raw data, so the mask $\mathbf{M}$ is uniquely defined and contains no uncertainty. Let $\mathbf{X}^{obs}$ be the elements of $\mathbf{X}$ for which $M_{ij} = 0$, and $\mathbf{X}^{miss}$ the elements for which $M_{ij} = 1$. The observed data from which we learn then can be written as $\mathbf{X}^{obs}, \mathbf{Y}, \mathbf{M}$. We note that we here make the assumption that $\mathbf{Y}$ is fully observed in the (training) data, and that there is no uncertainty about the graph structure $E$. The distribution of the data then can be parameterized as

$$P_{\boldsymbol{\theta},\boldsymbol{\gamma},\boldsymbol{\lambda}}(\mathbf{X}^{obs}, \mathbf{Y}, \mathbf{M}) = \int_{\mathbf{X}^{miss}} P_{\boldsymbol{\theta}}(\mathbf{X}) P_{\boldsymbol{\gamma}}(\mathbf{Y}|\mathbf{X}) P_{\boldsymbol{\lambda}}(\mathbf{M}|\mathbf{X}, \mathbf{Y}), \tag{1}$$

where $\mathbf{X} = \mathbf{X}^{obs} \cup \mathbf{X}^{miss}$, $P_{\boldsymbol{\theta}}$ is the node feature distribution, $P_{\boldsymbol{\gamma}}$ is the conditional label distribution, and $P_{\boldsymbol{\lambda}}$ represents the *missingness mechanism*. Though not explicitly reflected in the notation, all these distributions will usually depend on the underlying graph structure, which will typically induce dependencies among the rows of $\mathbf{X}$, and among the elements of $\mathbf{Y}$.

A GNN for node classification with complete feature data is a model $P_{\boldsymbol{\gamma}}(\mathbf{Y}|\mathbf{X})$ with $\boldsymbol{\gamma}$ the weights of the GNN. For classification with incomplete data we need to learn the conditional model

$$P_{\boldsymbol{\theta},\boldsymbol{\gamma},\boldsymbol{\lambda}}(\mathbf{Y}|\mathbf{X}^{obs}, \mathbf{M}) = \int_{\mathbf{X}^{miss}} P_{\boldsymbol{\theta},\boldsymbol{\gamma},\boldsymbol{\lambda}}(Y|X, M) P_{\boldsymbol{\theta},\boldsymbol{\gamma},\boldsymbol{\lambda}}(\mathbf{X}^{miss}|\mathbf{X}^{obs}, M). \tag{2}$$

The classical *missing (completely) at random (M(C)AR)* assumptions (Rubin, 1976) simplify this problem. The original M(C)AR assumptions have been formulated in the context of estimating the parameter of a generative distribution. It has been observed that more specialized variations of the original definitions can be more pertinent in the context of classification (Ding & Simonoff, 2010; Ghorbani & Zou, 2018). In the following we give formulations of M(C)AR for classification that provide the foundations for our theoretical analysis.

**Definition 1.** The joint distribution $P_{\boldsymbol{\theta},\boldsymbol{\gamma},\boldsymbol{\lambda}}$ is *feature*-MAR, if

$$P_{\boldsymbol{\gamma},\boldsymbol{\lambda}}(\mathbf{M}|\mathbf{X}^{miss}, \mathbf{X}^{obs}) = P_{\boldsymbol{\theta},\boldsymbol{\gamma},\boldsymbol{\lambda}}(\mathbf{M}|\mathbf{X}^{obs}). \tag{3}$$

It is *label*-MAR if

$$P_{\boldsymbol{\lambda}}(\mathbf{M}|X, Y) = P_{\boldsymbol{\gamma},\boldsymbol{\lambda}}(\mathbf{M}|X). \tag{4}$$

The distribution is MCAR, if

$$P_{\boldsymbol{\lambda}}(\mathbf{M}|X, Y) = P_{\boldsymbol{\theta},\boldsymbol{\gamma},\boldsymbol{\lambda}}(\mathbf{M}). \tag{5}$$

In (3)-(5) all probability functions are indexed with the parameters they actually depend on. Note, for example, that the conditional of $\mathbf{M}$ given $\mathbf{X}$ requires marginalization over $\mathbf{Y}$, and thereby also depends on the parameter $\boldsymbol{\gamma}$. MCAR implies both feature- and label-MAR.

The simplest realization of an MCAR mechanism is *uniform missingness (U-MCAR)* in which entries of $\mathbf{X}$ are independently missing with a fixed missingness probability $\mu$. This can be generalized by defining a missingness probability matrix $\boldsymbol{\mu} \in [0,1]^{n \times d}$ specifying potentially different missingness probabilities for different entries of $\mathbf{X}$.

MAR assumptions allow us to eliminate the missingness model $P_{\boldsymbol{\lambda}}$ from (2). The following proposition states this classical *ignorability* result in a version most suitable in our context.

**Theorem 1.** If $P_{\boldsymbol{\theta},\boldsymbol{\gamma},\boldsymbol{\lambda}}$ is feature-MAR and label-MAR, then (2) simplifies to

$$\int_{\mathbf{X}^{miss}} P_{\boldsymbol{\gamma}}(Y|X) P_{\boldsymbol{\theta}}(\mathbf{X}^{miss}|\mathbf{X}^{obs}). \tag{6}$$

**Intuition.** Under feature-MAR and label-MAR, the missingness pattern carries no predictive information. The learning problem reduces to the usual classification task with imputed features, meaning that methods explicitly modeling the missingness mask do not gain theoretical advantage in this regime.

The proof is straightforward by rewriting the two factors on the right of (2) using Bayes's rule, and plugging in (3) and (4). Formulation (6) still poses two major challenges: it requires a feature distribution model $P_{\boldsymbol{\theta}}$ when in reality we only are interested in the conditional model $P_{\boldsymbol{\gamma}}$, and the integration over $\mathbf{X}^{miss}$ is usually intractable (Ipsen et al., 2022). The simplest approach to address these problems is to approximate the integral (6) by evaluating $P_{\boldsymbol{\gamma}}(\boldsymbol{Y}|\boldsymbol{X})$ at a single imputed value $\mathbf{X} = impute(\mathbf{X}^{miss})$ (Rubin, 1988). This does not require an explicit model for $P_{\boldsymbol{\theta}}$, but relies on the implicit assumption that the imputed value $impute(\mathbf{X}^{miss})$ has high probability under $P_{\boldsymbol{\theta}}$. A simple example is *mean-imputation*, in which missing values of a given feature are filled with the mean of that feature; we will refer to this approach combined with a standard GNN as GNNmi (You et al., 2020). In addition, we also consider *zero-imputation*, where missing entries are replaced with zeros (GNNzero), and *median-imputation*, where they are filled with the feature median (GNNmedian). Similarly, PCFI (Um et al., 2023) does not require an explicit model for $P_{\boldsymbol{\theta}}$; it introduces a confidence-guided imputation scheme where pseudo-confidence is derived from the shortest-path distance to observed features, and combines channel-wise diffusion with inter-channel propagation to recover a single estimate of $\mathbf{X}$. GOODIE (Yun et al., 2024) approximates the integral in (6) using a combination of label propagation and FP (Rossi et al., 2022), which propagates features by minimizing a Dirichlet energy function, whereas FairAC (Guo et al., 2023) does so by aggregating, via an attention mechanism, the representations from neighbors of nodes with missing features.

Other methods explicitly model $P_{\boldsymbol{\theta}}$. The GCNmf approach of Taguchi et al. (2021) introduces a model of $P_{\boldsymbol{\theta}}$ in the form of a mixture of Gaussians, and approximates (6) by $P_{\boldsymbol{\gamma}}(\boldsymbol{Y}, |, \mathbb{E}_{\theta}[\mathbf{L}1 \mid \mathbf{X}^{obs}])$, where $\mathbb{E}_{\theta}[\mathbf{L}_1 \mid \mathbf{X}^{obs}]$ is the expected activation at the first layer of the GNN defining $P_{\boldsymbol{\gamma}}$. Finally, GSPN (Errica & Niepert, 2024) explicitly models $P_{\boldsymbol{\theta}}$ with graph-induced sum–product networks, so missing features are handled by exact marginalization.

An alternative to all these approaches that work entirely with models $P_{\boldsymbol{\theta}}, P_{\boldsymbol{\gamma}}$ for the (complete) data distribution is to include the missingness mechanism explicitly in a model $P_{\boldsymbol{\gamma}^+}(\boldsymbol{Y}|\mathbf{X}^{obs}, \boldsymbol{M})$, that directly captures the left side of (2). We here write $\boldsymbol{\gamma}^+$ for the parameters of the model to emphasize that it can be structurally similar to a model $P_{\boldsymbol{\gamma}}(\boldsymbol{Y}|\boldsymbol{X})$, but different in that it has the missingness matrix $\boldsymbol{M}$ as an explicit extra input.

This modeling strategy, often referred to as the Missing Indicator Method (MIM), has been studied in the context of supervised learning with missing features (Van Ness et al., 2023), but, to the best of our knowledge, it has not been explored in the context of graph machine learning. In this work, we propose a GNN-based instantiation of the MIM framework, which we call GNNmim. In GNNmim, we implement $P_{\boldsymbol{\gamma}^+}$ as a GNN, we construct the matrix *zero-pad*($\mathbf{X}^{obs}$) in which missing values are filled in by zeros, and use the concatenation *zero-pad*($\mathbf{X}^{obs}$)$[i,:]||\boldsymbol{M}[i,:]$ as the feature vector for node $i$ in an otherwise standard GNN architecture[1]. GNNmim does not rely on any MAR assumptions, and thereby can be expected to perform more robustly than other approaches under different missingness mechanisms. As our experiments in Section 5 show, this simple yet principled strategy yields robust performance across a wide variety of missingness scenarios. In Appendix I, we provide additional analyses where the missing-feature mask is applied not only to zero imputation but also to the existing models presented in this section.

# 3 ARE WE EVALUATING GNNS FOR MISSING FEATURES ON THE RIGHT DATA?

A rigorous evaluation of GNNs under feature missingness requires not only well-designed models, but also datasets that are suitable for the problem at hand. Recent work in the graph learning community has emphasized the importance of dataset suitability in benchmarking (Bechler-Speicher et al., 2025; Coupette et al., 2025). In the context of learning with missing node features, dataset suitability is even more critical. Models designed to handle missingness should be tested on datasets where the

---

[1]We deliberately here say "zero-padding" rather than "zero-imputation". The latter would imply that we view the zeros as somehow reasonable stand-ins for the true unobserved values. We view the zeros as arbitrary placeholders. Ideally, the trained model will learn to ignore these values when the corresponding missingness indicator is 1.

presence of missing features meaningfully affects model performance and where reasoning under missingness is necessary and non-trivial.

The current standard practice in the literature is to evaluate state-of-the-art methods on a set of widely-used benchmarks for node-level tasks, namely, CORA, CITESEER, PUBMED, AMAZON-COMPUTERS, and AMAZONPHOTO. In these datasets, node features are constructed as follows: CORA, CITESEER and PUBMED use binary bag-of-words features, while AMAZONCOMPUTERS and AMAZONPHOTO use TF-IDF vectors (Aizawa, 2003). These feature matrices are typically very sparse, which we quantify using the notion of *feature sparsity*, formally defined as below:

**Definition 2** (Feature Sparsity). Given a node feature matrix $\mathbf{X} \in \mathbb{R}^{n \times d}$, the *feature sparsity* is defined as the proportion of zero entries: $s(\mathbf{X}) = \frac{1}{nd} \sum_{i=1}^{n} \sum_{j=1}^{d} \mathbf{1}[X_{ij} = 0]$, where $\mathbf{1}[\cdot]$ denotes the indicator function.

The sparsity values of the benchmark datasets are reported in Table 1 (first three rows). All datasets exhibit substantial sparsity, with more than 50% of features being zero across all the datasets, with Citeseer reaching an extreme sparsity level of approximately 99%. This raises a crucial question: does it make sense to evaluate models designed to handle missing features on datasets where the feature representations are already extremely sparse? In such sparse settings, a high probability of missingness is needed to induce a meaningful information loss. Otherwise, the observed model performance under missingness may reflect artifacts of the dataset rather than the robustness of the method. We formalize this observation in the following theorem.

Table 1: Feature sparsity across benchmarks and custom datasets.

| Dataset | #Features | Sparsity ↓ | Type of features |
|---|---|---|---|
| CORA | 1433 | 0.9873 | BoW (binary) |
| CITESEER | 3703 | 0.9915 | BoW (binary) |
| PUBMED | 500 | 0.8998 | BoW (binary) |
| SYNTHETIC | 5 | 0.0000 | Gaussian |
| AIR | 7 | 0.1615 | Raw |
| ELECTRIC | 5 | 0.2000 | Raw |
| TADPOLE | 15 | 0.0000 | Raw |

**Theorem 2.** Let $\mathbf{X} \in \mathbb{R}^{n \times d}$ and $\mathbf{Y} \in \mathcal{Y}^n$ be random variables, $\mathbf{M} \in \{0, 1\}^{n \times d}$ be a missingness mask and $\mathbf{X}^{obs}$ denotes the observed (incomplete) data. We encode the pair $(\mathbf{X}^{obs}, \mathbf{M})$ with the random variable $\tilde{\mathbf{X}}$ with

$$\tilde{X}_{ij} = \begin{cases} X_{ij}, & M_{ij} = 0, \\ ?, & M_{ij} = 1. \end{cases}$$

Let the change in the information be defined as $\Delta := I(\mathbf{Y}; \tilde{\mathbf{X}}) - I(\mathbf{Y}; \mathbf{X})$, where $I(\cdot; \cdot)$ denotes the mutual information. Then,

1. If the missingness is label-MAR, then $\Delta \leq 0$.

2. If $\mathbf{X} \in \{0, 1\}^{n \times d}$ and the missingness is U-MCAR with missingness probability $\mu$, and $s(\mathbf{X})$ is the sample sparsity as in Definition 2, then

$$- nd\,\mu\,h_2\big(\mathbb{E}[s(\mathbf{X})]\big) \leq \Delta \leq 0,$$

where $h_2(u) = -u \log u - (1 - u) \log(1 - u)$.

**Intuition.** When node features are extremely sparse (e.g., BoW/TF-IDF), the information loss induced by missingness is provably negligible unless missingness is extremely high. As a result, existing sparse benchmarks inherently make all methods appear robust, preventing meaningful comparison.

The proof can be found in Appendix A. Theorem 2 demonstrates that when feature sparsity is high, a very large amount of missingness is required to produce a meaningful loss of information. This confirms that such benchmarks do not meaningfully differentiate between approaches, casting doubt on their suitability for evaluating GNNs under feature missingness. As a consequence, we argue for the use of datasets where missingness poses a real challenge. In particular, we introduce a set of four alternative datasets, one new synthetic and three real-world. More details about the datasets are reported in Appendix C.

**(1) A synthetic dataset tailored to controlled missingness.** We construct a dataset based on a Barabási–Albert graph topology, where node features are sampled from a Gaussian distribution. Node labels are assigned using a fixed two-layer GCN applied to the full, complete features, ensuring that a GNN model has the capacity to achieve high classification accuracy in the absence of missingness. This controlled setting provides a testbed for isolating the effects of missingness under varying sparsity, while maintaining a well-defined ground truth.

**(2) Real-world datasets with semantically meaningful features.** We also advocate for the use of real datasets in which node features correspond to raw, observable properties: 1) **AIR** (Zheng et al., 2015), a sensor network dataset from IoT applications, where node features correspond to environmental measurements and node labels indicate sensor status categories; 2) **ELECTRIC** (Birchfield et al., 2016; Baek & Birchfield, 2023), a dataset of interconnected electrical sensors, with real-valued measurements as features and operational condition classification as the target task; 3) **TADPOLE** (Zhu et al., 2019), a medical graph dataset derived from the TADPOLE challenge, where each node represents a patient, node features include clinical and imaging biomarkers, and the goal is to predict diagnostic labels.

Table 2: Evaluation of P1 (feature-structure separability) and P2 (feature-structure complementarity) on our custom datasets. Each cell reports the KS statistic and associated $p$-value for separability under six perturbation settings. $\gamma_{1,1}$ indicates the feature-structure complementarity. Datasets satisfying each property (as per Coupette et al. (2025)) are marked with ✓.

| Dataset | Empty Feat. | Random Feat. | Complete Feat. | Empty Graph | Random Graph | Complete Graph | $\gamma_{1,1}$ | P1 | P2 |
|---|---|---|---|---|---|---|---|---|---|
| SYNTHETIC | 1.00 (8.80e-62) | 1.00 (8.80e-62) | 1.00 (1.93e-14) | 1.00 (1.03e-17) | 1.00 (8.80e-62) | 1.00 (8.80e-62) | 0.62 | ✓ | ✓ |
| AIR | 1.00 (8.80e-62) | 1.00 (8.80e-62) | 1.00 (8.80e-62) | 0.67 (1.53e-30) | 1.00 (8.80e-62) | 1.00 (8.80e-62) | 0.68 | ✓ | ✓ |
| ELECTRIC | 1.00 (8.80e-62) | 1.00 (8.80e-62) | 1.00 (8.80e-62) | 0.98 (1.90e-57) | 1.00 (8.80e-62) | 1.00 (8.80e-62) | 0.69 | ✓ | ✓ |
| TADPOLE | 1.00 (8.80e-62) | 0.90 (5.31e-44) | 0.61 (4.22e-18) | 0.77 (1.53e-30) | 1.00 (8.80e-62) | 1.00 (8.80e-62) | 0.64 | ✓ | ✓ |

Both the synthetic and real-world datasets exhibit low feature sparsity (Table 1), a necessary condition for studying missingness. However, sparsity alone is not sufficient: suitable datasets must also ensure that both features and structure are task-informative and interact non-trivially. We assess this using the RINGS framework (Coupette et al., 2025), which measures performance separability via KS statistics under perturbations (e.g., removing all edges or replacing features with noise), and features-topology complementarity via the normalized Gromov–Wasserstein distance $\gamma_{1,1}$ between the structural and feature-induced metric spaces (values above 0.5 are considered satisfacotry). As shown in Table 2, all proposed datasets satisfy both mode complementarity and performance separability. Combined with their low feature sparsity, these properties make the datasets more suitable than traditional benchmarks for evaluating robustness to incomplete node attributes.

While the real-world datasets we introduce have moderate numbers of nodes and features (Table 3), they satisfy the three key requirements for evaluating robustness to missing node attributes: (i) dense, semantically meaningful, low-dimensional features; (ii) non-trivial predictive signal under complete information; and (iii) complementary and separable contributions of features and structure. To the best of our knowledge, no existing large-scale graph datasets simultaneously meet all these criteria. This limitation is structural to current benchmarks and has been noted in recent work (Bechler-Speicher et al., 2025). Importantly, the effect of missingness on model performance does not depend on graph size: in Appendix E we replicate our experiments on a larger variant of the SYNTHETIC dataset (both in number of nodes and features) and observe trends fully consistent with those reported in the main analysis.

## 4 BEYOND UNIFORM MISSINGNESS

Dataset suitability is only one dimension of the evaluation problem. A second, equally important factor is the choice of the missingness mechanism under which models are tested. In the literature, nearly all prior works adopt a masking scheme based on *U-MCAR* mechanism. In other works (Taguchi et al., 2021; Um et al., 2023), a different variant is used where entire feature vectors of randomly selected nodes are masked. We denote this as ***Structural MCAR (S-MCAR)***. These two settings have become the default evaluation standards in the context of graph learning. We argue that more challenging and realistic missing data patterns need to be considered for a more infor-

mative evaluation of different methods' capabilities. We first introduce a more challenging MCAR mechanism:

***Label–Dependent MCAR (LD-MCAR).*** Missingness here is applied at the feature (column) level, assigning higher missingness probability to features $X_j$ that are more informative for the label, as measured by the mutual information $I(X_j; Y)$. Then, each entry $X_{ij}$ is masked independently with probability $P(M_{ij} = 1) = \rho \cdot I(X_j; Y)$, where $\rho \in [0, 1]$ is a scaling factor selected to achieve the overall desired expected missingness rate across the dataset. Importantly, this mechanism is still MCAR: the probability that a specific entry is missing does not depend on the actual value of the feature or the label, but only on the mutual information of the feature column and the label.

Outside of graph learning, authors have also emphasized the importance of MAR and MNAR mechanisms that reflect more realistically the kinds of missingness encountered in real-world applications(Ghorbani & Zou, 2018; Mohan & Pearl, 2021; Jaeger, 2022; Van Ness et al., 2023). In many practical scenarios, missing features are indeed related to their values or to the prediction target. For instance, a patient might be less likely to report their weight if it is above a certain threshold. This corresponds to a Missing Not At Random (MNAR) mechanism (Rubin, 1976). Testing GNN models exclusively under MCAR conditions thus fails to capture the challenge of more realistic settings. We therefore propose two different MNAR scenarios:

***Feature-Dependant MNAR (FD-MNAR).*** In this mechanism the probability of missingness depends on the value of the feature itself. In particular, we assume that extreme feature values, e.g., high quantiles, are more likely to be missing, as often observed in real-world settings such as healthcare, where abnormal values may be withheld. Formally, for each feature column $j$, let $q_j^{(\tau)}$ denote the $\tau$-quantile of the observed values. We define the missingness probability for entry $X_{ij}$ as:

$$P(M_{ij} = 1) = \begin{cases} \mu^{\text{hi}} & \text{if } X_{ij} \geq q_j^{(\tau)}, \\ \mu^{\text{lo}} & \text{otherwise,} \end{cases}$$

with $\mu^{\text{hi}} > \mu^{\text{lo}}$ and both chosen selected to match a desired overall missingness rate.

***Class–Dependent MNAR (CD-MNAR).*** In this mechanism, features whose values are informative for the label, are more likely to be omitted. For example, in medical datasets, patients may be less likely to disclose whether they smoke, a feature strongly associated with the label indicating a history of heart attack. To identify such dependencies, we train a decision tree classifier in a one-vs-rest setting, using the observed features to predict class membership. For each class $c \in \{1, \ldots, C\}$, we extract decision paths that lead to leaf nodes predicting $c$. These paths define a set of feature-value conditions that contribute to the prediction of class $c$, which we denote as $\mathcal{R}_c$. Let $\text{Cond}_c(j, X_{ij})$ be a predicate that evaluates to true if the value of feature $j$ for node $i$ satisfies at least one condition in $\mathcal{R}_c$. Then, the missingness probability is defined as:

$$P(M_{ij} = 1 \mid Y_i = c) = \begin{cases} \mu^{\text{hi}} & \text{if } \text{Cond}_c(j, X_{ij}) = \text{true}, \\ \mu^{\text{lo}} & \text{otherwise,} \end{cases}$$

where $\mu^{\text{hi}} > \mu^{\text{lo}}$, and both are selected to meet a target overall missingness rate.

In almost all existing experimental studies the missingness mechanism is the same in training and test data. An exception is (Ding & Simonoff, 2010), where two types of test data are considered: data that underlies the same missingness as the training data, and complete data. We consider a possible distribution shift in $P_\lambda(M|X, Y)$ to be an important concern for two reasons: first, it represents a realistic scenario in practical applications. For instance, training data may consist of historical records collected over time, which may contain missing features due to manual entry or outdated systems. In contrast, test data are collected in real time with modern infrastructure, and all feature values are available. This results in a shift from incomplete to complete data between training and testing. The second reason for considering distribution shifts in $P_\lambda$ is to assess a possible weakness of GNNmim: as a model of the form $P_{\gamma+}(Y|X^{obs}, M)$ it explicitly incorporates a model of the missingness mechanism, and thereby could be expected to be less robust under missingness distribution shifts than models that are based on MAR assumptions and (6) (which would be expected to be robust as long as the mechanism is feature and label MAR in both training and test data). We therefore define two evaluation regimes (R1 and R2) with and without a shift in the

missingness process. Let $\mu_{\mathrm{tr}}(\mathbf{M} \mid \mathbf{X}, \mathbf{Y})$ and $\mu_{\mathrm{te}}(\mathbf{M} \mid \mathbf{X}, \mathbf{Y})$ denote the missingness distributions in training and testing, respectively.

**R1:** *i.i.d. missingness* (**no shift**). The same missingness mechanism (*U-MCAR*, *S-MCAR*, *LD-MCAR*, *FD-MNAR*, *CD-MNAR*) and rate are applied to training and test data, i.e., $\mu_{\mathrm{tr}} = \mu_{\mathrm{te}}$.

**R2:** *missingness distribution shift* (**train $\neq$ test**). In this setting, we evaluate combinations of a training missingness mechanism $M_{\mathrm{tr}} \in \{FD\text{-}MNAR, CD\text{-}MNAR\}$ with missingness probability $\mu_{\mathrm{tr}} = 50\%$, and a test missingness mechanism $M_{\mathrm{te}} = U\text{-}MCAR$ with missingness probability $\mu_{\mathrm{te}} \in \{0\%, 25\%, 50\%\}$.

## 5 Experimental Results

We conduct experiments on node classification task using the datasets introduced in Section 3 and the more realistic missingness protocols described in Section 4. We compare a range of GNN-based models specifically designed to handle missing features described in Section 2, namely GNNzero, GNNmedian, GNNmi, GCNmf, GOODIE, GSPN, PCFI, FP, and FairAC as well as our proposed method, GNNmim. Following the evaluation protocol adopted by these competitors, we perform all main experiments in a transductive setting. However, we note that GNNmim can also be applied in an inductive scenario; for completeness, in Appendix H we report additional experiments conducted under an inductive setting. For all the experiments, we decide to treat the specific GNN layer type in GNNmim as a hyperparameter. Full implementation details and hyperparameter settings are provided in Appendix D. The code is provided in the supplementary material. The experiments are designed to answer the following research questions:

- **Q1**: Do the datasets of Section 3 provide new and complementary insights regarding the robustness of GNNs under varying rates of missing features?
- **Q2:** How robust are different models for handling incomplete features to different types of missingness?
- **Q3**: Do different models maintain their performance under distribution shifts in missingness between training and test sets?

**Q1:** To assess the impact of the dataset on evaluating robustness under different missingness rates, we compute the F1 score for each model as a function of the missingness rate $\mu$. Figure 1 reports these curves under *Structural MCAR (S-MCAR)* under R1 regimes (see Section 4) for both the standard benchmarks (CORA, CITESEER, PUBMED) and the datasets we propose (ELECTRIC, AIR, TADPOLE, and SYNTHETIC). Results for other missingness mechanisms lead to equal conclusions and are included in Appendix B.

On CORA, CITESEER, PUBMED, all models appear robust, as their F1 score remains high across a wide range of $\mu$, and only drops at very high missingness rates (85-90%). In contrast, on our proposed datasets, performance drops much earlier, often already at low missingness rates. On TADPOLE, the degradation is less pronounced at low $\mu$ overall; however, two models, GOODIE and GSPN, notably diverge from the rest, showing much weaker performance even with limited missingness.

These results show that evaluating robustness solely on traditional benchmarks may lead to overly optimistic conclusions on the robustness of the methods. To properly assess the behavior of GNNs under different missing rates, it is essential to use more challenging datasets.

**Q2:** To assess robustness across mechanisms, we compute the area under the F1–missingness curve (AUC) for each dataset, model, and missingness mechanism under R1 regimes (complete F1 results by model, dataset, missingness rate, and mechanism are reported in Appendix F).

Figure 2 reports the AUC scores as heatmaps, where lighter colors indicate better model performance for each mechanism within each dataset. We observe that many existing methods exhibit strong sensitivity to the missingness type. For example, FairAC performs well under *S-MCAR* settings on ELECTRIC (0.870 AUC, ranking first among all the models), but its performance degrades significantly under *FD-MNAR* on SYNTHETIC (0.641, ranking second-last). Similarly, GOODIE ranks highest on SYNTHETIC with uniform missingness (0.771), yet drops to 0.587 under *CD-MNAR*.

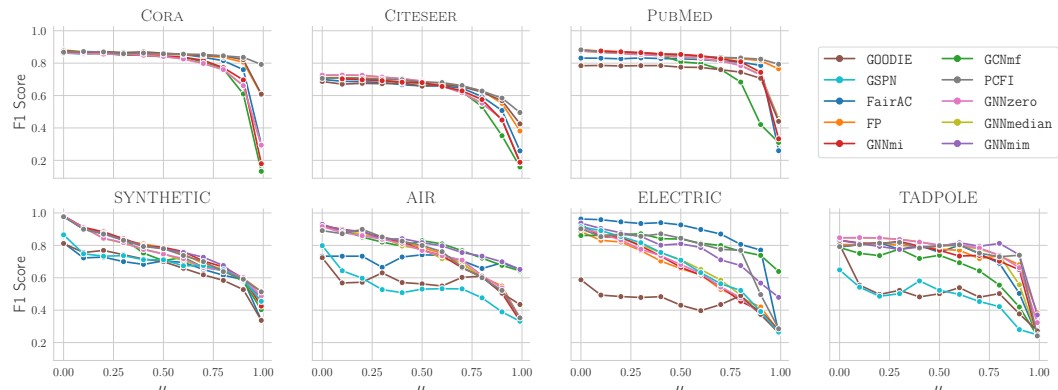

Figure 1: Mean F1-score across 5 runs as a function of the missingness probability $\mu$ on the proposed datasets and established benchmarks. Each panel reports the performance of all models on a specific dataset under the **S-MCAR** setting. The complete tables for all missingness mechanisms are provided in Appendix B.

These results confirm that performance under *U-MCAR* is not predictive of robustness under more realistic *FD-MNAR* scenarios. This calls into question the validity of evaluations based only on uniform or structure-based missingness. Our proposed method, GNNmim, exhibits consistently high AUC across all missingness types and datasets. These results suggest that broad robustness to diverse and realistic missingness mechanisms is achievable, even with lightweight models that do not rely on any MAR assumptions.

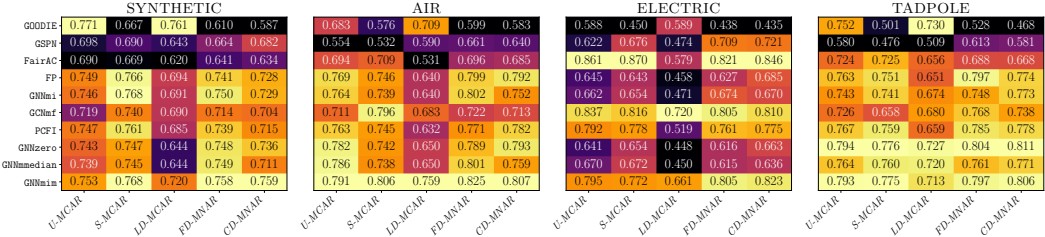

Figure 2: Column-normalized heatmaps showing the AUC (area under the F1 vs. missingness rate $\mu$ curve) for each model, dataset, and missingness mechanism. Higher values (lighter colors) indicate better overall robustness across increasing levels of missingness.

**Q3:** To evaluate model robustness under distribution shifts in missingness, we compute the F1 score (mean ± standard deviation over 5 runs) for each dataset, model, and shift configuration of the R2 regime (Section 4). Full results are in Appendix G; Figure 3 shows a representative subset of the best-performing models from Q2 (GNNmim, GNNmi, GCNmf, FP, PCFI), trained on *FD-MNAR* with $\mu_{\text{tr}} = 50\%$ and tested on *U-MCAR* with $\mu_{\text{te}} \in \{0\%, 25\%, 50\%\}$. Similar results hold for other models and for the case where the training missing mechanisms is *CD-MNAR* (Appendix G).

Each panel shows one dataset, with F1 on the x-axis, models on the y-axis, and color indicating $\mu_{\text{te}}$ (yellow 0%, blue 25%, green 50%). Dots show mean F1, horizontal lines the standard deviation, and the red vertical bar marks the results obtained in the regime R1 with *FD-MNAR* mechanism on both training and test and $\mu_{\text{tr}} = \mu_{\text{te}} = 50\%$. We observe two findings.

1. Distribution shift generalization is challenging: in almost all cases, performance under R2 test conditions *U-MCAR* 25% is lower than in the i.i.d. R1 setting, despite the test missingness being less severe. This is visible when the blue dot ($\mu_{\text{te}} = 25\%$) lies to the left of the red vertical bar ($\mu_{\text{tr}} = \mu_{\text{te}} = 50\%$). This shows that distribution shifts in missingness create a harder generalization challenge that is not explained solely by missingness severity. The effect

is also dataset-dependent, further reinforcing the need to evaluate robustness under these shifts and under different datasets.

2. `GNNmim` is competitive with respect to other models even under R2 conditions. Across datasets and levels of test missingness, `GNNmim` tends to achieve the highest F1 scores (i.e., yellow, blue, and green dots are consistently farther to the right). In spite of its potential vulnerability in the R2 setting, `GNNmim` is seen to maintain its advantage over the alternative approaches.

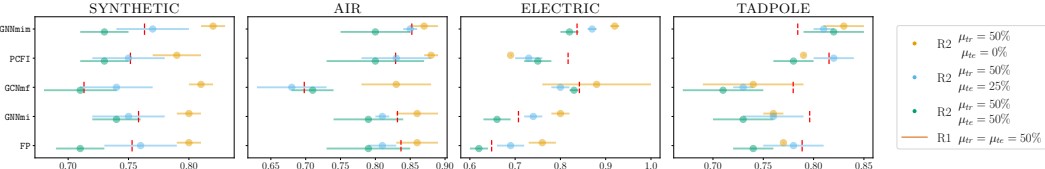

Figure 3: F1 scores (mean ± std over 5 runs) under distribution shifts in missingness between training and test data. All models are trained with *FD-MNAR* missingness at 50%. Each panel corresponds to a dataset; each row to a model. Colored dots represent test-time F1 under *U-MCAR* with varying missingness rates: yellow = 0%, blue = 25%, green = 50%. Vertical red lines indicate the F1 achieved in the i.i.d. setting (*FD-MNAR* 50% at both train and test).

# 6    CONCLUSION AND FUTURE WORK

We revisited the problem of learning GNNs under missing node features, highlighting fundamental limitations of current evaluation protocols, namely the reliance on benchmarks with sparse features and oversimplified missingness mechanisms. To address these issues, we introduced new datasets with dense, informative features and more realistic missingness patterns that go beyond MCAR, and proposed `GNNmim`, a simple yet effective method that explicitly models missingness through the missing-indicator approach. Our experiments show that `GNNmim` is competitive with respect to more complex architectures across diverse datasets, missingness types, and train–test shifts. This work calls for a shift towards more realistic evaluation settings and demonstrates that lightweight yet principled strategies can achieve strong robustness in challenging missing-feature scenarios.

As a direction for future work, our study underscores the need for larger and more diverse benchmarks specifically designed for missing features, aligning with recent calls for better datasets in graph learning (Bechler-Speicher et al., 2025), and reveals that there remains substantial room for developing models that are robust to diverse rates and types of missingness. Another promising direction concerns the development of more realistic MNAR mechanisms, potentially incorporating graph-specific dependencies where missingness is influenced by structural properties of the graph itself. Designing richer, structurally grounded MNAR processes would allow for more faithful stress-testing of models in settings that better reflect more complex patterns.

## USE OF LARGE LANGUAGE MODELS (LLMS)

We used LLMs to improve the readability of the manuscript, rephrase selected passages, and assist in code debugging. All content was initially written by the authors, with LLMs employed solely to enhance clarity and presentation.

## ETHICS STATEMENT

Our study does not involve human subjects or personally identifiable data. The datasets used are publicly available benchmarks or synthetically generated. We follow the ICLR Code of Ethics and note that our work raises no foreseeable ethical concerns beyond those inherent to the general study of machine learning with missing data.

## REPRODUCIBILITY STATEMENT

We have made every effort to ensure reproducibility. Details of the experimental setup are provided in Section 5, with dataset descriptions in Appendix 3 and complete training configurations in Appendix D. All proofs are included in Appendix A. Anonymous source code to reproduce our experiments is provided in the supplementary material.

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

## A    PROOFS

**Theorem 1.** If $P_{\boldsymbol{\theta},\boldsymbol{\gamma},\boldsymbol{\lambda}}$ is feature-MAR and label-MAR, then (2) simplifies to

$$\int_{\mathbf{X}^{miss}} P_{\boldsymbol{\gamma}}(\boldsymbol{Y}|\boldsymbol{X})P_{\boldsymbol{\theta}}(\mathbf{X}^{miss}|\mathbf{X}^{obs}). \tag{6}$$

*Proof.*

$$P_{\boldsymbol{\theta},\boldsymbol{\gamma},\boldsymbol{\lambda}}(\boldsymbol{Y}|\boldsymbol{X},\boldsymbol{M}) = P_{\boldsymbol{\lambda}}(\boldsymbol{M}|\boldsymbol{X},\boldsymbol{Y})\frac{P_{\boldsymbol{\gamma}}(\boldsymbol{Y}|\boldsymbol{X})}{P_{\boldsymbol{\gamma},\boldsymbol{\lambda}}(\boldsymbol{M}|\boldsymbol{X})} \stackrel{(4)}{=} P_{\boldsymbol{\gamma}}(\boldsymbol{Y}|\boldsymbol{X})$$

$$P_{\boldsymbol{\theta},\boldsymbol{\gamma},\boldsymbol{\lambda}}(\mathbf{X}^{miss}|\mathbf{X}^{obs},\boldsymbol{M}) = P_{\boldsymbol{\gamma},\boldsymbol{\lambda}}(\boldsymbol{M}|\mathbf{X}^{obs},\mathbf{X}^{miss})\frac{P_{\boldsymbol{\theta}}(\mathbf{X}^{miss}|\mathbf{X}^{obs})}{P_{\boldsymbol{\theta},\boldsymbol{\gamma},\boldsymbol{\lambda}}(\boldsymbol{M}|\mathbf{X}^{obs})} \stackrel{(3)}{=} P_{\boldsymbol{\theta}}(\mathbf{X}^{miss}|\mathbf{X}^{obs})$$

$\square$

**Theorem 2.** Let $\mathbf{X} \in \mathbb{R}^{n \times d}$ and $\mathbf{Y} \in \mathcal{Y}^n$ be random variables, $\mathbf{M} \in \{0,1\}^{n \times d}$ be a missingness mask and $\mathbf{X}^{obs}$ denotes the observed (incomplete) data. We encode the pair $(\mathbf{X}^{obs}, \mathbf{M})$ with the random variable $\tilde{\mathbf{X}}$ with

$$\tilde{X}_{ij} = \begin{cases} X_{ij}, & M_{ij} = 0, \\ ?, & M_{ij} = 1. \end{cases}$$

Let the change in the information be defined as $\Delta := I(\mathbf{Y};\tilde{\mathbf{X}}) - I(\mathbf{Y};\mathbf{X})$, where $I(\cdot;\cdot)$ denotes the mutual information. Then,

1. If the missingness is label-MAR, then $\Delta \leq 0$.

2. If $\mathbf{X} \in \{0,1\}^{n \times d}$ and the missingness is U-MCAR with missingness probability $\mu$, and $s(\mathbf{X})$ is the sample sparsity as in Definition 2, then

$$- nd\,\mu\,h_2\big(\mathbb{E}[s(\mathbf{X})]\big) \leq \Delta \leq 0,$$

where $h_2(u) = -u\log u - (1-u)\log(1-u)$.

*Proof.* By construction $\tilde{\mathbf{X}} = g(\mathbf{X},\mathbf{M})$ for some measurable $g$. Thus $(\mathbf{Y}) \to (\mathbf{X},\mathbf{M}) \to \tilde{\mathbf{X}}$ is a Markov chain, and the data–processing inequality implies

$$I(\mathbf{Y};\tilde{\mathbf{X}}) \leq I(\mathbf{Y};\mathbf{X},\mathbf{M}). \tag{7}$$

Moreover, for any three random elements $(A,B,C)$ we have the chain–rule identities

$$I(A;B,C) = I(A;C) + I(A;B \mid C). \tag{8}$$

**(1) Label-MAR** $\Delta \leq 0$**.** Assume label-MAR: $\mathbb{P}(\mathbf{M} \mid \mathbf{X},\mathbf{Y}) = \mathbb{P}(\mathbf{M} \mid \mathbf{X})$, which is equivalent to $\mathbf{Y} \perp \mathbf{M} \mid \mathbf{X}$. Applying equation 8 with $(A,B,C) = (\mathbf{Y},\mathbf{X},\mathbf{M})$,

$$I(\mathbf{Y};\mathbf{X},\mathbf{M}) = I(\mathbf{Y};\mathbf{X}) + I(\mathbf{Y};\mathbf{M} \mid \mathbf{X}).$$

Under label-MAR, $I(\mathbf{Y};\mathbf{M} \mid \mathbf{X}) = 0$, hence

$$I(\mathbf{Y};\mathbf{X},\mathbf{M}) = I(\mathbf{Y};\mathbf{X}). \tag{9}$$

Combining equation 7 and equation 9 yields

$$I(\mathbf{Y};\tilde{\mathbf{X}}) \leq I(\mathbf{Y};\mathbf{X}) \iff \Delta = I(\mathbf{Y};\tilde{\mathbf{X}}) - I(\mathbf{Y};\mathbf{X}) \leq 0.$$

**(2) Two-sided bound under uniform MCAR and $\alpha$-$\beta$ sparsity.** Assume uniform MCAR: $M_{ij} \sim$ Bernoulli$(1-\mu)$ independently of $(\mathbf{X},\mathbf{Y})$ and i.i.d. across $(i,j)$, and that $\mathbb{P}\big(s(\mathbf{X}) \geq \alpha\big) \geq \beta$, where $s(\mathbf{X}) = \frac{1}{nd}\sum_{i,j}\mathbb{I}\{X_{ij} = 0\}$.

*Upper side.* MCAR implies label-MAR, so by part (1): $\Delta \leq 0$.

*Lower side.* We start from the chain–rule identity applied to $(A, B, C) = (\mathbf{Y}, \mathbf{X}, \tilde{\mathbf{X}})$:

$$I(\mathbf{Y}; \mathbf{X}, \tilde{\mathbf{X}}) = I(\mathbf{Y}; \tilde{\mathbf{X}}) + I(\mathbf{Y}; \mathbf{X} \mid \tilde{\mathbf{X}}) = I(\mathbf{Y}; \mathbf{X}) + I(\mathbf{Y}; \tilde{\mathbf{X}} \mid \mathbf{X}).$$

Rearranging gives

$$-\Delta = I(\mathbf{Y}; \mathbf{X}) - I(\mathbf{Y}; \tilde{\mathbf{X}}) = I(\mathbf{Y}; \mathbf{X} \mid \tilde{\mathbf{X}}) - I(\mathbf{Y}; \tilde{\mathbf{X}} \mid \mathbf{X}). \tag{10}$$

The second term on the right is nonnegative, hence

$$-\Delta \leq I(\mathbf{Y}; \mathbf{X} \mid \tilde{\mathbf{X}}). \tag{11}$$

Using the bound $I(U; V \mid W) \leq H(V \mid W)$, we get

$$-\Delta \leq H(\mathbf{X} \mid \tilde{\mathbf{X}}). \tag{12}$$

Index the matrix entries by a total order $\prec$ on pairs $(i, j)$ and apply the chain rule:

$$H(\mathbf{X} \mid \tilde{\mathbf{X}}) = \sum_{(i,j)} H\big(X_{ij} \,\big|\, \tilde{\mathbf{X}}, \{X_{kl} : (k,l) \prec (i,j)\}\big).$$

Since conditioning reduces entropy,

$$H(\mathbf{X} \mid \tilde{\mathbf{X}}) \leq \sum_{i,j} H\big(X_{ij} \mid \tilde{X}_{ij}\big). \tag{13}$$

Fix $(i, j)$ and denote $\pi_{ij} = \Pr[X_{ij} = 1]$. Under uniform MCAR,

$$\Pr[\tilde{X}_{ij} =?] = \mu, \qquad \Pr[\tilde{X}_{ij} = x] = (1 - \mu) \Pr[X_{ij} = x], \quad x \in \{0, 1\}.$$

Hence: (i) if $\tilde{X}_{ij} \in \{0, 1\}$ then $X_{ij}$ is revealed, so $H(X_{ij} \mid \tilde{X}_{ij} \in \{0, 1\}) = 0$; (ii) if $\tilde{X}_{ij} =?$, then $\Pr[X_{ij} = 1 \mid \tilde{X}_{ij} =?] = \pi_{ij}$ and $H(X_{ij} \mid \tilde{X}_{ij} =?) = h_2(\pi_{ij})$. Averaging over $\tilde{X}_{ij}$ gives

$$H(X_{ij} \mid \tilde{X}_{ij}) = \mu \, h_2(\pi_{ij}). \tag{14}$$

Combining equation 13 and equation 14:

$$H(\mathbf{X} \mid \tilde{\mathbf{X}}) \leq \sum_{i,j} \mu \, h_2(\pi_{ij}) = nd \, \mu \cdot \frac{1}{nd} \sum_{i,j} h_2(\pi_{ij}) \leq nd \, \mu \cdot h_2 \left( \frac{1}{nd} \sum_{i,j} \pi_{ij} \right),$$

since $h_2$ is concave. Note that

$$\frac{1}{nd} \sum_{i,j} \pi_{ij} = \frac{1}{nd} \sum_{i,j} \Pr[X_{ij} = 1] = \mathbb{E} \left[ \frac{1}{nd} \sum_{i,j} \mathbb{I}\{X_{ij} = 1\} \right] = 1 - \mathbb{E}[s(\mathbf{X})].$$

Using the symmetry $h_2(u) = h_2(1 - u)$, we conclude

$$H(\mathbf{X} \mid \tilde{\mathbf{X}}) \leq nd \, \mu \cdot h_2\big(\mathbb{E}[s(\mathbf{X})]\big).$$

Combining with $-\Delta \leq H(\mathbf{X} \mid \tilde{\mathbf{X}})$ gives

$$- nd \, \mu \, h_2\big(\mathbb{E}[s(\mathbf{X})]\big) \leq \Delta \leq 0.$$

This concludes the proof. $\qquad\square$

## B  ADDITIONAL RESULTS ON BENCHMARKS AND PROPOSED DATASETS

This section presents the full plots of the results under the R1 regime introduced in Section 4.

Figure 4 shows the complete set of results across all datasets, whose statistics are summarized in Table 3. The top three rows correspond to the classic benchmarks (CORA, CITESEER, PUBMED). Consistently with Proposition 2, models maintain nearly constant F1 scores up to extremely high missingness levels ($\sim 90\%$), confirming that these benchmarks are of limited value for evaluating robustness to missing features.

The bottom four rows correspond to our proposed datasets (SYNTHETIC, AIR, ELECTRIC, TAD-POLE). In these cases, performance degrades much earlier and more severely, highlighting the higher realism and difficulty of our benchmarks.

Table 3: Dataset statistics and feature sparsity. Classic benchmarks (CORA, CITESEER, PUBMED) exhibit extremely sparse bag-of-words features, while our proposed datasets (SYNTHETIC, AIR, ELECTRIC, TADPOLE) provide less sparse representations.

| Dataset | #Nodes | #Features | Sparsity ↓ | Type of features |
|---|---|---|---|---|
| CORA | 2708 | 1433 | 0.9873 | BoW (binary) |
| CITESEER | 3327 | 3703 | 0.9915 | BoW (binary) |
| PUBMED | 19717 | 500 | 0.8998 | BoW (binary) |
| SYNTHETIC | 1000 | 5 | 0.0000 | Gaussian |
| AIR | 430 | 7 | 0.1615 | Raw |
| ELECTRIC | 2000 | 5 | 0.2000 | Raw |
| TADPOLE | 555 | 15 | 0.0000 | Raw |

## C  MORE CHALLENGING DATASETS

In Section 3, we introduced the synthetic and real-world datasets employed in our experiments. We now provide additional details on their construction and characteristics.

**SYNTHETIC**   Synthetic dataset based on a Barabási–Albert graph topology. Each node is associated with five real-valued features sampled from a Gaussian distribution. Node labels are generated deterministically by applying a fixed two-layer GCN with hard-coded weights to the complete feature matrix. This construction ensures that the ground-truth labeling function is fully expressible by a GNN, allowing models to achieve near-perfect accuracy in the absence of missingness. The resulting task is a binary node classification problem, with classes separated according to structured feature combinations defined by the fixed GCN. This controlled setup provides a principled testbed to isolate and analyze the effects of different missingness mechanisms, while preserving a well-defined ground truth.

**AIR**   Dataset (Zheng et al., 2015) built from a network of air quality monitoring stations deployed in an urban area. Each node corresponds to a station and is associated with a set of environmental measurements. The node features include both air pollutant concentrations ($CO$, $NO_2$, $PM_{10}$, $O_3$, $SO_2$) and meteorological variables (`temperature`, `humidity`, `wind speed`, `wind direction`). Edges are constructed based on the geographical distance between stations, with two nodes connected if their distance is below a given threshold. The target variable is derived from the $PM_{2.5}$ concentration, which is discretized into three balanced categories (low, medium, high) according to the distribution of observed values. This formulation allows us to frame the problem as a semi-supervised node classification task with three classes.

**ELECTRIC**   Dataset (Birchfield et al., 2016; Baek & Birchfield, 2023) derived from a large-scale model of the Texas power grid. Nodes correspond to buses in the electrical network, each enriched with both structural and operational attributes. The node features include identifiers (`area`, `zone`), electrical measurements (`voltage magnitude`, `voltage angle`), and a topological property (`betweenness centrality`). Edges are constructed directly from the transmission lines specified in the raw grid data, connecting pairs of buses. The classification target is the nominal voltage level of each bus (`base kV`), which we discretize into three categories: low voltage ($<100$ kV), medium voltage (100–200 kV), and high voltage ($>200$ kV). This setup results in a three-class node classification problem reflecting operational conditions across the grid.

**TADPOLE**   The TADPOLEdataset (Zhu et al., 2019) originates from the TADPOLE challenge, which provides longitudinal clinical and imaging data for patients at risk of developing Alzheimer's disease. In our graph formulation, each node corresponds to a patient and is associated with a set of features encompassing clinical scores, cerebrospinal fluid (CSF) biomarkers, and neuroimaging measures such as MRI- and PET-derived variables. Since the original dataset does not provide graph connectivity, we construct edges using a $k$-nearest neighbors approach over the most informative biomarkers, so that patients with similar profiles are connected. The target variable is the diagnostic label, categorized into three classes (cognitively normal, mild cognitive impairment, Alzheimer's disease). This results in a semi-supervised node classification problem where the goal is to pre-

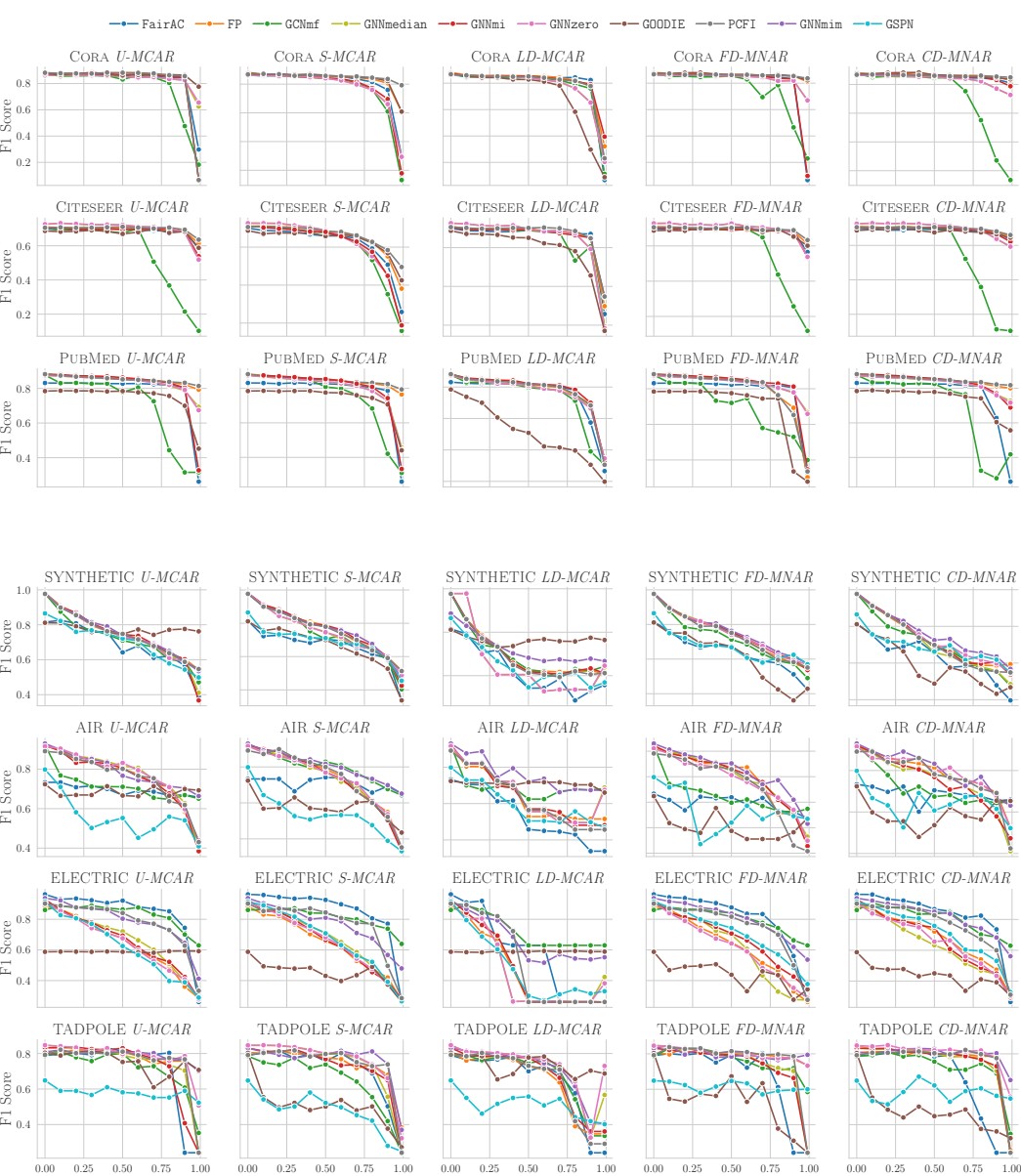

Figure 4: F1 score as a function of feature missingness ($\mu$) for both classic benchmarks (top three rows) and our proposed datasets (bottom four rows), under the mechanisms described in Section 4. Classic benchmarks show almost no degradation until extremely high $\mu$, while the proposed datasets reveal model weaknesses at more realistic missingness levels. Tables for numeric results are in App. F

dict the diagnostic status of patients based on multimodal biomedical features and patient similarity structure.

Table 3 reports, for each dataset, the number of nodes, number of features, feature sparsity, and the type of features. While the number of nodes and features may seem small compared to standard benchmark graph datasets, we emphasize that using real features (as in AIR, ELECTRIC, and TAD-POLE) is more realistic in the context of feature missingness. In fact, it is not meaningful to study missingness on pre-computed embeddings, since embeddings are typically high-dimensional representations mapped to wide feature spaces and are not expected to exhibit missingness in practice.

## D  EXPERIMENTAL DETAILS

All baseline and competitor methods are implemented using the official code released in their respective repositories, following the recommended training protocols and hyperparameter settings. For GNNmi and GNNmim, we adopt a standard GNN architecture where the convolutional layer type (Table 4), the number of layers (1-3), the learning rate ($10^{-4}$-$10^{-2}$), and the weight decay ($10^{-5}$-$10^{-3}$) are tuned via grid search on the validation set. All models are trained on the same data splits with early stopping to ensure a fair comparison.

Table 4: Best GNN encoder selected within GNNmim for each dataset and missingness mechanism.

| Dataset | U-MCAR | S-MCAR | LD-MCAR | FD-MNAR | CD-MNAR |
|---|---|---|---|---|---|
| SYNTHETIC | GCN | GCN | GraphSAGE | GCN | GCN |
| AIR | GraphSAGE | GraphSAGE | GraphSAGE | GraphSAGE | GraphSAGE |
| ELECTRIC | GIN | GIN | GraphSAGE | GIN | GIN |
| TADPOLE | GCN | GraphSAGE | GraphSAGE | GraphSAGE | GCN |

## E  SCALING THE SYNTHETIC DATASET

In this section, we analyze what happens when either the number of features or the number of nodes in the synthetic dataset is increased. To this end, we constructed three additional synthetic datasets (SYNTHETIC2, SYNTHETIC3, SYNTHETIC4) following the same design principles as SYNTHETIC. Table 5 reports their statistics.

As shown in Figure 5, the behavior of the models in this larger-scale setting is consistent with the one observed in our original setup. In this case, we experimented with the *uniform random missingness* mechanism, and we observe a monotonic decrease in performance for all models as the missingness rate $\mu$ increases. This confirms that dataset size does not affect the overall trend of performance degradation under feature missingness.

To further support this point, we also report the runtime and GPU memory consumption of all models on both the main synthetic dataset (SYNTHETIC) and its larger-scale counterpart (SYNTHETIC3), which features an increased number of features. As shown in Table 6, the runtime and memory requirements remain substantially stable across datasets, with negligible variations between models. This behavior confirms that our approach scales efficiently with the dataset size, as it only involves a standard GNN architecture augmented with a simple MIM mask concatenated to the input features, introducing minimal computational overhead.

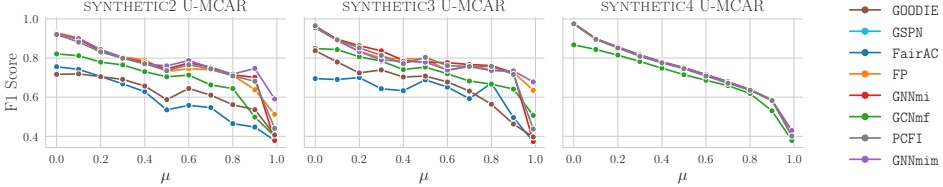

Figure 5: F1 score as a function of feature missingness ($\mu$) for additional synthetic datasets generated with the same procedure as SYNTHETIC, but with either an increased number of nodes or features. For SYNTHETIC4, the  model is not reported since training exceeded the 12-hour time limit, while GOODIE is excluded due to out-of-memory errors.

## F  COMPLETE RESULT TABLES – R1 REGIME

Table 5: Datasets information.

| Dataset | #Nodes | #Features | Sparsity $\downarrow$ | Type of features |
|---|---|---|---|---|
| SYNTHETIC | 1000 | 5 | 0.0000 | Gaussian |
| SYNTHETIC2 | 1000 | 20 | 0.0000 | Gaussian |
| SYNTHETIC3 | 1000 | 50 | 0.0000 | Gaussian |
| SYNTHETIC4 | 50000 | 5 | 0.0000 | Gaussian |

Table 6: Runtime and GPU peak memory consumption for the main synthetic dataset (SYNTHETIC) and the scaled version (SYNTHETIC3). Each value corresponds to the average across all missingness levels under the UMCAR mechanism.

| Model | SYNTHETIC | | SYNTHETIC4 | |
|---|---|---|---|---|
| | Runtime [s] $\downarrow$ | GPU Mem [GB] $\downarrow$ | Runtime [s] $\downarrow$ | GPU Mem [GB] $\downarrow$ |
| GNNmi | 1.7 | 0.03 | 5.3 | 0.78 |
| GNNzero | 1.6 | 0.03 | 5.0 | 0.77 |
| GNNmedian | 1.6 | 0.03 | 5.0 | 0.77 |
| GNNmim | 1.8 | 0.03 | 6.3 | 0.77 |
| GCNmf | 4.5 | 0.02 | 28.0 | 0.53 |
| FP | 1.5 | 0.02 | 5.3 | 0.77 |
| PCFI | 1.8 | 0.02 | 5.2 | 0.77 |
| FairAC | 3.9 | 0.04 | – | – |
| GSPN | 55.0 | 0.03 | 150.0 | 0.84 |
| GOODIE | 2.3 | 0.06 | – | – |

Table 7: F1 scores for CORA under mechanism *U-MCAR* and varying $\mu$ (GSPN is not reported as it is not designed for categorical features).

| $\mu$ | GOODIE | FairAC | FP | GNNmi | GCNmf | PCFI | GNNzero | GNNmedian |
|---|---|---|---|---|---|---|---|---|
| 0.00 | 0.875 $(\pm 0.00)$ | 0.863 $(\pm 0.01)$ | 0.882 $(\pm 0.00)$ | 0.873 $(\pm 0.00)$ | 0.875 $(\pm 0.00)$ | **0.882** $(\pm 0.00)$ | 0.862 $(\pm 0.02)$ | 0.862 $(\pm 0.02)$ |
| 0.10 | 0.867 $(\pm 0.00)$ | 0.866 $(\pm 0.00)$ | 0.877 $(\pm 0.00)$ | 0.876 $(\pm 0.00)$ | 0.856 $(\pm 0.00)$ | **0.878** $(\pm 0.00)$ | 0.868 $(\pm 0.01)$ | 0.868 $(\pm 0.01)$ |
| 0.20 | 0.875 $(\pm 0.00)$ | 0.862 $(\pm 0.00)$ | **0.878** $(\pm 0.00)$ | 0.873 $(\pm 0.00)$ | 0.858 $(\pm 0.00)$ | 0.877 $(\pm 0.00)$ | 0.864 $(\pm 0.02)$ | 0.864 $(\pm 0.02)$ |
| 0.30 | 0.873 $(\pm 0.00)$ | 0.865 $(\pm 0.00)$ | 0.881 $(\pm 0.00)$ | **0.885** $(\pm 0.00)$ | 0.860 $(\pm 0.00)$ | 0.876 $(\pm 0.00)$ | 0.863 $(\pm 0.01)$ | 0.863 $(\pm 0.01)$ |
| 0.40 | 0.869 $(\pm 0.00)$ | 0.857 $(\pm 0.00)$ | 0.878 $(\pm 0.00)$ | 0.873 $(\pm 0.00)$ | 0.860 $(\pm 0.00)$ | **0.884** $(\pm 0.00)$ | 0.860 $(\pm 0.02)$ | 0.860 $(\pm 0.02)$ |
| 0.50 | 0.861 $(\pm 0.00)$ | 0.856 $(\pm 0.00)$ | **0.882** $(\pm 0.00)$ | 0.867 $(\pm 0.00)$ | 0.831 $(\pm 0.00)$ | 0.882 $(\pm 0.00)$ | 0.856 $(\pm 0.01)$ | 0.856 $(\pm 0.01)$ |
| 0.60 | 0.866 $(\pm 0.00)$ | 0.847 $(\pm 0.00)$ | **0.882** $(\pm 0.00)$ | 0.871 $(\pm 0.00)$ | 0.862 $(\pm 0.00)$ | 0.881 $(\pm 0.00)$ | 0.847 $(\pm 0.01)$ | 0.847 $(\pm 0.01)$ |
| 0.70 | 0.866 $(\pm 0.00)$ | 0.858 $(\pm 0.00)$ | 0.869 $(\pm 0.00)$ | 0.865 $(\pm 0.00)$ | 0.847 $(\pm 0.00)$ | **0.877** $(\pm 0.00)$ | 0.849 $(\pm 0.01)$ | 0.849 $(\pm 0.01)$ |
| 0.80 | **0.868** $(\pm 0.00)$ | 0.843 $(\pm 0.00)$ | 0.864 $(\pm 0.00)$ | 0.854 $(\pm 0.00)$ | 0.805 $(\pm 0.00)$ | 0.863 $(\pm 0.00)$ | 0.835 $(\pm 0.01)$ | 0.835 $(\pm 0.01)$ |
| 0.90 | **0.864** $(\pm 0.00)$ | 0.845 $(\pm 0.00)$ | 0.860 $(\pm 0.00)$ | 0.848 $(\pm 0.00)$ | 0.476 $(\pm 0.00)$ | 0.856 $(\pm 0.00)$ | 0.826 $(\pm 0.00)$ | 0.826 $(\pm 0.00)$ |
| 0.99 | **0.776** $(\pm 0.00)$ | 0.298 $(\pm 0.00)$ | 0.066 $(\pm 0.00)$ | 0.066 $(\pm 0.00)$ | 0.183 $(\pm 0.00)$ | 0.065 $(\pm 0.00)$ | 0.655 $(\pm 0.03)$ | 0.625 $(\pm 0.02)$ |

Table 8: F1 scores for CORA under mechanism *S-MCAR* and varying $\mu$ (GSPN is not reported as it is not designed for categorical features).

| $\mu$ | GOODIE | FairAC | FP | GNNmi | GCNmf | PCFI | GNNzero | GNNmedian |
|---|---|---|---|---|---|---|---|---|
| 0.00 | 0.875 $(\pm 0.00)$ | 0.863 $(\pm 0.01)$ | **0.882** $(\pm 0.00)$ | 0.872 $(\pm 0.00)$ | 0.875 $(\pm 0.00)$ | 0.868 $(\pm 0.00)$ | 0.862 $(\pm 0.02)$ | 0.862 $(\pm 0.02)$ |
| 0.10 | 0.868 $(\pm 0.00)$ | 0.857 $(\pm 0.00)$ | 0.869 $(\pm 0.00)$ | 0.862 $(\pm 0.00)$ | 0.869 $(\pm 0.00)$ | **0.872** $(\pm 0.00)$ | 0.862 $(\pm 0.02)$ | 0.862 $(\pm 0.02)$ |
| 0.20 | **0.872** $(\pm 0.00)$ | 0.860 $(\pm 0.00)$ | 0.863 $(\pm 0.00)$ | 0.863 $(\pm 0.00)$ | 0.858 $(\pm 0.00)$ | 0.869 $(\pm 0.00)$ | 0.856 $(\pm 0.02)$ | 0.856 $(\pm 0.02)$ |
| 0.30 | **0.865** $(\pm 0.00)$ | 0.850 $(\pm 0.00)$ | 0.854 $(\pm 0.00)$ | 0.855 $(\pm 0.00)$ | 0.852 $(\pm 0.00)$ | 0.858 $(\pm 0.00)$ | 0.857 $(\pm 0.02)$ | 0.857 $(\pm 0.02)$ |
| 0.40 | **0.870** $(\pm 0.00)$ | 0.857 $(\pm 0.00)$ | 0.859 $(\pm 0.00)$ | 0.848 $(\pm 0.00)$ | 0.848 $(\pm 0.00)$ | 0.862 $(\pm 0.00)$ | 0.849 $(\pm 0.02)$ | 0.849 $(\pm 0.02)$ |
| 0.50 | **0.862** $(\pm 0.00)$ | 0.854 $(\pm 0.00)$ | 0.854 $(\pm 0.00)$ | 0.844 $(\pm 0.00)$ | 0.839 $(\pm 0.00)$ | 0.858 $(\pm 0.00)$ | 0.841 $(\pm 0.01)$ | 0.841 $(\pm 0.01)$ |
| 0.60 | 0.855 $(\pm 0.00)$ | 0.854 $(\pm 0.00)$ | 0.853 $(\pm 0.00)$ | 0.837 $(\pm 0.00)$ | 0.837 $(\pm 0.00)$ | **0.856** $(\pm 0.00)$ | 0.826 $(\pm 0.01)$ | 0.826 $(\pm 0.01)$ |
| 0.70 | 0.847 $(\pm 0.00)$ | 0.836 $(\pm 0.00)$ | 0.845 $(\pm 0.00)$ | 0.817 $(\pm 0.00)$ | 0.807 $(\pm 0.00)$ | **0.854** $(\pm 0.00)$ | 0.798 $(\pm 0.02)$ | 0.798 $(\pm 0.02)$ |
| 0.80 | **0.845** $(\pm 0.00)$ | 0.815 $(\pm 0.00)$ | 0.836 $(\pm 0.00)$ | 0.772 $(\pm 0.00)$ | 0.764 $(\pm 0.00)$ | 0.845 $(\pm 0.00)$ | 0.760 $(\pm 0.02)$ | 0.760 $(\pm 0.02)$ |
| 0.90 | 0.822 $(\pm 0.00)$ | 0.760 $(\pm 0.00)$ | 0.806 $(\pm 0.00)$ | 0.696 $(\pm 0.00)$ | 0.610 $(\pm 0.00)$ | **0.836** $(\pm 0.00)$ | 0.661 $(\pm 0.02)$ | 0.661 $(\pm 0.02)$ |
| 0.99 | 0.609 $(\pm 0.00)$ | 0.300 $(\pm 0.00)$ | 0.606 $(\pm 0.00)$ | 0.179 $(\pm 0.00)$ | 0.132 $(\pm 0.00)$ | **0.792** $(\pm 0.00)$ | 0.294 $(\pm 0.05)$ | 0.294 $(\pm 0.05)$ |

Table 9: F1 scores for CORA under mechanism *CD-MCAR* and varying $\mu$ (GSPNis not reported as it is not designed for categorical features).

| $\mu$ | GOODIE | FairAC | FP | GNNmi | GCNmf | PCFI | GNNzero | GNNmedian |
|---|---|---|---|---|---|---|---|---|
| 0.00 | 0.875 (± 0.00) | 0.863 (± 0.01) | **0.882** (± **0.00**) | 0.873 (± 0.00) | 0.875 (± 0.00) | 0.868 (± 0.00) | 0.862 (± 0.02) | 0.862 (± 0.02) |
| 0.10 | 0.852 (± 0.00) | 0.851 (± 0.00) | **0.862** (± **0.00**) | 0.857 (± 0.00) | 0.846 (± 0.00) | 0.860 (± 0.00) | 0.858 (± 0.02) | 0.858 (± 0.02) |
| 0.20 | 0.843 (± 0.00) | 0.854 (± 0.00) | **0.859** (± **0.00**) | 0.854 (± 0.00) | 0.850 (± 0.00) | 0.855 (± 0.00) | 0.854 (± 0.02) | 0.854 (± 0.02) |
| 0.30 | 0.843 (± 0.00) | 0.856 (± 0.00) | **0.859** (± **0.00**) | 0.855 (± 0.00) | 0.846 (± 0.00) | 0.852 (± 0.00) | 0.853 (± 0.02) | 0.853 (± 0.02) |
| 0.40 | 0.828 (± 0.00) | 0.854 (± 0.00) | **0.858** (± **0.00**) | 0.853 (± 0.00) | 0.838 (± 0.00) | 0.849 (± 0.00) | 0.849 (± 0.02) | 0.849 (± 0.02) |
| 0.50 | 0.828 (± 0.00) | 0.854 (± 0.00) | 0.855 (± 0.00) | **0.855** (± **0.00**) | 0.848 (± 0.00) | 0.852 (± 0.00) | 0.844 (± 0.02) | 0.844 (± 0.02) |
| 0.60 | 0.812 (± 0.00) | 0.847 (± 0.00) | **0.853** (± **0.00**) | 0.844 (± 0.00) | 0.837 (± 0.00) | 0.841 (± 0.00) | 0.825 (± 0.02) | 0.825 (± 0.02) |
| 0.70 | 0.782 (± 0.00) | 0.841 (± 0.00) | **0.842** (± **0.00**) | 0.831 (± 0.00) | 0.822 (± 0.00) | 0.827 (± 0.00) | 0.810 (± 0.02) | 0.810 (± 0.02) |
| 0.80 | 0.584 (± 0.00) | **0.844** (± **0.00**) | 0.822 (± 0.00) | 0.815 (± 0.00) | 0.792 (± 0.00) | 0.818 (± 0.00) | 0.761 (± 0.01) | 0.761 (± 0.01) |
| 0.90 | 0.297 (± 0.00) | **0.824** (± **0.00**) | 0.777 (± 0.00) | 0.793 (± 0.00) | 0.760 (± 0.00) | 0.778 (± 0.00) | 0.653 (± 0.02) | 0.654 (± 0.02) |
| 0.99 | 0.088 (± 0.00) | 0.066 (± 0.00) | 0.322 (± 0.00) | **0.395** (± **0.00**) | 0.113 (± 0.00) | 0.231 (± 0.00) | 0.204 (± 0.03) | 0.204 (± 0.03) |

Table 10: F1 scores for CORA under mechanism *FD-MNAR* and varying $\mu$ (GSPNis not reported as it is not designed for categorical features).

| $\mu$ | GOODIE | FairAC | FP | GNNmi | GCNmf | PCFI | GNNzero | GNNmedian |
|---|---|---|---|---|---|---|---|---|
| 0.00 | 0.875 (± 0.00) | 0.863 (± 0.01) | **0.882** (± **0.00**) | 0.873 (± 0.00) | 0.875 (± 0.00) | 0.868 (± 0.00) | 0.864 (± 0.02) | 0.864 (± 0.02) |
| 0.10 | 0.872 (± 0.01) | 0.862 (± 0.01) | **0.873** (± **0.01**) | 0.868 (± 0.01) | 0.851 (± 0.01) | 0.873 (± 0.00) | 0.862 (± 0.02) | 0.862 (± 0.02) |
| 0.20 | **0.879** (± **0.00**) | 0.870 (± 0.01) | 0.874 (± 0.00) | 0.865 (± 0.01) | 0.853 (± 0.01) | 0.863 (± 0.01) | 0.858 (± 0.01) | 0.858 (± 0.01) |
| 0.30 | **0.880** (± **0.00**) | 0.864 (± 0.01) | 0.869 (± 0.01) | 0.867 (± 0.01) | 0.847 (± 0.01) | 0.864 (± 0.01) | 0.864 (± 0.01) | 0.864 (± 0.01) |
| 0.40 | **0.869** (± **0.01**) | 0.855 (± 0.01) | 0.864 (± 0.01) | 0.856 (± 0.01) | 0.849 (± 0.01) | 0.866 (± 0.01) | 0.858 (± 0.02) | 0.858 (± 0.02) |
| 0.50 | 0.865 (± 0.01) | 0.860 (± 0.01) | **0.866** (± **0.01**) | 0.859 (± 0.01) | 0.854 (± 0.01) | 0.863 (± 0.01) | 0.854 (± 0.02) | 0.854 (± 0.02) |
| 0.60 | **0.866** (± **0.01**) | 0.853 (± 0.01) | 0.865 (± 0.01) | 0.863 (± 0.01) | 0.829 (± 0.02) | 0.864 (± 0.01) | 0.851 (± 0.01) | 0.851 (± 0.01) |
| 0.70 | 0.859 (± 0.01) | 0.847 (± 0.00) | **0.862** (± **0.01**) | 0.853 (± 0.00) | 0.695 (± 0.14) | 0.860 (± 0.00) | 0.846 (± 0.01) | 0.846 (± 0.01) |
| 0.80 | **0.865** (± **0.01**) | 0.845 (± 0.01) | 0.861 (± 0.01) | 0.837 (± 0.00) | 0.785 (± 0.05) | 0.857 (± 0.01) | 0.817 (± 0.02) | 0.817 (± 0.02) |
| 0.90 | 0.854 (± 0.01) | 0.833 (± 0.01) | **0.855** (± **0.00**) | 0.833 (± 0.00) | 0.465 (± 0.21) | 0.854 (± 0.01) | 0.819 (± 0.01) | 0.819 (± 0.01) |
| 0.99 | 0.822 (± 0.01) | 0.066 (± 0.00) | 0.810 (± 0.02) | 0.098 (± 0.01) | 0.230 (± 0.05) | **0.837** (± **0.02**) | 0.670 (± 0.02) | 0.670 (± 0.02) |

Table 11: F1 scores for CORA under mechanism *CD-MNAR* and varying $\mu$ (GSPNis not reported as it is not designed for categorical features).

| $\mu$ | GOODIE | FairAC | FP | GNNmi | GCNmf | PCFI | GNNzero | GNNmedian |
|---|---|---|---|---|---|---|---|---|
| 0.00 | 0.875 (± 0.00) | 0.863 (± 0.01) | **0.882** (± **0.00**) | 0.873 (± 0.00) | 0.875 (± 0.00) | 0.868 (± 0.00) | 0.863 (± 0.02) | 0.863 (± 0.02) |
| 0.10 | **0.875** (± **0.00**) | 0.864 (± 0.01) | 0.870 (± 0.01) | 0.862 (± 0.01) | 0.850 (± 0.00) | 0.869 (± 0.01) | 0.863 (± 0.02) | 0.863 (± 0.02) |
| 0.20 | **0.881** (± **0.01**) | 0.865 (± 0.01) | 0.874 (± 0.01) | 0.868 (± 0.01) | 0.856 (± 0.01) | 0.869 (± 0.01) | 0.860 (± 0.02) | 0.860 (± 0.02) |
| 0.30 | **0.882** (± **0.00**) | 0.858 (± 0.00) | 0.873 (± 0.00) | 0.871 (± 0.01) | 0.854 (± 0.00) | 0.866 (± 0.01) | 0.860 (± 0.02) | 0.860 (± 0.02) |
| 0.40 | **0.884** (± **0.01**) | 0.862 (± 0.01) | 0.870 (± 0.00) | 0.864 (± 0.00) | 0.853 (± 0.01) | 0.865 (± 0.01) | 0.853 (± 0.02) | 0.853 (± 0.02) |
| 0.50 | 0.867 (± 0.01) | 0.852 (± 0.01) | **0.867** (± **0.00**) | 0.861 (± 0.00) | 0.844 (± 0.02) | 0.861 (± 0.01) | 0.855 (± 0.02) | 0.855 (± 0.02) |
| 0.60 | **0.864** (± **0.01**) | 0.847 (± 0.00) | 0.860 (± 0.01) | 0.856 (± 0.01) | 0.849 (± 0.00) | 0.857 (± 0.00) | 0.842 (± 0.02) | 0.842 (± 0.02) |
| 0.70 | 0.860 (± 0.01) | 0.845 (± 0.01) | **0.864** (± **0.01**) | 0.852 (± 0.01) | 0.753 (± 0.12) | 0.856 (± 0.01) | 0.840 (± 0.02) | 0.840 (± 0.02) |
| 0.80 | 0.853 (± 0.01) | 0.844 (± 0.02) | **0.862** (± **0.01**) | 0.852 (± 0.01) | 0.551 (± 0.10) | 0.861 (± 0.01) | 0.822 (± 0.03) | 0.822 (± 0.03) |
| 0.90 | 0.848 (± 0.01) | 0.835 (± 0.01) | 0.852 (± 0.00) | 0.831 (± 0.01) | 0.271 (± 0.23) | **0.855** (± **0.01**) | 0.771 (± 0.03) | 0.771 (± 0.03) |
| 0.99 | 0.836 (± 0.01) | 0.810 (± 0.01) | 0.828 (± 0.01) | 0.788 (± 0.02) | 0.135 (± 0.05) | **0.849** (± **0.01**) | 0.727 (± 0.04) | 0.725 (± 0.03) |

Table 12: F1 scores for CITESEER under mechanism *U-MCAR* and varying $\mu$ (GSPNis not reported as it is not designed for categorical features).

| $\mu$ | GOODIE | FairAC | FP | GNNmi | GCNmf | PCFI | GNNzero | GNNmedian |
|---|---|---|---|---|---|---|---|---|
| 0.00 | 0.687 (± 0.00) | 0.700 (± 0.00) | 0.710 (± 0.02) | 0.704 (± 0.02) | 0.707 (± 0.00) | 0.706 (± 0.02) | **0.726** (± **0.02**) | 0.726 (± 0.02) |
| 0.10 | 0.682 (± 0.00) | 0.693 (± 0.00) | 0.707 (± 0.00) | 0.705 (± 0.00) | 0.692 (± 0.00) | 0.708 (± 0.00) | **0.732** (± **0.02**) | 0.732 (± 0.02) |
| 0.20 | 0.684 (± 0.00) | 0.693 (± 0.00) | 0.706 (± 0.00) | 0.695 (± 0.00) | 0.698 (± 0.00) | 0.705 (± 0.00) | **0.728** (± **0.02**) | 0.728 (± 0.02) |
| 0.30 | 0.691 (± 0.00) | 0.691 (± 0.00) | 0.705 (± 0.00) | 0.696 (± 0.00) | 0.697 (± 0.00) | 0.706 (± 0.00) | **0.723** (± **0.03**) | 0.723 (± 0.03) |
| 0.40 | 0.685 (± 0.00) | 0.700 (± 0.00) | 0.706 (± 0.00) | 0.698 (± 0.00) | 0.684 (± 0.00) | 0.708 (± 0.00) | **0.724** (± **0.02**) | 0.724 (± 0.02) |
| 0.50 | 0.669 (± 0.00) | 0.697 (± 0.00) | 0.702 (± 0.00) | 0.695 (± 0.00) | 0.675 (± 0.00) | 0.711 (± 0.00) | **0.722** (± **0.02**) | 0.722 (± 0.02) |
| 0.60 | 0.680 (± 0.00) | 0.695 (± 0.00) | 0.697 (± 0.00) | 0.699 (± 0.00) | 0.700 (± 0.00) | 0.707 (± 0.00) | **0.712** (± **0.02**) | 0.712 (± 0.02) |
| 0.70 | 0.699 (± 0.00) | 0.688 (± 0.00) | 0.694 (± 0.00) | 0.700 (± 0.00) | 0.507 (± 0.00) | 0.701 (± 0.00) | **0.710** (± **0.02**) | 0.710 (± 0.02) |
| 0.80 | 0.675 (± 0.00) | 0.687 (± 0.00) | 0.694 (± 0.00) | 0.696 (± 0.00) | 0.368 (± 0.00) | **0.707** (± **0.00**) | 0.701 (± 0.01) | 0.701 (± 0.01) |
| 0.90 | 0.684 (± 0.00) | 0.680 (± 0.00) | 0.686 (± 0.00) | 0.680 (± 0.00) | 0.215 (± 0.00) | **0.694** (± **0.00**) | 0.678 (± 0.02) | 0.678 (± 0.02) |
| 0.99 | 0.588 (± 0.00) | 0.584 (± 0.00) | 0.613 (± 0.00) | 0.539 (± 0.00) | 0.102 (± 0.00) | **0.636** (± **0.00**) | 0.519 (± 0.03) | 0.519 (± 0.03) |

Table 13: F1 scores for CITESEER under mechanism *S-MCAR* and varying $\mu$ (GSPNis not reported as it is not designed for categorical features).

| $\mu$ | GOODIE | FairAC | FP | GNNmi | GCNmf | PCFI | GNNzero | GNNmedian |
|---|---|---|---|---|---|---|---|---|
| 0.00 | 0.687 (± 0.00) | 0.700 (± 0.00) | 0.710 (± 0.02) | - | 0.707 (± 0.00) | 0.706 (± 0.02) | **0.726** (± **0.02**) | 0.726 (± 0.02) |
| 0.10 | 0.670 (± 0.00) | 0.688 (± 0.00) | 0.711 (± 0.00) | 0.703 (± 0.00) | 0.708 (± 0.00) | 0.708 (± 0.00) | **0.726** (± **0.03**) | 0.726 (± 0.03) |
| 0.20 | 0.675 (± 0.00) | 0.685 (± 0.00) | 0.707 (± 0.00) | 0.697 (± 0.00) | 0.707 (± 0.00) | 0.706 (± 0.00) | **0.725** (± **0.03**) | 0.725 (± 0.03) |
| 0.30 | 0.673 (± 0.00) | 0.681 (± 0.00) | 0.705 (± 0.00) | 0.692 (± 0.00) | 0.693 (± 0.00) | 0.701 (± 0.00) | **0.714** (± **0.02**) | 0.714 (± 0.02) |
| 0.40 | 0.677 (± 0.00) | 0.667 (± 0.00) | 0.698 (± 0.00) | 0.682 (± 0.00) | 0.682 (± 0.00) | 0.698 (± 0.00) | **0.704** (± **0.03**) | 0.704 (± 0.03) |
| 0.50 | 0.658 (± 0.00) | 0.659 (± 0.00) | 0.685 (± 0.00) | 0.680 (± 0.00) | 0.676 (± 0.00) | 0.683 (± 0.00) | **0.689** (± **0.03**) | 0.689 (± 0.03) |
| 0.60 | 0.667 (± 0.00) | 0.659 (± 0.00) | 0.676 (± 0.00) | 0.656 (± 0.00) | 0.659 (± 0.00) | **0.680** (± **0.00**) | 0.659 (± 0.02) | 0.659 (± 0.02) |
| 0.70 | 0.655 (± 0.00) | 0.646 (± 0.00) | 0.656 (± 0.00) | 0.629 (± 0.00) | 0.624 (± 0.00) | **0.662** (± **0.00**) | 0.617 (± 0.02) | 0.617 (± 0.02) |
| 0.80 | 0.621 (± 0.00) | 0.593 (± 0.00) | **0.629** (± **0.00**) | 0.575 (± 0.00) | 0.531 (± 0.00) | 0.628 (± 0.00) | 0.553 (± 0.03) | 0.553 (± 0.03) |
| 0.90 | 0.568 (± 0.00) | 0.508 (± 0.00) | 0.552 (± 0.00) | 0.449 (± 0.00) | 0.352 (± 0.00) | **0.584** (± **0.00**) | 0.455 (± 0.03) | 0.455 (± 0.03) |
| 0.99 | 0.425 (± 0.00) | 0.258 (± 0.00) | 0.381 (± 0.00) | 0.188 (± 0.00) | 0.159 (± 0.00) | **0.495** (± **0.00**) | 0.186 (± 0.01) | 0.186 (± 0.01) |

Table 14: F1 scores for CITESEER under mechanism *CD-MCAR* and varying $\mu$ (GSPNis not reported as it is not designed for categorical features).

| $\mu$ | GOODIE | FairAC | FP | GNNmi | GCNmf | PCFI | GNNzero | GNNmedian |
|---|---|---|---|---|---|---|---|---|
| 0.00 | 0.687 (± 0.00) | 0.700 (± 0.00) | 0.710 (± 0.02) | 0.704 (± 0.02) | 0.707 (± 0.00) | 0.706 (± 0.02) | **0.726** (± **0.02**) | 0.726 (± 0.02) |
| 0.10 | 0.671 (± 0.00) | 0.687 (± 0.00) | 0.698 (± 0.00) | 0.694 (± 0.00) | 0.693 (± 0.00) | 0.702 (± 0.00) | **0.723** (± **0.02**) | 0.723 (± 0.02) |
| 0.20 | 0.670 (± 0.00) | 0.686 (± 0.00) | 0.699 (± 0.00) | 0.691 (± 0.00) | 0.696 (± 0.00) | 0.698 (± 0.00) | **0.713** (± **0.02**) | 0.713 (± 0.02) |
| 0.30 | 0.666 (± 0.00) | 0.682 (± 0.00) | 0.697 (± 0.00) | 0.691 (± 0.00) | 0.694 (± 0.00) | 0.699 (± 0.00) | **0.711** (± **0.03**) | 0.711 (± 0.03) |
| 0.40 | 0.652 (± 0.00) | 0.683 (± 0.00) | 0.698 (± 0.00) | 0.691 (± 0.00) | 0.688 (± 0.00) | 0.701 (± 0.00) | **0.715** (± **0.02**) | 0.715 (± 0.02) |
| 0.50 | 0.650 (± 0.00) | 0.690 (± 0.00) | 0.699 (± 0.00) | 0.693 (± 0.00) | 0.688 (± 0.00) | **0.702** (± **0.00**) | 0.694 (± 0.02) | 0.694 (± 0.02) |
| 0.60 | 0.622 (± 0.00) | 0.686 (± 0.00) | 0.685 (± 0.00) | 0.685 (± 0.00) | 0.681 (± 0.00) | **0.704** (± **0.00**) | 0.684 (± 0.02) | 0.684 (± 0.02) |
| 0.70 | 0.613 (± 0.00) | 0.687 (± 0.00) | 0.686 (± 0.00) | 0.674 (± 0.00) | 0.677 (± 0.00) | **0.700** (± **0.00**) | 0.685 (± 0.03) | 0.685 (± 0.03) |
| 0.80 | 0.582 (± 0.00) | 0.671 (± 0.00) | 0.677 (± 0.00) | 0.664 (± 0.00) | 0.534 (± 0.00) | **0.686** (± **0.00**) | 0.674 (± 0.02) | 0.674 (± 0.02) |
| 0.90 | 0.456 (± 0.00) | **0.671** (± **0.00**) | 0.650 (± 0.00) | 0.650 (± 0.00) | 0.607 (± 0.00) | 0.648 (± 0.00) | 0.593 (± 0.02) | 0.593 (± 0.02) |
| 0.99 | 0.171 (± 0.00) | 0.257 (± 0.00) | 0.298 (± 0.00) | 0.346 (± 0.00) | 0.195 (± 0.00) | **0.348** (± **0.00**) | 0.184 (± 0.02) | 0.194 (± 0.03) |

Table 15: F1 scores for CITESEER under mechanism *FD-MNAR* and varying $\mu$ (GSPNis not reported as it is not designed for categorical features).

| $\mu$ | GOODIE | FairAC | FP | GNNmi | GCNmf | PCFI | GNNzero | GNNmedian |
|---|---|---|---|---|---|---|---|---|
| 0.00 | 0.687 (± 0.00) | 0.700 (± 0.00) | 0.710 (± 0.02) | 0.704 (± 0.02) | 0.707 (± 0.00) | 0.706 (± 0.02) | **0.728** (± **0.02**) | 0.728 (± 0.02) |
| 0.10 | 0.689 (± 0.03) | 0.691 (± 0.03) | 0.706 (± 0.02) | 0.699 (± 0.02) | 0.699 (± 0.02) | 0.708 (± 0.03) | **0.729** (± **0.02**) | 0.729 (± 0.02) |
| 0.20 | 0.686 (± 0.02) | 0.698 (± 0.03) | 0.703 (± 0.02) | 0.697 (± 0.02) | 0.696 (± 0.02) | 0.704 (± 0.02) | **0.720** (± **0.02**) | 0.720 (± 0.02) |
| 0.30 | 0.701 (± 0.04) | 0.690 (± 0.03) | 0.701 (± 0.03) | 0.693 (± 0.02) | 0.704 (± 0.02) | 0.700 (± 0.03) | **0.721** (± **0.03**) | 0.721 (± 0.03) |
| 0.40 | 0.696 (± 0.04) | 0.699 (± 0.04) | 0.695 (± 0.02) | 0.695 (± 0.02) | 0.692 (± 0.03) | 0.701 (± 0.03) | **0.717** (± **0.02**) | 0.717 (± 0.02) |
| 0.50 | 0.707 (± 0.03) | 0.688 (± 0.04) | 0.698 (± 0.03) | 0.693 (± 0.03) | 0.690 (± 0.02) | 0.702 (± 0.03) | **0.727** (± **0.02**) | 0.727 (± 0.02) |
| 0.60 | 0.708 (± 0.02) | 0.694 (± 0.03) | 0.691 (± 0.03) | 0.693 (± 0.03) | 0.696 (± 0.02) | 0.702 (± 0.03) | **0.712** (± **0.03**) | 0.712 (± 0.03) |
| 0.70 | 0.678 (± 0.04) | 0.688 (± 0.03) | 0.688 (± 0.02) | 0.686 (± 0.02) | 0.649 (± 0.03) | 0.690 (± 0.04) | **0.705** (± **0.02**) | 0.705 (± 0.02) |
| 0.80 | 0.695 (± 0.03) | 0.689 (± 0.04) | 0.689 (± 0.02) | 0.685 (± 0.02) | 0.437 (± 0.27) | 0.694 (± 0.03) | **0.696** (± **0.03**) | 0.696 (± 0.03) |
| 0.90 | 0.653 (± 0.03) | 0.681 (± 0.04) | 0.682 (± 0.02) | 0.687 (± 0.03) | 0.257 (± 0.17) | **0.689** (± **0.02**) | 0.676 (± 0.02) | 0.676 (± 0.02) |
| 0.99 | 0.601 (± 0.01) | 0.566 (± 0.01) | 0.611 (± 0.01) | 0.535 (± 0.02) | 0.118 (± 0.04) | **0.633** (± **0.01**) | 0.538 (± 0.03) | 0.538 (± 0.03) |

Table 16: F1 scores for CITESEER under mechanism *CD-MNAR* and varying $\mu$ (GSPNis not reported as it is not designed for categorical features).

| $\mu$ | GOODIE | FairAC | FP | GNNmi | GCNmf | PCFI | GNNzero | GNNmedian |
|---|---|---|---|---|---|---|---|---|
| 0.00 | 0.687 (± 0.00) | 0.700 (± 0.05) | 0.710 (± 0.02) | 0.704 (± 0.02) | 0.707 (± 0.00) | 0.706 (± 0.02) | **0.726** (± **0.02**) | 0.726 (± 0.02) |
| 0.10 | 0.692 (± 0.04) | 0.696 (± 0.04) | 0.708 (± 0.02) | 0.705 (± 0.02) | 0.702 (± 0.03) | 0.705 (± 0.02) | **0.729** (± **0.02**) | 0.729 (± 0.02) |
| 0.20 | 0.690 (± 0.04) | 0.689 (± 0.04) | 0.703 (± 0.03) | 0.702 (± 0.02) | 0.705 (± 0.02) | 0.704 (± 0.02) | **0.727** (± **0.02**) | 0.727 (± 0.02) |
| 0.30 | 0.700 (± 0.02) | 0.689 (± 0.04) | 0.708 (± 0.03) | 0.706 (± 0.02) | 0.708 (± 0.02) | 0.705 (± 0.02) | **0.728** (± **0.02**) | 0.728 (± 0.02) |
| 0.40 | 0.687 (± 0.04) | 0.695 (± 0.04) | 0.707 (± 0.03) | 0.704 (± 0.02) | 0.703 (± 0.03) | 0.704 (± 0.03) | **0.725** (± **0.02**) | 0.725 (± 0.02) |
| 0.50 | 0.675 (± 0.03) | 0.692 (± 0.03) | 0.699 (± 0.03) | 0.700 (± 0.03) | 0.697 (± 0.02) | 0.706 (± 0.03) | **0.718** (± **0.02**) | 0.718 (± 0.02) |
| 0.60 | 0.689 (± 0.03) | 0.689 (± 0.03) | 0.702 (± 0.03) | 0.699 (± 0.03) | 0.693 (± 0.03) | 0.706 (± 0.03) | **0.714** (± **0.02**) | 0.714 (± 0.02) |
| 0.70 | 0.681 (± 0.03) | 0.685 (± 0.03) | 0.692 (± 0.03) | 0.691 (± 0.03) | 0.522 (± 0.20) | 0.696 (± 0.03) | **0.702** (± **0.03**) | 0.702 (± 0.03) |
| 0.80 | 0.676 (± 0.05) | 0.685 (± 0.03) | 0.690 (± 0.03) | 0.689 (± 0.02) | 0.359 (± 0.15) | **0.696** (± **0.04**) | 0.689 (± 0.03) | 0.689 (± 0.03) |
| 0.90 | 0.665 (± 0.02) | **0.681** (± **0.03**) | 0.677 (± 0.03) | 0.666 (± 0.03) | 0.113 (± 0.06) | 0.681 (± 0.03) | 0.638 (± 0.02) | 0.638 (± 0.02) |
| 0.99 | 0.645 (± 0.03) | 0.631 (± 0.02) | 0.652 (± 0.02) | 0.621 (± 0.02) | 0.104 (± 0.06) | **0.660** (± **0.02**) | 0.593 (± 0.03) | 0.592 (± 0.03) |

Table 17: F1 scores for PUBMED under mechanism *U-MCAR* and varying $\mu$ (GSPNis not reported as it is not designed for categorical features).

| $\mu$ | GOODIE | FairAC | FP | GNNmi | GCNmf | PCFI | GNNzero | GNNmedian |
|---|---|---|---|---|---|---|---|---|
| 0.00 | 0.784 ($\pm$ 0.01) | 0.831 ($\pm$ 0.00) | **0.883** ($\pm$ **0.00**) | 0.881 ($\pm$ 0.00) | 0.877 ($\pm$ 0.00) | 0.882 ($\pm$ 0.00) | 0.875 ($\pm$ 0.00) | 0.875 ($\pm$ 0.00) |
| 0.10 | 0.787 ($\pm$ 0.00) | 0.830 ($\pm$ 0.00) | 0.877 ($\pm$ 0.00) | **0.879** ($\pm$ **0.00**) | 0.830 ($\pm$ 0.00) | 0.874 ($\pm$ 0.00) | 0.871 ($\pm$ 0.00) | 0.871 ($\pm$ 0.00) |
| 0.20 | 0.786 ($\pm$ 0.00) | 0.831 ($\pm$ 0.00) | 0.868 ($\pm$ 0.00) | **0.873** ($\pm$ **0.00**) | 0.832 ($\pm$ 0.00) | 0.868 ($\pm$ 0.00) | 0.866 ($\pm$ 0.00) | 0.866 ($\pm$ 0.00) |
| 0.30 | 0.785 ($\pm$ 0.00) | 0.830 ($\pm$ 0.00) | 0.870 ($\pm$ 0.00) | **0.872** ($\pm$ **0.00**) | 0.827 ($\pm$ 0.00) | 0.864 ($\pm$ 0.00) | 0.862 ($\pm$ 0.00) | 0.860 ($\pm$ 0.00) |
| 0.40 | 0.782 ($\pm$ 0.00) | 0.828 ($\pm$ 0.00) | 0.861 ($\pm$ 0.00) | **0.869** ($\pm$ **0.00**) | 0.828 ($\pm$ 0.00) | 0.858 ($\pm$ 0.00) | 0.857 ($\pm$ 0.01) | 0.857 ($\pm$ 0.00) |
| 0.50 | 0.784 ($\pm$ 0.00) | 0.827 ($\pm$ 0.00) | 0.856 ($\pm$ 0.00) | **0.862** ($\pm$ **0.00**) | 0.778 ($\pm$ 0.00) | 0.852 ($\pm$ 0.00) | 0.851 ($\pm$ 0.01) | 0.852 ($\pm$ 0.00) |
| 0.60 | 0.777 ($\pm$ 0.00) | 0.828 ($\pm$ 0.00) | 0.851 ($\pm$ 0.00) | **0.855** ($\pm$ **0.00**) | 0.805 ($\pm$ 0.00) | 0.849 ($\pm$ 0.00) | 0.846 ($\pm$ 0.00) | 0.845 ($\pm$ 0.00) |
| 0.70 | 0.772 ($\pm$ 0.00) | 0.824 ($\pm$ 0.00) | **0.847** ($\pm$ **0.00**) | 0.845 ($\pm$ 0.00) | 0.726 ($\pm$ 0.00) | 0.844 ($\pm$ 0.00) | 0.834 ($\pm$ 0.01) | 0.835 ($\pm$ 0.01) |
| 0.80 | 0.756 ($\pm$ 0.00) | 0.819 ($\pm$ 0.00) | 0.836 ($\pm$ 0.00) | 0.832 ($\pm$ 0.00) | 0.443 ($\pm$ 0.00) | **0.837** ($\pm$ **0.00**) | 0.820 ($\pm$ 0.00) | 0.816 ($\pm$ 0.00) |
| 0.90 | 0.700 ($\pm$ 0.00) | 0.806 ($\pm$ 0.00) | 0.822 ($\pm$ 0.00) | 0.803 ($\pm$ 0.00) | 0.315 ($\pm$ 0.00) | **0.832** ($\pm$ **0.00**) | 0.791 ($\pm$ 0.01) | 0.786 ($\pm$ 0.01) |
| 0.99 | 0.452 ($\pm$ 0.00) | 0.262 ($\pm$ 0.00) | 0.793 ($\pm$ 0.00) | 0.327 ($\pm$ 0.00) | 0.315 ($\pm$ 0.00) | **0.814** ($\pm$ **0.00**) | 0.674 ($\pm$ 0.02) | 0.693 ($\pm$ 0.01) |

Table 18: F1 scores for PUBMED under mechanism *S-MCAR* and varying $\mu$ (GSPNis not reported as it is not designed for categorical features).

| $\mu$ | GOODIE | FairAC | FP | GNNmi | GCNmf | PCFI | GNNzero | GNNmedian |
|---|---|---|---|---|---|---|---|---|
| 0.00 | 0.784 ($\pm$ 0.01) | 0.831 ($\pm$ 0.00) | **0.883** ($\pm$ **0.00**) | - | 0.877 ($\pm$ 0.00) | 0.882 ($\pm$ 0.00) | 0.875 ($\pm$ 0.00) | 0.875 ($\pm$ 0.00) |
| 0.10 | 0.786 ($\pm$ 0.00) | 0.831 ($\pm$ 0.00) | 0.875 ($\pm$ 0.00) | **0.875** ($\pm$ **0.00**) | 0.870 ($\pm$ 0.00) | 0.871 ($\pm$ 0.00) | 0.868 ($\pm$ 0.01) | 0.866 ($\pm$ 0.01) |
| 0.20 | 0.783 ($\pm$ 0.00) | 0.827 ($\pm$ 0.00) | 0.869 ($\pm$ 0.00) | **0.870** ($\pm$ **0.00**) | 0.861 ($\pm$ 0.00) | 0.867 ($\pm$ 0.00) | 0.860 ($\pm$ 0.01) | 0.859 ($\pm$ 0.01) |
| 0.30 | 0.785 ($\pm$ 0.00) | 0.832 ($\pm$ 0.00) | 0.863 ($\pm$ 0.00) | **0.865** ($\pm$ **0.00**) | 0.861 ($\pm$ 0.00) | 0.863 ($\pm$ 0.00) | 0.853 ($\pm$ 0.01) | 0.852 ($\pm$ 0.00) |
| 0.40 | 0.785 ($\pm$ 0.00) | 0.828 ($\pm$ 0.00) | 0.856 ($\pm$ 0.00) | **0.857** ($\pm$ **0.00**) | 0.848 ($\pm$ 0.00) | 0.856 ($\pm$ 0.00) | 0.846 ($\pm$ 0.00) | 0.847 ($\pm$ 0.01) |
| 0.50 | 0.775 ($\pm$ 0.00) | 0.827 ($\pm$ 0.00) | 0.853 ($\pm$ 0.00) | **0.854** ($\pm$ **0.00**) | 0.808 ($\pm$ 0.00) | 0.848 ($\pm$ 0.00) | 0.838 ($\pm$ 0.00) | 0.837 ($\pm$ 0.00) |
| 0.60 | 0.774 ($\pm$ 0.00) | 0.822 ($\pm$ 0.00) | 0.843 ($\pm$ 0.00) | **0.845** ($\pm$ **0.00**) | 0.798 ($\pm$ 0.00) | 0.843 ($\pm$ 0.00) | 0.829 ($\pm$ 0.00) | 0.827 ($\pm$ 0.00) |
| 0.70 | 0.760 ($\pm$ 0.00) | 0.813 ($\pm$ 0.00) | 0.832 ($\pm$ 0.00) | 0.827 ($\pm$ 0.00) | 0.762 ($\pm$ 0.00) | **0.836** ($\pm$ **0.00**) | 0.815 ($\pm$ 0.00) | 0.814 ($\pm$ 0.00) |
| 0.80 | 0.744 ($\pm$ 0.00) | 0.806 ($\pm$ 0.00) | 0.828 ($\pm$ 0.00) | 0.808 ($\pm$ 0.00) | 0.683 ($\pm$ 0.00) | **0.832** ($\pm$ **0.00**) | 0.785 ($\pm$ 0.01) | 0.788 ($\pm$ 0.01) |
| 0.90 | 0.706 ($\pm$ 0.00) | 0.786 ($\pm$ 0.00) | 0.815 ($\pm$ 0.00) | 0.743 ($\pm$ 0.00) | 0.421 ($\pm$ 0.00) | **0.825** ($\pm$ **0.00**) | 0.727 ($\pm$ 0.01) | 0.729 ($\pm$ 0.01) |
| 0.99 | 0.441 ($\pm$ 0.00) | 0.259 ($\pm$ 0.00) | 0.765 ($\pm$ 0.00) | 0.333 ($\pm$ 0.00) | 0.310 ($\pm$ 0.00) | **0.794** ($\pm$ **0.00**) | 0.446 ($\pm$ 0.03) | 0.458 ($\pm$ 0.02) |

Table 19: F1 scores for PUBMED under mechanism *CD-MCAR* and varying $\mu$ (GSPNis not reported as it is not designed for categorical features).

| $\mu$ | GOODIE | FairAC | FP | GNNmi | GCNmf | PCFI | GNNzero | GNNmedian |
|---|---|---|---|---|---|---|---|---|
| 0.00 | 0.784 ($\pm$ 0.01) | 0.831 ($\pm$ 0.00) | **0.883** ($\pm$ **0.00**) | 0.881 ($\pm$ 0.00) | 0.877 ($\pm$ 0.00) | 0.882 ($\pm$ 0.00) | 0.875 ($\pm$ 0.00) | 0.876 ($\pm$ 0.00) |
| 0.10 | 0.738 ($\pm$ 0.00) | 0.824 ($\pm$ 0.00) | 0.855 ($\pm$ 0.00) | **0.857** ($\pm$ **0.00**) | 0.830 ($\pm$ 0.00) | 0.852 ($\pm$ 0.00) | 0.848 ($\pm$ 0.00) | 0.846 ($\pm$ 0.00) |
| 0.20 | 0.700 ($\pm$ 0.00) | 0.820 ($\pm$ 0.00) | 0.845 ($\pm$ 0.00) | **0.851** ($\pm$ **0.00**) | 0.828 ($\pm$ 0.00) | 0.844 ($\pm$ 0.00) | 0.837 ($\pm$ 0.00) | 0.836 ($\pm$ 0.00) |
| 0.30 | 0.607 ($\pm$ 0.00) | 0.823 ($\pm$ 0.00) | 0.843 ($\pm$ 0.00) | **0.844** ($\pm$ **0.00**) | 0.823 ($\pm$ 0.00) | 0.836 ($\pm$ 0.00) | 0.822 ($\pm$ 0.00) | 0.822 ($\pm$ 0.00) |
| 0.40 | 0.534 ($\pm$ 0.00) | 0.821 ($\pm$ 0.00) | 0.834 ($\pm$ 0.00) | **0.842** ($\pm$ **0.00**) | 0.818 ($\pm$ 0.00) | 0.830 ($\pm$ 0.00) | 0.821 ($\pm$ 0.01) | 0.821 ($\pm$ 0.01) |
| 0.50 | 0.509 ($\pm$ 0.00) | 0.814 ($\pm$ 0.00) | 0.818 ($\pm$ 0.00) | **0.823** ($\pm$ **0.00**) | 0.797 ($\pm$ 0.00) | 0.820 ($\pm$ 0.00) | 0.808 ($\pm$ 0.01) | 0.806 ($\pm$ 0.01) |
| 0.60 | 0.422 ($\pm$ 0.00) | 0.812 ($\pm$ 0.00) | 0.808 ($\pm$ 0.00) | **0.816** ($\pm$ **0.00**) | 0.787 ($\pm$ 0.00) | 0.812 ($\pm$ 0.00) | 0.790 ($\pm$ 0.00) | 0.793 ($\pm$ 0.01) |
| 0.70 | 0.415 ($\pm$ 0.00) | 0.802 ($\pm$ 0.00) | 0.797 ($\pm$ 0.00) | **0.811** ($\pm$ **0.00**) | 0.779 ($\pm$ 0.00) | 0.801 ($\pm$ 0.00) | 0.778 ($\pm$ 0.01) | 0.774 ($\pm$ 0.01) |
| 0.80 | 0.396 ($\pm$ 0.00) | 0.779 ($\pm$ 0.00) | 0.749 ($\pm$ 0.00) | **0.783** ($\pm$ **0.00**) | 0.713 ($\pm$ 0.00) | 0.754 ($\pm$ 0.00) | 0.738 ($\pm$ 0.01) | 0.749 ($\pm$ 0.02) |
| 0.90 | 0.306 ($\pm$ 0.00) | 0.574 ($\pm$ 0.00) | 0.693 ($\pm$ 0.00) | **0.700** ($\pm$ **0.00**) | 0.391 ($\pm$ 0.00) | 0.683 ($\pm$ 0.00) | 0.664 ($\pm$ 0.01) | 0.667 ($\pm$ 0.02) |
| 0.99 | 0.198 ($\pm$ 0.00) | 0.266 ($\pm$ 0.00) | 0.303 ($\pm$ 0.00) | 0.330 ($\pm$ 0.00) | 0.306 ($\pm$ 0.00) | 0.305 ($\pm$ 0.00) | **0.346** ($\pm$ **0.02**) | 0.345 ($\pm$ 0.02) |

Table 20: F1 scores for PUBMED under mechanism *FD-MNAR* and varying $\mu$ (GSPNis not reported as it is not designed for categorical features).

| $\mu$ | GOODIE | FairAC | FP | GNNmi | GCNmf | PCFI | GNNzero | GNNmedian |
|---|---|---|---|---|---|---|---|---|
| 0.00 | 0.784 ($\pm$ 0.01) | 0.831 ($\pm$ 0.00) | **0.883** ($\pm$ **0.00**) | 0.881 ($\pm$ 0.00) | 0.877 ($\pm$ 0.00) | 0.882 ($\pm$ 0.00) | 0.875 ($\pm$ 0.00) | 0.874 ($\pm$ 0.00) |
| 0.10 | 0.785 ($\pm$ 0.02) | 0.832 ($\pm$ 0.00) | 0.876 ($\pm$ 0.01) | **0.880** ($\pm$ **0.01**) | 0.834 ($\pm$ 0.00) | 0.874 ($\pm$ 0.01) | 0.867 ($\pm$ 0.01) | 0.868 ($\pm$ 0.00) |
| 0.20 | 0.785 ($\pm$ 0.02) | 0.834 ($\pm$ 0.00) | 0.869 ($\pm$ 0.00) | **0.875** ($\pm$ **0.00**) | 0.832 ($\pm$ 0.00) | 0.869 ($\pm$ 0.01) | 0.864 ($\pm$ 0.01) | 0.864 ($\pm$ 0.00) |
| 0.30 | 0.785 ($\pm$ 0.02) | 0.830 ($\pm$ 0.00) | 0.865 ($\pm$ 0.00) | **0.870** ($\pm$ **0.00**) | 0.829 ($\pm$ 0.00) | 0.860 ($\pm$ 0.00) | 0.858 ($\pm$ 0.00) | 0.858 ($\pm$ 0.01) |
| 0.40 | 0.780 ($\pm$ 0.01) | 0.827 ($\pm$ 0.00) | 0.860 ($\pm$ 0.00) | **0.866** ($\pm$ **0.00**) | 0.733 ($\pm$ 0.11) | 0.856 ($\pm$ 0.00) | 0.853 ($\pm$ 0.01) | 0.854 ($\pm$ 0.00) |
| 0.50 | 0.775 ($\pm$ 0.02) | 0.822 ($\pm$ 0.00) | 0.853 ($\pm$ 0.00) | **0.859** ($\pm$ **0.00**) | 0.720 ($\pm$ 0.12) | 0.850 ($\pm$ 0.00) | 0.844 ($\pm$ 0.01) | 0.846 ($\pm$ 0.00) |
| 0.60 | 0.763 ($\pm$ 0.02) | 0.824 ($\pm$ 0.01) | 0.847 ($\pm$ 0.01) | **0.850** ($\pm$ **0.00**) | 0.746 ($\pm$ 0.04) | 0.842 ($\pm$ 0.00) | 0.836 ($\pm$ 0.01) | 0.836 ($\pm$ 0.00) |
| 0.70 | 0.745 ($\pm$ 0.03) | 0.813 ($\pm$ 0.00) | 0.836 ($\pm$ 0.00) | 0.834 ($\pm$ 0.00) | 0.579 ($\pm$ 0.25) | **0.837** ($\pm$ **0.00**) | 0.827 ($\pm$ 0.00) | 0.826 ($\pm$ 0.00) |
| 0.80 | 0.745 ($\pm$ 0.03) | 0.819 ($\pm$ 0.00) | 0.759 ($\pm$ 0.04) | **0.829** ($\pm$ **0.00**) | 0.555 ($\pm$ 0.14) | 0.764 ($\pm$ 0.00) | 0.805 ($\pm$ 0.01) | 0.805 ($\pm$ 0.01) |
| 0.90 | 0.336 ($\pm$ 0.01) | 0.806 ($\pm$ 0.00) | 0.693 ($\pm$ 0.01) | **0.812** ($\pm$ **0.00**) | 0.529 ($\pm$ 0.13) | 0.653 ($\pm$ 0.00) | 0.780 ($\pm$ 0.01) | 0.777 ($\pm$ 0.01) |
| 0.99 | 0.278 ($\pm$ 0.01) | 0.282 ($\pm$ 0.01) | 0.303 ($\pm$ 0.05) | 0.347 ($\pm$ 0.00) | 0.399 ($\pm$ 0.33) | 0.335 ($\pm$ 0.01) | 0.659 ($\pm$ 0.02) | **0.669** ($\pm$ **0.02**) |

Table 21: F1 scores for PUBMED under mechanism *CD-MNAR* and varying $\mu$ (GSPN is not reported as it is not designed for categorical features).

| $\mu$ | GOODIE | FairAC | FP | GNNmi | GCNmf | PCFI | GNNzero | GNNmedian |
|---|---|---|---|---|---|---|---|---|
| 0.00 | 0.784 (± 0.01) | 0.831 (± 0.00) | **0.883** (± **0.00**) | 0.881 (± 0.00) | 0.877 (± 0.00) | 0.882 (± 0.00) | 0.874 (± 0.00) | 0.875 (± 0.00) |
| 0.10 | 0.789 (± 0.02) | 0.829 (± 0.00) | 0.878 (± 0.00) | **0.880** (± **0.00**) | 0.835 (± 0.00) | 0.877 (± 0.00) | 0.866 (± 0.01) | 0.869 (± 0.00) |
| 0.20 | 0.783 (± 0.01) | 0.830 (± 0.00) | 0.870 (± 0.00) | **0.876** (± **0.00**) | 0.834 (± 0.00) | 0.867 (± 0.01) | 0.862 (± 0.00) | 0.861 (± 0.00) |
| 0.30 | 0.783 (± 0.02) | 0.828 (± 0.00) | 0.863 (± 0.00) | **0.871** (± **0.00**) | 0.823 (± 0.00) | 0.866 (± 0.00) | 0.860 (± 0.00) | 0.859 (± 0.00) |
| 0.40 | 0.777 (± 0.02) | 0.826 (± 0.00) | 0.858 (± 0.00) | **0.863** (± **0.00**) | 0.830 (± 0.00) | 0.857 (± 0.01) | 0.854 (± 0.00) | 0.852 (± 0.00) |
| 0.50 | 0.779 (± 0.01) | 0.825 (± 0.00) | 0.853 (± 0.00) | **0.858** (± **0.00**) | 0.826 (± 0.00) | 0.853 (± 0.00) | 0.847 (± 0.00) | 0.849 (± 0.00) |
| 0.60 | 0.769 (± 0.02) | 0.824 (± 0.00) | 0.847 (± 0.01) | **0.848** (± **0.01**) | 0.784 (± 0.04) | 0.848 (± 0.00) | 0.840 (± 0.01) | 0.840 (± 0.00) |
| 0.70 | 0.752 (± 0.03) | 0.816 (± 0.00) | 0.837 (± 0.00) | 0.835 (± 0.00) | 0.765 (± 0.02) | **0.837** (± **0.00**) | 0.827 (± 0.00) | 0.825 (± 0.00) |
| 0.80 | 0.742 (± 0.03) | 0.813 (± 0.00) | 0.828 (± 0.00) | 0.817 (± 0.00) | 0.323 (± 0.10) | **0.836** (± **0.00**) | 0.810 (± 0.01) | 0.809 (± 0.00) |
| 0.90 | 0.605 (± 0.13) | 0.628 (± 0.24) | 0.812 (± 0.00) | 0.770 (± 0.00) | 0.280 (± 0.05) | **0.823** (± **0.00**) | 0.760 (± 0.01) | 0.763 (± 0.01) |
| 0.99 | 0.557 (± 0.14) | 0.260 (± 0.00) | 0.800 (± 0.00) | 0.689 (± 0.01) | 0.418 (± 0.04) | **0.818** (± **0.00**) | 0.717 (± 0.01) | 0.728 (± 0.02) |

Table 22: F1 scores for SYNTHETIC under mechanism *U-MCAR* and varying $\mu$

| $\mu$ | GOODIE | GSPN | FairAC | FP | GNNmi | GCNmf | PCFI | GNNzero | GNNmedian | GNNmim |
|---|---|---|---|---|---|---|---|---|---|---|
| 0.00 | 0.812 (± 0.00) | 0.865 (± 0.00) | 0.815 (± 0.00) | 0.980 (± 0.00) | 0.982 (± 0.00) | 0.978 (± 0.00) | 0.977 (± 0.00) | 0.978 (± 0.01) | 0.978 (± 0.01) | **0.983** (± **0.01**) |
| 0.10 | 0.810 (± 0.00) | 0.822 (± 0.00) | 0.825 (± 0.00) | **0.910** (± **0.00**) | 0.902 (± 0.00) | 0.875 (± 0.00) | 0.898 (± 0.00) | 0.902 (± 0.02) | 0.903 (± 0.02) | 0.901 (± 0.00) |
| 0.20 | 0.792 (± 0.00) | 0.759 (± 0.00) | 0.808 (± 0.00) | 0.863 (± 0.00) | **0.870** (± **0.00**) | 0.790 (± 0.00) | 0.855 (± 0.00) | 0.853 (± 0.02) | 0.853 (± 0.02) | 0.861 (± 0.00) |
| 0.30 | 0.758 (± 0.00) | 0.768 (± 0.00) | 0.762 (± 0.00) | 0.795 (± 0.00) | 0.808 (± 0.00) | 0.770 (± 0.00) | 0.805 (± 0.00) | 0.800 (± 0.03) | 0.801 (± 0.03) | **0.815** (± **0.00**) |
| 0.40 | 0.758 (± 0.00) | 0.749 (± 0.00) | 0.759 (± 0.00) | 0.764 (± 0.00) | 0.771 (± 0.00) | 0.745 (± 0.00) | 0.763 (± 0.00) | 0.766 (± 0.00) | 0.766 (± 0.02) | **0.791** (± **0.00**) |
| 0.50 | 0.747 (± 0.00) | 0.721 (± 0.00) | 0.642 (± 0.00) | 0.745 (± 0.00) | 0.745 (± 0.00) | 0.710 (± 0.00) | **0.748** (± **0.00**) | 0.732 (± 0.04) | 0.730 (± 0.04) | 0.739 (± 0.00) |
| 0.60 | **0.773** (± **0.00**) | 0.708 (± 0.00) | 0.680 (± 0.00) | 0.720 (± 0.00) | 0.737 (± 0.00) | 0.692 (± 0.00) | 0.717 (± 0.00) | 0.714 (± 0.04) | 0.710 (± 0.04) | 0.714 (± 0.00) |
| 0.70 | **0.742** (± **0.00**) | 0.629 (± 0.00) | 0.611 (± 0.00) | 0.683 (± 0.00) | 0.689 (± 0.00) | 0.673 (± 0.00) | 0.678 (± 0.00) | 0.687 (± 0.03) | 0.693 (± 0.03) | 0.693 (± 0.00) |
| 0.80 | **0.771** (± **0.00**) | 0.579 (± 0.00) | 0.621 (± 0.00) | 0.632 (± 0.00) | 0.638 (± 0.00) | 0.601 (± 0.00) | 0.638 (± 0.00) | 0.610 (± 0.05) | 0.621 (± 0.05) | 0.649 (± 0.00) |
| 0.90 | **0.776** (± **0.00**) | 0.544 (± 0.00) | 0.567 (± 0.00) | 0.605 (± 0.00) | 0.602 (± 0.00) | 0.592 (± 0.00) | 0.588 (± 0.00) | 0.589 (± 0.04) | 0.599 (± 0.04) | 0.590 (± 0.00) |
| 0.99 | **0.762** (± **0.00**) | 0.499 (± 0.00) | 0.391 (± 0.00) | 0.542 (± 0.00) | 0.367 (± 0.00) | 0.471 (± 0.00) | 0.547 (± 0.00) | 0.548 (± 0.04) | 0.411 (± 0.07) | 0.535 (± 0.00) |

Table 23: F1 scores for SYNTHETIC under mechanism *S-MCAR* and varying $\mu$

| $\mu$ | GOODIE | GSPN | FairAC | FP | GNNmi | GCNmf | PCFI | GNNzero | GNNmedian | GNNmim |
|---|---|---|---|---|---|---|---|---|---|---|
| 0.00 | 0.812 (± 0.00) | 0.865 (± 0.00) | 0.815 (± 0.00) | 0.980 (± 0.00) | 0.982 (± 0.00) | 0.978 (± 0.00) | 0.977 (± 0.00) | 0.978 (± 0.01) | 0.978 (± 0.01) | **0.983** (± **0.01**) |
| 0.10 | 0.756 (± 0.00) | 0.748 (± 0.00) | 0.723 (± 0.00) | 0.903 (± 0.00) | **0.912** (± **0.00**) | 0.903 (± 0.00) | 0.900 (± 0.00) | 0.909 (± 0.01) | 0.911 (± 0.01) | 0.898 (± 0.00) |
| 0.20 | 0.769 (± 0.00) | 0.733 (± 0.00) | 0.727 (± 0.00) | **0.883** (± **0.00**) | 0.883 (± 0.00) | 0.872 (± 0.00) | 0.870 (± 0.00) | 0.844 (± 0.02) | 0.843 (± 0.02) | 0.875 (± 0.00) |
| 0.30 | 0.742 (± 0.00) | 0.737 (± 0.00) | 0.700 (± 0.00) | 0.830 (± 0.00) | **0.842** (± **0.00**) | 0.841 (± 0.00) | 0.831 (± 0.00) | 0.817 (± 0.02) | 0.813 (± 0.01) | 0.833 (± 0.00) |
| 0.40 | 0.716 (± 0.00) | 0.712 (± 0.00) | 0.683 (± 0.00) | **0.810** (± **0.00**) | 0.798 (± 0.00) | 0.752 (± 0.00) | 0.793 (± 0.00) | 0.775 (± 0.02) | 0.777 (± 0.02) | 0.799 (± 0.00) |
| 0.50 | 0.700 (± 0.00) | 0.711 (± 0.00) | 0.704 (± 0.00) | 0.785 (± 0.00) | **0.788** (± **0.00**) | 0.705 (± 0.00) | 0.780 (± 0.00) | 0.746 (± 0.02) | 0.748 (± 0.02) | 0.779 (± 0.00) |
| 0.60 | 0.658 (± 0.00) | 0.674 (± 0.00) | 0.695 (± 0.00) | 0.747 (± 0.00) | **0.761** (± **0.00**) | 0.726 (± 0.00) | 0.738 (± 0.00) | 0.718 (± 0.03) | 0.705 (± 0.04) | 0.756 (± 0.00) |
| 0.70 | 0.618 (± 0.00) | 0.675 (± 0.00) | 0.652 (± 0.00) | 0.687 (± 0.00) | 0.703 (± 0.00) | 0.665 (± 0.00) | 0.700 (± 0.00) | 0.663 (± 0.03) | 0.667 (± 0.02) | **0.727** (± **0.00**) |
| 0.80 | 0.584 (± 0.00) | 0.649 (± 0.00) | 0.616 (± 0.00) | 0.653 (± 0.00) | 0.667 (± 0.00) | 0.645 (± 0.00) | 0.638 (± 0.00) | 0.647 (± 0.05) | 0.656 (± 0.04) | **0.676** (± **0.00**) |
| 0.90 | 0.527 (± 0.00) | 0.588 (± 0.00) | 0.589 (± 0.00) | 0.597 (± 0.00) | 0.597 (± 0.00) | 0.578 (± 0.00) | 0.591 (± 0.00) | **0.601** (± **0.02**) | 0.593 (± 0.02) | 0.582 (± 0.00) |
| 0.99 | 0.337 (± 0.00) | 0.455 (± 0.00) | 0.338 (± 0.00) | **0.515** (± **0.00**) | 0.425 (± 0.00) | 0.403 (± 0.00) | 0.513 (± 0.00) | 0.488 (± 0.02) | 0.444 (± 0.05) | 0.477 (± 0.00) |

Table 24: F1 scores for SYNTHETIC under mechanism *CD-MCAR* and varying $\mu$

| $\mu$ | GOODIE | GSPN | FairAC | FP | GNNmi | GCNmf | PCFI | GNNzero | GNNmedian | GNNmim |
|---|---|---|---|---|---|---|---|---|---|---|
| 0.00 | 0.812 (± 0.00) | 0.865 (± 0.00) | 0.815 (± 0.00) | 0.980 (± 0.00) | **0.982** (± **0.00**) | 0.978 (± 0.00) | 0.977 (± 0.00) | 0.978 (± 0.01) | 0.978 (± 0.01) | 0.886 (± 0.00) |
| 0.10 | 0.778 (± 0.00) | 0.785 (± 0.00) | 0.792 (± 0.00) | 0.860 (± 0.00) | 0.857 (± 0.00) | 0.845 (± 0.00) | 0.860 (± 0.00) | **0.978** (± **0.01**) | 0.978 (± 0.01) | 0.829 (± 0.00) |
| 0.20 | 0.760 (± 0.00) | 0.731 (± 0.00) | 0.705 (± 0.00) | **0.788** (± **0.00**) | 0.770 (± 0.00) | 0.741 (± 0.00) | 0.772 (± 0.00) | 0.699 (± 0.02) | 0.699 (± 0.02) | 0.780 (± 0.00) |
| 0.30 | 0.730 (± 0.00) | 0.666 (± 0.00) | 0.718 (± 0.00) | 0.736 (± 0.00) | 0.733 (± 0.00) | 0.730 (± 0.00) | 0.734 (± 0.00) | 0.605 (± 0.03) | 0.605 (± 0.03) | **0.738** (± **0.00**) |
| 0.40 | **0.736** (± **0.00**) | 0.625 (± 0.00) | 0.607 (± 0.00) | 0.661 (± 0.00) | 0.659 (± 0.00) | 0.673 (± 0.00) | 0.649 (± 0.00) | 0.605 (± 0.03) | 0.605 (± 0.03) | 0.703 (± 0.00) |
| 0.50 | **0.761** (± **0.00**) | 0.547 (± 0.00) | 0.542 (± 0.00) | 0.619 (± 0.00) | 0.618 (± 0.00) | 0.628 (± 0.00) | 0.613 (± 0.00) | 0.605 (± 0.03) | 0.605 (± 0.03) | 0.682 (± 0.00) |
| 0.60 | **0.768** (± **0.00**) | 0.594 (± 0.00) | 0.543 (± 0.00) | 0.621 (± 0.00) | 0.613 (± 0.00) | 0.619 (± 0.00) | 0.605 (± 0.00) | 0.528 (± 0.03) | 0.528 (± 0.03) | 0.667 (± 0.00) |
| 0.70 | **0.759** (± **0.00**) | 0.603 (± 0.00) | 0.586 (± 0.00) | 0.617 (± 0.00) | 0.607 (± 0.00) | 0.591 (± 0.00) | 0.594 (± 0.00) | 0.536 (± 0.03) | 0.536 (± 0.03) | 0.675 (± 0.00) |
| 0.80 | **0.758** (± **0.00**) | 0.613 (± 0.00) | 0.486 (± 0.00) | 0.617 (± 0.00) | 0.622 (± 0.00) | 0.631 (± 0.00) | 0.620 (± 0.00) | 0.536 (± 0.03) | 0.536 (± 0.03) | 0.666 (± 0.00) |
| 0.90 | **0.775** (± **0.00**) | 0.544 (± 0.00) | 0.529 (± 0.00) | 0.623 (± 0.00) | 0.633 (± 0.00) | 0.623 (± 0.00) | 0.606 (± 0.00) | 0.535 (± 0.02) | 0.536 (± 0.03) | 0.678 (± 0.00) |
| 0.99 | **0.764** (± **0.00**) | 0.569 (± 0.00) | 0.557 (± 0.00) | 0.609 (± 0.00) | 0.611 (± 0.00) | 0.643 (± 0.00) | 0.612 (± 0.00) | 0.646 (± 0.03) | 0.638 (± 0.03) | 0.667 (± 0.00) |

Table 25: F1 scores for SYNTHETIC under mechanism *FD-MNAR* and varying $\mu$

| $\mu$ | GOODIE | GSPN | FairAC | FP | GNNmi | GCNmf | PCFI | GNNzero | GNNmedian | GNNmim |
|---|---|---|---|---|---|---|---|---|---|---|
| 0.00 | 0.812 (± 0.00) | 0.865 (± 0.00) | 0.815 (± 0.00) | 0.980 (± 0.00) | 0.982 (± 0.00) | 0.978 (± 0.00) | 0.977 (± 0.00) | 0.976 (± 0.01) | 0.976 (± 0.01) | **0.983** (± **0.01**) |
| 0.10 | 0.751 (± 0.05) | 0.750 (± 0.03) | 0.761 (± 0.02) | 0.893 (± 0.01) | **0.900** (± **0.02**) | 0.878 (± 0.02) | 0.895 (± 0.01) | 0.891 (± 0.02) | 0.894 (± 0.02) | 0.895 (± 0.01) |
| 0.20 | 0.750 (± 0.03) | 0.721 (± 0.01) | 0.699 (± 0.04) | 0.836 (± 0.02) | 0.845 (± 0.02) | 0.785 (± 0.04) | 0.847 (± 0.02) | 0.849 (± 0.03) | **0.854** (± **0.02**) | 0.843 (± 0.04) |
| 0.30 | 0.691 (± 0.04) | 0.678 (± 0.02) | 0.667 (± 0.03) | 0.810 (± 0.01) | 0.812 (± 0.01) | 0.771 (± 0.03) | 0.789 (± 0.01) | 0.819 (± 0.02) | **0.821** (± **0.01**) | 0.812 (± 0.01) |
| 0.40 | 0.693 (± 0.03) | 0.678 (± 0.03) | 0.682 (± 0.03) | 0.791 (± 0.02) | 0.798 (± 0.02) | 0.763 (± 0.02) | 0.791 (± 0.00) | 0.785 (± 0.02) | 0.793 (± 0.02) | **0.806** (± **0.01**) |
| 0.50 | 0.673 (± 0.02) | 0.668 (± 0.01) | 0.676 (± 0.03) | 0.753 (± 0.02) | 0.758 (± 0.02) | 0.713 (± 0.03) | 0.752 (± 0.01) | 0.741 (± 0.02) | 0.737 (± 0.02) | **0.763** (± **0.01**) |
| 0.60 | 0.620 (± 0.02) | 0.608 (± 0.02) | 0.610 (± 0.02) | 0.708 (± 0.01) | 0.715 (± 0.02) | 0.685 (± 0.02) | 0.702 (± 0.02) | 0.714 (± 0.01) | 0.719 (± 0.01) | **0.727** (± **0.01**) |
| 0.70 | 0.494 (± 0.07) | 0.580 (± 0.06) | 0.588 (± 0.02) | 0.651 (± 0.03) | 0.670 (± 0.04) | 0.631 (± 0.03) | 0.653 (± 0.04) | 0.676 (± 0.02) | 0.673 (± 0.03) | **0.688** (± **0.02**) |
| 0.80 | 0.425 (± 0.07) | 0.607 (± 0.04) | 0.577 (± 0.01) | 0.611 (± 0.01) | 0.627 (± 0.04) | 0.589 (± 0.02) | 0.596 (± 0.02) | 0.619 (± 0.01) | 0.624 (± 0.01) | **0.639** (± **0.02**) |
| 0.90 | 0.362 (± 0.02) | **0.625** (± **0.02**) | 0.512 (± 0.05) | 0.575 (± 0.02) | 0.595 (± 0.02) | 0.573 (± 0.02) | 0.582 (± 0.01) | 0.594 (± 0.04) | 0.601 (± 0.02) | 0.612 (± 0.02) |
| 0.99 | 0.429 (± 0.13) | 0.570 (± 0.02) | 0.423 (± 0.11) | 0.547 (± 0.02) | 0.536 (± 0.01) | 0.490 (± 0.05) | 0.551 (± 0.01) | 0.569 (± 0.03) | 0.545 (± 0.04) | **0.576** (± **0.02**) |

Table 26: F1 scores for SYNTHETIC under mechanism *CD-MNAR* and varying $\mu$

| $\mu$ | GOODIE | GSPN | FairAC | FP | GNNmi | GCNmf | PCFI | GNNzero | GNNmedian | GNNmim |
|---|---|---|---|---|---|---|---|---|---|---|
| 0.00 | 0.812 (± 0.00) | 0.865 (± 0.00) | 0.815 (± 0.00) | 0.980 (± 0.00) | 0.982 (± 0.00) | 0.978 (± 0.00) | 0.977 (± 0.00) | 0.978 (± 0.01) | 0.978 (± 0.01) | **0.983** (± 0.01) |
| 0.10 | 0.756 (± 0.04) | 0.757 (± 0.02) | 0.752 (± 0.02) | 0.913 (± 0.02) | **0.918** (± 0.02) | 0.882 (± 0.02) | 0.912 (± 0.01) | 0.912 (± 0.02) | 0.912 (± 0.02) | 0.913 (± 0.02) |
| 0.20 | 0.730 (± 0.05) | 0.718 (± 0.03) | 0.674 (± 0.05) | 0.856 (± 0.03) | **0.868** (± 0.03) | 0.800 (± 0.04) | 0.861 (± 0.04) | 0.864 (± 0.02) | 0.865 (± 0.03) | 0.865 (± 0.03) |
| 0.30 | 0.663 (± 0.05) | 0.716 (± 0.02) | 0.689 (± 0.03) | 0.803 (± 0.02) | 0.820 (± 0.02) | 0.768 (± 0.03) | 0.810 (± 0.03) | 0.807 (± 0.02) | 0.804 (± 0.02) | **0.830** (± 0.03) |
| 0.40 | 0.530 (± 0.16) | 0.678 (± 0.01) | 0.718 (± 0.03) | 0.744 (± 0.01) | 0.749 (± 0.01) | 0.753 (± 0.01) | 0.739 (± 0.03) | 0.756 (± 0.01) | 0.742 (± 0.01) | **0.776** (± 0.01) |
| 0.50 | 0.487 (± 0.12) | 0.662 (± 0.03) | 0.655 (± 0.04) | 0.697 (± 0.02) | 0.695 (± 0.03) | 0.683 (± 0.04) | 0.699 (± 0.04) | 0.689 (± 0.03) | 0.657 (± 0.02) | **0.725** (± 0.01) |
| 0.60 | 0.575 (± 0.06) | 0.696 (± 0.03) | 0.577 (± 0.03) | 0.683 (± 0.03) | 0.658 (± 0.03) | 0.666 (± 0.02) | 0.645 (± 0.03) | 0.694 (± 0.04) | 0.638 (± 0.03) | **0.731** (± 0.03) |
| 0.70 | 0.553 (± 0.03) | 0.616 (± 0.03) | 0.583 (± 0.02) | 0.613 (± 0.02) | 0.600 (± 0.04) | 0.617 (± 0.04) | 0.592 (± 0.05) | 0.642 (± 0.03) | 0.603 (± 0.04) | **0.668** (± 0.01) |
| 0.80 | 0.486 (± 0.06) | 0.638 (± 0.03) | 0.592 (± 0.03) | 0.588 (± 0.02) | 0.596 (± 0.03) | 0.570 (± 0.02) | 0.563 (± 0.03) | 0.618 (± 0.02) | 0.580 (± 0.04) | **0.655** (± 0.02) |
| 0.90 | 0.432 (± 0.08) | 0.618 (± 0.05) | 0.479 (± 0.10) | 0.586 (± 0.04) | 0.607 (± 0.03) | 0.556 (± 0.03) | 0.553 (± 0.01) | 0.598 (± 0.03) | 0.557 (± 0.04) | **0.635** (± 0.04) |
| 0.99 | 0.468 (± 0.03) | 0.545 (± 0.06) | 0.396 (± 0.08) | **0.594** (± 0.01) | 0.537 (± 0.01) | 0.475 (± 0.06) | 0.549 (± 0.03) | 0.550 (± 0.03) | 0.485 (± 0.06) | 0.568 (± 0.01) |

Table 27: F1 scores for AIR under mechanism *U-MCAR* and varying $\mu$

| $\mu$ | GOODIE | GSPN | FairAC | FP | GNNmi | GCNmf | PCFI | GNNzero | GNNmedian | GNNmim |
|---|---|---|---|---|---|---|---|---|---|---|
| 0.00 | 0.724 (± 0.00) | 0.798 (± 0.02) | 0.733 (± 0.00) | 0.918 (± 0.00) | 0.922 (± 0.01) | 0.922 (± 0.00) | 0.891 (± 0.00) | 0.916 (± 0.02) | 0.916 (± 0.02) | **0.930** (± 0.00) |
| 0.10 | 0.665 (± 0.00) | 0.710 (± 0.00) | 0.733 (± 0.00) | 0.895 (± 0.00) | 0.891 (± 0.00) | 0.768 (± 0.00) | 0.883 (± 0.00) | **0.904** (± 0.03) | 0.902 (± 0.03) | 0.899 (± 0.00) |
| 0.20 | 0.669 (± 0.00) | 0.582 (± 0.00) | 0.709 (± 0.00) | 0.848 (± 0.00) | 0.833 (± 0.00) | 0.747 (± 0.00) | 0.852 (± 0.00) | **0.874** (± 0.03) | 0.865 (± 0.03) | 0.859 (± 0.00) |
| 0.30 | 0.669 (± 0.00) | 0.502 (± 0.00) | 0.715 (± 0.00) | 0.836 (± 0.00) | 0.837 (± 0.00) | 0.712 (± 0.00) | 0.836 (± 0.00) | 0.837 (± 0.04) | **0.857** (± 0.03) | 0.852 (± 0.00) |
| 0.40 | 0.714 (± 0.00) | 0.532 (± 0.00) | 0.700 (± 0.00) | 0.805 (± 0.00) | 0.829 (± 0.00) | 0.712 (± 0.00) | 0.797 (± 0.00) | 0.813 (± 0.02) | **0.839** (± 0.02) | 0.833 (± 0.00) |
| 0.50 | 0.666 (± 0.00) | 0.553 (± 0.00) | 0.669 (± 0.00) | 0.801 (± 0.00) | 0.805 (± 0.00) | 0.711 (± 0.00) | 0.802 (± 0.00) | **0.832** (± 0.04) | 0.815 (± 0.03) | 0.767 (± 0.00) |
| 0.60 | 0.663 (± 0.00) | 0.452 (± 0.00) | 0.691 (± 0.00) | 0.775 (± 0.00) | 0.762 (± 0.00) | 0.701 (± 0.00) | 0.767 (± 0.00) | 0.795 (± 0.04) | **0.807** (± 0.06) | 0.744 (± 0.00) |
| 0.70 | 0.714 (± 0.00) | 0.495 (± 0.00) | 0.686 (± 0.00) | 0.724 (± 0.00) | 0.736 (± 0.00) | 0.656 (± 0.00) | **0.754** (± 0.00) | 0.753 (± 0.07) | 0.746 (± 0.05) | 0.736 (± 0.00) |
| 0.80 | 0.666 (± 0.00) | 0.559 (± 0.00) | 0.667 (± 0.00) | 0.712 (± 0.00) | 0.677 (± 0.00) | 0.647 (± 0.00) | 0.637 (± 0.00) | 0.709 (± 0.03) | **0.715** (± 0.03) | 0.713 (± 0.00) |
| 0.90 | 0.700 (± 0.00) | 0.541 (± 0.00) | 0.670 (± 0.00) | 0.585 (± 0.00) | 0.593 (± 0.00) | 0.669 (± 0.00) | 0.619 (± 0.00) | 0.598 (± 0.06) | 0.628 (± 0.04) | **0.705** (± 0.00) |
| 0.99 | **0.693** (± 0.00) | 0.409 (± 0.00) | 0.658 (± 0.00) | 0.436 (± 0.00) | 0.384 (± 0.00) | 0.651 (± 0.00) | 0.431 (± 0.00) | 0.440 (± 0.05) | 0.397 (± 0.04) | 0.664 (± 0.00) |

Table 28: F1 scores for AIR under mechanism *S-MCAR* and varying $\mu$

| $\mu$ | GOODIE | GSPN | FairAC | FP | GNNmi | GCNmf | PCFI | GNNzero | GNNmedian | GNNmim |
|---|---|---|---|---|---|---|---|---|---|---|
| 0.00 | 0.724 (± 0.00) | 0.798 (± 0.02) | 0.733 (± 0.00) | 0.918 (± 0.00) | 0.922 (± 0.01) | 0.922 (± 0.00) | 0.891 (± 0.00) | 0.916 (± 0.02) | 0.916 (± 0.02) | **0.930** (± 0.00) |
| 0.10 | 0.568 (± 0.00) | 0.644 (± 0.00) | 0.733 (± 0.00) | 0.891 (± 0.00) | 0.899 (± 0.00) | 0.895 (± 0.00) | 0.872 (± 0.00) | 0.879 (± 0.02) | **0.900** (± 0.02) | 0.891 (± 0.00) |
| 0.20 | 0.573 (± 0.00) | 0.597 (± 0.00) | 0.733 (± 0.00) | 0.860 (± 0.00) | 0.883 (± 0.00) | 0.851 (± 0.00) | **0.899** (± 0.00) | 0.860 (± 0.03) | 0.865 (± 0.03) | 0.890 (± 0.00) |
| 0.30 | 0.630 (± 0.00) | 0.527 (± 0.00) | 0.665 (± 0.00) | 0.850 (± 0.00) | 0.847 (± 0.00) | 0.820 (± 0.00) | 0.852 (± 0.00) | 0.838 (± 0.04) | **0.853** (± 0.03) | 0.835 (± 0.00) |
| 0.40 | 0.571 (± 0.00) | 0.508 (± 0.00) | 0.728 (± 0.00) | 0.819 (± 0.00) | 0.819 (± 0.00) | 0.795 (± 0.00) | 0.826 (± 0.00) | 0.812 (± 0.03) | 0.796 (± 0.04) | **0.842** (± 0.00) |
| 0.50 | 0.562 (± 0.00) | 0.530 (± 0.00) | 0.742 (± 0.00) | 0.787 (± 0.00) | 0.770 (± 0.00) | **0.829** (± 0.00) | 0.799 (± 0.00) | 0.769 (± 0.03) | 0.778 (± 0.03) | 0.817 (± 0.00) |
| 0.60 | 0.549 (± 0.00) | 0.532 (± 0.00) | 0.739 (± 0.00) | 0.750 (± 0.00) | 0.737 (± 0.00) | **0.809** (± 0.00) | 0.761 (± 0.00) | 0.736 (± 0.06) | 0.718 (± 0.04) | 0.797 (± 0.00) |
| 0.70 | 0.603 (± 0.00) | 0.532 (± 0.00) | 0.706 (± 0.00) | 0.686 (± 0.00) | 0.661 (± 0.00) | **0.767** (± 0.00) | 0.666 (± 0.00) | 0.709 (± 0.05) | 0.693 (± 0.03) | 0.756 (± 0.00) |
| 0.80 | 0.610 (± 0.00) | 0.476 (± 0.00) | 0.657 (± 0.00) | 0.607 (± 0.00) | 0.605 (± 0.00) | 0.721 (± 0.00) | 0.601 (± 0.00) | 0.614 (± 0.00) | 0.603 (± 0.04) | **0.734** (± 0.00) |
| 0.90 | 0.504 (± 0.00) | 0.389 (± 0.00) | 0.692 (± 0.00) | 0.549 (± 0.00) | 0.505 (± 0.00) | 0.677 (± 0.00) | 0.522 (± 0.00) | 0.537 (± 0.03) | 0.511 (± 0.02) | **0.699** (± 0.00) |
| 0.99 | 0.435 (± 0.00) | 0.332 (± 0.00) | **0.652** (± 0.00) | 0.350 (± 0.00) | 0.333 (± 0.00) | 0.643 (± 0.00) | 0.353 (± 0.00) | 0.351 (± 0.01) | 0.354 (± 0.01) | 0.652 (± 0.00) |

Table 29: F1 scores for AIR under mechanism *CD-MCAR* and varying $\mu$

| $\mu$ | GOODIE | GSPN | FairAC | FP | GNNmi | GCNmf | PCFI | GNNzero | GNNmedian | GNNmim |
|---|---|---|---|---|---|---|---|---|---|---|
| 0.00 | 0.724 (± 0.00) | 0.798 (± 0.02) | 0.733 (± 0.00) | 0.918 (± 0.00) | 0.922 (± 0.01) | 0.922 (± 0.00) | 0.891 (± 0.00) | 0.916 (± 0.02) | 0.916 (± 0.02) | **0.930** (± 0.00) |
| 0.10 | 0.714 (± 0.00) | 0.730 (± 0.00) | 0.706 (± 0.00) | 0.804 (± 0.00) | 0.819 (± 0.00) | 0.700 (± 0.00) | 0.820 (± 0.00) | 0.825 (± 0.05) | 0.825 (± 0.05) | **0.876** (± 0.00) |
| 0.20 | 0.714 (± 0.00) | 0.730 (± 0.00) | 0.703 (± 0.00) | 0.804 (± 0.00) | 0.819 (± 0.00) | 0.677 (± 0.00) | 0.820 (± 0.00) | 0.825 (± 0.05) | 0.825 (± 0.05) | **0.887** (± 0.00) |
| 0.30 | 0.710 (± 0.00) | 0.651 (± 0.00) | 0.613 (± 0.00) | 0.721 (± 0.00) | 0.697 (± 0.00) | 0.696 (± 0.00) | 0.726 (± 0.00) | 0.725 (± 0.07) | 0.725 (± 0.07) | **0.744** (± 0.00) |
| 0.40 | 0.701 (± 0.00) | 0.587 (± 0.00) | 0.617 (± 0.00) | 0.717 (± 0.00) | 0.687 (± 0.00) | 0.691 (± 0.00) | 0.701 (± 0.00) | 0.719 (± 0.00) | 0.719 (± 0.00) | **0.794** (± 0.00) |
| 0.50 | 0.717 (± 0.00) | 0.504 (± 0.00) | 0.458 (± 0.00) | 0.528 (± 0.00) | 0.571 (± 0.00) | 0.625 (± 0.00) | 0.564 (± 0.00) | 0.556 (± 0.08) | 0.556 (± 0.08) | **0.722** (± 0.00) |
| 0.60 | 0.717 (± 0.00) | 0.504 (± 0.00) | 0.450 (± 0.00) | 0.528 (± 0.00) | 0.571 (± 0.00) | 0.625 (± 0.00) | 0.564 (± 0.00) | 0.556 (± 0.08) | 0.556 (± 0.08) | **0.737** (± 0.00) |
| 0.70 | **0.717** (± 0.00) | 0.498 (± 0.00) | 0.446 (± 0.00) | 0.540 (± 0.00) | 0.553 (± 0.00) | 0.668 (± 0.00) | 0.518 (± 0.00) | 0.498 (± 0.04) | 0.498 (± 0.04) | 0.662 (± 0.00) |
| 0.80 | **0.703** (± 0.00) | 0.557 (± 0.00) | 0.430 (± 0.00) | 0.515 (± 0.00) | 0.481 (± 0.00) | 0.676 (± 0.00) | 0.457 (± 0.00) | 0.495 (± 0.05) | 0.495 (± 0.05) | 0.680 (± 0.00) |
| 0.90 | **0.703** (± 0.00) | 0.498 (± 0.00) | 0.338 (± 0.00) | 0.515 (± 0.00) | 0.481 (± 0.00) | 0.676 (± 0.00) | 0.457 (± 0.00) | 0.495 (± 0.05) | 0.495 (± 0.05) | 0.674 (± 0.00) |
| 0.99 | 0.660 (± 0.00) | 0.468 (± 0.00) | 0.338 (± 0.00) | 0.515 (± 0.00) | 0.481 (± 0.00) | 0.682 (± 0.00) | 0.457 (± 0.00) | 0.675 (± 0.05) | **0.688** (± 0.05) | 0.673 (± 0.00) |

Table 30: F1 scores for AIR under mechanism *FD-MNAR* and varying $\mu$

| $\mu$ | GOODIE | GSPN | FairAC | FP | GNNmi | GCNmf | PCFI | GNNzero | GNNmedian | GNNmim |
|---|---|---|---|---|---|---|---|---|---|---|
| 0.00 | 0.724 (± 0.00) | 0.798 (± 0.02) | 0.733 (± 0.00) | 0.918 (± 0.00) | 0.922 (± 0.01) | 0.922 (± 0.00) | 0.891 (± 0.00) | 0.911 (± 0.03) | 0.914 (± 0.02) | **0.930** (± 0.00) |
| 0.10 | 0.618 (± 0.10) | 0.758 (± 0.05) | 0.709 (± 0.03) | 0.895 (± 0.01) | 0.891 (± 0.04) | 0.772 (± 0.02) | 0.883 (± 0.03) | 0.890 (± 0.03) | 0.897 (± 0.03) | **0.906** (± 0.02) |
| 0.20 | 0.595 (± 0.10) | 0.776 (± 0.05) | 0.668 (± 0.08) | 0.883 (± 0.03) | 0.879 (± 0.01) | 0.756 (± 0.03) | 0.867 (± 0.02) | 0.852 (± 0.02) | **0.888** (± 0.02) | 0.887 (± 0.01) |
| 0.30 | 0.580 (± 0.12) | 0.536 (± 0.15) | 0.721 (± 0.01) | 0.852 (± 0.03) | 0.859 (± 0.01) | 0.745 (± 0.03) | 0.833 (± 0.02) | 0.845 (± 0.02) | 0.864 (± 0.03) | **0.875** (± 0.01) |
| 0.40 | 0.677 (± 0.03) | 0.575 (± 0.09) | 0.716 (± 0.02) | 0.852 (± 0.02) | **0.855** (± 0.03) | 0.725 (± 0.02) | 0.840 (± 0.04) | 0.839 (± 0.02) | 0.848 (± 0.04) | 0.852 (± 0.02) |
| 0.50 | 0.587 (± 0.13) | 0.620 (± 0.08) | 0.719 (± 0.02) | 0.837 (± 0.00) | 0.832 (± 0.03) | 0.698 (± 0.04) | 0.829 (± 0.04) | 0.806 (± 0.01) | 0.822 (± 0.03) | **0.852** (± 0.03) |
| 0.60 | 0.556 (± 0.16) | 0.686 (± 0.05) | 0.692 (± 0.02) | **0.837** (± 0.02) | 0.808 (± 0.06) | 0.711 (± 0.03) | 0.793 (± 0.02) | 0.780 (± 0.04) | 0.783 (± 0.05) | 0.817 (± 0.03) |
| 0.70 | 0.556 (± 0.16) | 0.634 (± 0.02) | 0.717 (± 0.02) | 0.769 (± 0.03) | **0.779** (± 0.05) | 0.685 (± 0.01) | 0.750 (± 0.04) | 0.745 (± 0.05) | 0.771 (± 0.03) | 0.770 (± 0.03) |
| 0.80 | 0.556 (± 0.16) | 0.665 (± 0.02) | 0.665 (± 0.03) | 0.654 (± 0.05) | 0.709 (± 0.03) | 0.667 (± 0.03) | 0.660 (± 0.08) | 0.718 (± 0.07) | 0.719 (± 0.04) | **0.786** (± 0.02) |
| 0.90 | 0.582 (± 0.09) | 0.645 (± 0.04) | 0.662 (± 0.01) | 0.658 (± 0.05) | 0.661 (± 0.02) | 0.659 (± 0.03) | 0.530 (± 0.05) | 0.670 (± 0.06) | 0.655 (± 0.05) | **0.710** (± 0.05) |
| 0.99 | 0.638 (± 0.05) | 0.635 (± 0.02) | 0.637 (± 0.04) | 0.557 (± 0.04) | 0.528 (± 0.03) | **0.674** (± 0.02) | 0.508 (± 0.07) | 0.549 (± 0.06) | 0.565 (± 0.04) | 0.616 (± 0.05) |

Table 31: F1 scores for AIR under mechanism *CD-MNAR* and varying $\mu$

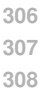

| $\mu$ | GOODIE | GSPN | FairAC | FP | GNNmi | GCNmf | PCFI | GNNzero | GNNmedian | GNNmim |
|---|---|---|---|---|---|---|---|---|---|---|
| 0.00 | 0.724 (± 0.00) | 0.798 (± 0.02) | 0.733 (± 0.00) | 0.918 (± 0.00) | 0.922 (± 0.01) | 0.922 (± 0.00) | 0.891 (± 0.00) | 0.916 (± 0.02) | 0.916 (± 0.02) | **0.930** (± 0.00) |
| 0.10 | 0.598 (± 0.11) | 0.667 (± 0.02) | 0.722 (± 0.02) | 0.888 (± 0.05) | 0.887 (± 0.04) | 0.851 (± 0.01) | 0.891 (± 0.03) | 0.860 (± 0.04) | 0.883 (± 0.04) | **0.895** (± 0.04) |
| 0.20 | 0.556 (± 0.16) | 0.632 (± 0.19) | 0.697 (± 0.02) | **0.864** (± 0.02) | 0.848 (± 0.06) | 0.778 (± 0.04) | 0.841 (± 0.01) | 0.853 (± 0.04) | 0.836 (± 0.04) | 0.864 (± 0.01) |
| 0.30 | 0.556 (± 0.16) | 0.526 (± 0.13) | 0.722 (± 0.03) | 0.845 (± 0.01) | 0.825 (± 0.04) | 0.689 (± 0.02) | 0.841 (± 0.03) | 0.855 (± 0.02) | 0.806 (± 0.04) | **0.891** (± 0.04) |
| 0.40 | 0.480 (± 0.16) | 0.691 (± 0.14) | 0.601 (± 0.12) | 0.833 (± 0.02) | 0.805 (± 0.03) | 0.722 (± 0.02) | 0.860 (± 0.03) | 0.856 (± 0.02) | 0.811 (± 0.02) | **0.860** (± 0.03) |
| 0.50 | 0.536 (± 0.16) | 0.607 (± 0.09) | 0.705 (± 0.02) | 0.813 (± 0.02) | 0.769 (± 0.04) | 0.674 (± 0.01) | 0.783 (± 0.04) | 0.790 (± 0.05) | 0.777 (± 0.05) | **0.833** (± 0.03) |
| 0.60 | 0.622 (± 0.06) | 0.636 (± 0.04) | 0.694 (± 0.01) | 0.758 (± 0.05) | 0.708 (± 0.07) | 0.681 (± 0.01) | 0.766 (± 0.06) | **0.814** (± 0.03) | 0.774 (± 0.07) | 0.766 (± 0.06) |
| 0.70 | 0.580 (± 0.10) | 0.672 (± 0.07) | 0.681 (± 0.01) | **0.757** (± 0.03) | 0.724 (± 0.04) | 0.644 (± 0.02) | 0.753 (± 0.05) | 0.755 (± 0.06) | 0.720 (± 0.02) | 0.726 (± 0.05) |
| 0.80 | 0.563 (± 0.12) | 0.681 (± 0.05) | 0.676 (± 0.01) | 0.733 (± 0.02) | 0.655 (± 0.02) | 0.658 (± 0.02) | 0.712 (± 0.01) | 0.735 (± 0.02) | 0.686 (± 0.06) | **0.769** (± 0.03) |
| 0.90 | 0.655 (± 0.03) | 0.615 (± 0.04) | 0.653 (± 0.01) | **0.693** (± 0.04) | 0.579 (± 0.04) | 0.643 (± 0.04) | 0.692 (± 0.06) | 0.678 (± 0.03) | 0.613 (± 0.04) | 0.668 (± 0.02) |
| 0.99 | 0.654 (± 0.03) | 0.522 (± 0.04) | **0.660** (± 0.05) | 0.524 (± 0.07) | 0.473 (± 0.05) | 0.650 (± 0.06) | 0.424 (± 0.06) | 0.523 (± 0.06) | 0.411 (± 0.03) | 0.631 (± 0.07) |

Table 32: F1 scores for ELECTRIC under mechanism *U-MCAR* and varying $\mu$

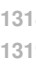

| $\mu$ | GOODIE | GSPN | FairAC | FP | GNNmi | GCNmf | PCFI | GNNzero | GNNmedian | GNNmim |
|---|---|---|---|---|---|---|---|---|---|---|
| 0.00 | 0.588 (± 0.00) | 0.915 (± 0.00) | **0.963** (± 0.01) | 0.885 (± 0.00) | 0.929 (± 0.00) | 0.861 (± 0.00) | 0.903 (± 0.00) | 0.912 (± 0.01) | 0.909 (± 0.01) | 0.938 (± 0.01) |
| 0.10 | 0.589 (± 0.00) | 0.827 (± 0.00) | **0.931** (± 0.00) | 0.865 (± 0.00) | 0.864 (± 0.00) | 0.887 (± 0.00) | 0.889 (± 0.00) | 0.855 (± 0.03) | 0.854 (± 0.02) | 0.923 (± 0.00) |
| 0.20 | 0.589 (± 0.00) | 0.806 (± 0.00) | **0.935** (± 0.00) | 0.821 (± 0.00) | 0.807 (± 0.00) | 0.876 (± 0.00) | 0.877 (± 0.00) | 0.805 (± 0.03) | 0.807 (± 0.03) | 0.877 (± 0.00) |
| 0.30 | 0.588 (± 0.00) | 0.770 (± 0.00) | **0.924** (± 0.00) | 0.758 (± 0.00) | 0.780 (± 0.00) | 0.889 (± 0.00) | 0.872 (± 0.00) | 0.742 (± 0.03) | 0.781 (± 0.04) | 0.868 (± 0.00) |
| 0.40 | 0.590 (± 0.00) | 0.703 (± 0.00) | **0.906** (± 0.00) | 0.711 (± 0.00) | 0.728 (± 0.00) | 0.874 (± 0.00) | 0.865 (± 0.00) | 0.710 (± 0.02) | 0.746 (± 0.04) | 0.859 (± 0.00) |
| 0.50 | 0.587 (± 0.00) | 0.626 (± 0.00) | **0.922** (± 0.00) | 0.676 (± 0.00) | 0.693 (± 0.00) | 0.864 (± 0.00) | 0.841 (± 0.00) | 0.676 (± 0.03) | 0.721 (± 0.04) | 0.804 (± 0.00) |
| 0.60 | 0.584 (± 0.00) | 0.567 (± 0.00) | **0.881** (± 0.00) | 0.598 (± 0.00) | 0.614 (± 0.00) | 0.877 (± 0.00) | 0.793 (± 0.00) | 0.597 (± 0.04) | 0.663 (± 0.06) | 0.779 (± 0.00) |
| 0.70 | 0.582 (± 0.00) | 0.506 (± 0.00) | **0.868** (± 0.00) | 0.548 (± 0.00) | 0.553 (± 0.00) | 0.831 (± 0.00) | 0.771 (± 0.00) | 0.528 (± 0.02) | 0.601 (± 0.06) | 0.766 (± 0.00) |
| 0.80 | 0.592 (± 0.00) | 0.397 (± 0.00) | **0.852** (± 0.00) | 0.496 (± 0.00) | 0.522 (± 0.00) | 0.807 (± 0.00) | 0.730 (± 0.00) | 0.465 (± 0.03) | 0.509 (± 0.06) | 0.728 (± 0.00) |
| 0.90 | 0.593 (± 0.00) | 0.389 (± 0.00) | **0.744** (± 0.00) | 0.361 (± 0.00) | 0.423 (± 0.00) | 0.701 (± 0.00) | 0.628 (± 0.00) | 0.407 (± 0.04) | 0.395 (± 0.02) | 0.646 (± 0.00) |
| 0.99 | 0.592 (± 0.00) | 0.289 (± 0.00) | 0.260 (± 0.00) | 0.285 (± 0.00) | 0.282 (± 0.00) | **0.630** (± 0.00) | 0.333 (± 0.00) | 0.278 (± 0.01) | 0.276 (± 0.01) | 0.412 (± 0.00) |

Table 33: F1 scores for ELECTRIC under mechanism *S-MCAR* and varying $\mu$

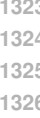

| $\mu$ | GOODIE | GSPN | FairAC | FP | GNNmi | GCNmf | PCFI | GNNzero | GNNmedian | GNNmim |
|---|---|---|---|---|---|---|---|---|---|---|
| 0.00 | 0.588 (± 0.00) | 0.915 (± 0.00) | **0.963** (± 0.01) | 0.885 (± 0.00) | 0.929 (± 0.00) | 0.861 (± 0.00) | 0.903 (± 0.00) | 0.909 (± 0.01) | 0.912 (± 0.01) | 0.938 (± 0.01) |
| 0.10 | 0.493 (± 0.00) | 0.891 (± 0.00) | **0.959** (± 0.00) | 0.831 (± 0.00) | 0.853 (± 0.00) | 0.862 (± 0.00) | 0.854 (± 0.00) | 0.872 (± 0.01) | 0.873 (± 0.02) | 0.904 (± 0.00) |
| 0.20 | 0.484 (± 0.00) | 0.855 (± 0.00) | **0.945** (± 0.00) | 0.821 (± 0.00) | 0.851 (± 0.00) | 0.867 (± 0.00) | 0.870 (± 0.00) | 0.833 (± 0.01) | 0.842 (± 0.03) | 0.878 (± 0.00) |
| 0.30 | 0.478 (± 0.00) | 0.816 (± 0.00) | **0.935** (± 0.00) | 0.768 (± 0.00) | 0.796 (± 0.00) | 0.872 (± 0.00) | 0.856 (± 0.00) | 0.776 (± 0.02) | 0.805 (± 0.02) | 0.855 (± 0.00) |
| 0.40 | 0.483 (± 0.00) | 0.756 (± 0.00) | **0.940** (± 0.00) | 0.703 (± 0.00) | 0.734 (± 0.00) | 0.842 (± 0.00) | 0.871 (± 0.00) | 0.736 (± 0.03) | 0.754 (± 0.01) | 0.801 (± 0.00) |
| 0.50 | 0.431 (± 0.00) | 0.708 (± 0.00) | **0.926** (± 0.00) | 0.656 (± 0.00) | 0.665 (± 0.00) | 0.839 (± 0.00) | 0.844 (± 0.00) | 0.682 (± 0.02) | 0.712 (± 0.01) | 0.810 (± 0.00) |
| 0.60 | 0.397 (± 0.00) | 0.632 (± 0.00) | **0.898** (± 0.00) | 0.619 (± 0.00) | 0.617 (± 0.00) | 0.813 (± 0.00) | 0.808 (± 0.00) | 0.627 (± 0.03) | 0.651 (± 0.01) | 0.787 (± 0.00) |
| 0.70 | 0.435 (± 0.00) | 0.563 (± 0.00) | **0.870** (± 0.00) | 0.528 (± 0.00) | 0.545 (± 0.00) | 0.799 (± 0.00) | 0.776 (± 0.00) | 0.543 (± 0.04) | 0.586 (± 0.05) | 0.711 (± 0.00) |
| 0.80 | 0.490 (± 0.00) | 0.522 (± 0.00) | **0.806** (± 0.00) | 0.475 (± 0.00) | 0.455 (± 0.00) | 0.764 (± 0.00) | 0.770 (± 0.00) | 0.477 (± 0.03) | 0.493 (± 0.00) | 0.676 (± 0.00) |
| 0.90 | 0.374 (± 0.00) | 0.392 (± 0.00) | **0.771** (± 0.00) | 0.420 (± 0.00) | 0.394 (± 0.00) | 0.738 (± 0.00) | 0.496 (± 0.00) | 0.374 (± 0.03) | 0.381 (± 0.03) | 0.567 (± 0.00) |
| 0.99 | 0.260 (± 0.00) | 0.265 (± 0.00) | 0.260 (± 0.00) | 0.269 (± 0.00) | 0.277 (± 0.00) | **0.639** (± 0.00) | 0.285 (± 0.00) | 0.267 (± 0.01) | 0.267 (± 0.01) | 0.479 (± 0.00) |

Table 34: F1 scores for ELECTRIC under mechanism *CD-MCAR* and varying $\mu$



| $\mu$ | GOODIE | GSPN | FairAC | FP | GNNmi | GCNmf | PCFI | GNNzero | GNNmedian | GNNmim |
|---|---|---|---|---|---|---|---|---|---|---|
| 0.00 | 0.588 (± 0.00) | 0.915 (± 0.00) | **0.963** (± 0.01) | 0.885 (± 0.00) | 0.929 (± 0.00) | 0.861 (± 0.00) | 0.903 (± 0.00) | 0.908 (± 0.01) | 0.911 (± 0.01) | 0.920 (± 0.00) |
| 0.10 | 0.585 (± 0.00) | 0.794 (± 0.00) | **0.910** (± 0.00) | 0.828 (± 0.00) | 0.843 (± 0.00) | 0.890 (± 0.00) | 0.894 (± 0.00) | 0.804 (± 0.03) | 0.804 (± 0.03) | 0.867 (± 0.00) |
| 0.20 | 0.584 (± 0.00) | 0.687 (± 0.00) | **0.920** (± 0.00) | 0.710 (± 0.00) | 0.762 (± 0.00) | 0.860 (± 0.00) | 0.842 (± 0.00) | 0.804 (± 0.03) | 0.805 (± 0.03) | 0.815 (± 0.00) |
| 0.30 | 0.591 (± 0.00) | 0.604 (± 0.00) | 0.650 (± 0.00) | 0.672 (± 0.00) | 0.693 (± 0.00) | 0.815 (± 0.00) | **0.820** (± 0.00) | 0.635 (± 0.01) | 0.635 (± 0.01) | 0.793 (± 0.00) |
| 0.40 | 0.587 (± 0.00) | 0.475 (± 0.00) | 0.630 (± 0.00) | 0.475 (± 0.00) | 0.494 (± 0.00) | **0.729** (± 0.00) | 0.723 (± 0.00) | 0.263 (± 0.01) | 0.263 (± 0.01) | 0.685 (± 0.00) |
| 0.50 | 0.589 (± 0.00) | 0.301 (± 0.00) | **0.630** (± 0.00) | 0.260 (± 0.00) | 0.260 (± 0.00) | 0.630 (± 0.00) | 0.260 (± 0.00) | 0.265 (± 0.01) | 0.265 (± 0.01) | 0.532 (± 0.00) |
| 0.60 | 0.593 (± 0.00) | 0.271 (± 0.00) | **0.630** (± 0.00) | 0.260 (± 0.00) | 0.260 (± 0.00) | 0.630 (± 0.00) | 0.260 (± 0.00) | 0.265 (± 0.01) | 0.265 (± 0.01) | 0.517 (± 0.00) |
| 0.70 | 0.589 (± 0.00) | 0.310 (± 0.00) | 0.260 (± 0.00) | 0.260 (± 0.00) | 0.260 (± 0.00) | **0.629** (± 0.00) | 0.260 (± 0.00) | 0.267 (± 0.01) | 0.267 (± 0.01) | 0.571 (± 0.00) |
| 0.80 | 0.593 (± 0.00) | 0.343 (± 0.00) | 0.260 (± 0.00) | 0.260 (± 0.00) | 0.263 (± 0.00) | **0.630** (± 0.00) | 0.260 (± 0.00) | 0.260 (± 0.00) | 0.260 (± 0.00) | 0.544 (± 0.00) |
| 0.90 | 0.589 (± 0.00) | 0.315 (± 0.00) | 0.260 (± 0.00) | 0.260 (± 0.00) | 0.263 (± 0.00) | **0.630** (± 0.00) | 0.260 (± 0.00) | 0.260 (± 0.00) | 0.260 (± 0.00) | 0.538 (± 0.00) |
| 0.99 | 0.589 (± 0.00) | 0.330 (± 0.00) | 0.260 (± 0.00) | 0.260 (± 0.00) | 0.260 (± 0.00) | **0.630** (± 0.00) | 0.260 (± 0.00) | 0.382 (± 0.01) | 0.423 (± 0.02) | 0.552 (± 0.00) |

Table 35: F1 scores for ELECTRIC under mechanism *FD-MNAR* and varying $\mu$

| $\mu$ | GOODIE | GSPN | FairAC | FP | GNNmi | GCNmf | PCFI | GNNzero | GNNmedian | GNNmim |
|---|---|---|---|---|---|---|---|---|---|---|
| 0.00 | 0.588 (± 0.00) | 0.915 (± 0.00) | **0.963** (± 0.01) | 0.885 (± 0.00) | 0.929 (± 0.00) | 0.861 (± 0.00) | 0.903 (± 0.00) | 0.911 (± 0.01) | 0.913 (± 0.01) | 0.938 (± 0.01) |
| 0.10 | 0.468 (± 0.15) | 0.879 (± 0.01) | **0.944** (± 0.02) | 0.862 (± 0.03) | 0.844 (± 0.02) | 0.878 (± 0.03) | 0.870 (± 0.04) | 0.840 (± 0.03) | 0.851 (± 0.02) | 0.916 (± 0.01) |
| 0.20 | 0.491 (± 0.13) | 0.850 (± 0.01) | **0.938** (± 0.01) | 0.808 (± 0.02) | 0.813 (± 0.02) | 0.867 (± 0.01) | 0.859 (± 0.02) | 0.789 (± 0.02) | 0.802 (± 0.03) | 0.906 (± 0.00) |
| 0.30 | 0.496 (± 0.13) | 0.800 (± 0.02) | **0.922** (± 0.03) | 0.744 (± 0.02) | 0.793 (± 0.01) | 0.864 (± 0.03) | 0.861 (± 0.01) | 0.727 (± 0.02) | 0.798 (± 0.03) | 0.877 (± 0.01) |
| 0.40 | 0.506 (± 0.12) | 0.772 (± 0.04) | **0.906** (± 0.03) | 0.701 (± 0.03) | 0.751 (± 0.03) | 0.850 (± 0.02) | 0.839 (± 0.01) | 0.674 (± 0.02) | 0.726 (± 0.03) | 0.864 (± 0.02) |
| 0.50 | 0.438 (± 0.12) | 0.743 (± 0.04) | **0.877** (± 0.01) | 0.648 (± 0.02) | 0.707 (± 0.02) | 0.842 (± 0.02) | 0.817 (± 0.04) | 0.642 (± 0.05) | 0.699 (± 0.08) | 0.837 (± 0.02) |
| 0.60 | 0.331 (± 0.05) | 0.688 (± 0.02) | **0.836** (± 0.03) | 0.594 (± 0.02) | 0.663 (± 0.01) | 0.807 (± 0.05) | 0.775 (± 0.02) | 0.590 (± 0.04) | 0.607 (± 0.03) | 0.806 (± 0.01) |
| 0.70 | 0.461 (± 0.14) | 0.626 (± 0.02) | **0.834** (± 0.04) | 0.514 (± 0.04) | 0.590 (± 0.04) | 0.776 (± 0.02) | 0.761 (± 0.04) | 0.482 (± 0.03) | 0.433 (± 0.09) | 0.760 (± 0.02) |
| 0.80 | 0.435 (± 0.12) | 0.570 (± 0.02) | 0.742 (± 0.06) | 0.463 (± 0.01) | 0.490 (± 0.04) | **0.743** (± 0.04) | 0.700 (± 0.01) | 0.436 (± 0.01) | 0.328 (± 0.03) | 0.707 (± 0.01) |
| 0.90 | 0.275 (± 0.01) | 0.484 (± 0.01) | 0.560 (± 0.22) | 0.330 (± 0.06) | 0.426 (± 0.03) | **0.663** (± 0.03) | 0.500 (± 0.17) | 0.352 (± 0.03) | 0.276 (± 0.01) | 0.620 (± 0.02) |
| 0.99 | 0.342 (± 0.12) | 0.377 (± 0.03) | 0.260 (± 0.00) | 0.260 (± 0.00) | 0.347 (± 0.01) | **0.629** (± 0.00) | 0.274 (± 0.01) | 0.286 (± 0.02) | 0.273 (± 0.01) | 0.537 (± 0.08) |

Table 36: F1 scores for ELECTRIC under mechanism *CD-MNAR* and varying $\mu$

| $\mu$ | GOODIE | GSPN | FairAC | FP | GNNmi | GCNmf | PCFI | GNNzero | GNNmedian | GNNmim |
|---|---|---|---|---|---|---|---|---|---|---|
| 0.00 | 0.588 (± 0.00) | 0.915 (± 0.00) | **0.963** (± 0.01) | 0.885 (± 0.00) | 0.929 (± 0.00) | 0.861 (± 0.00) | 0.903 (± 0.00) | 0.908 (± 0.01) | 0.908 (± 0.01) | 0.938 (± 0.01) |
| 0.10 | 0.486 (± 0.12) | 0.888 (± 0.01) | **0.962** (± 0.01) | 0.869 (± 0.01) | 0.874 (± 0.01) | 0.908 (± 0.02) | 0.885 (± 0.03) | 0.839 (± 0.03) | 0.860 (± 0.02) | 0.922 (± 0.00) |
| 0.20 | 0.476 (± 0.15) | 0.851 (± 0.02) | **0.931** (± 0.02) | 0.815 (± 0.01) | 0.802 (± 0.01) | 0.879 (± 0.01) | 0.879 (± 0.01) | 0.805 (± 0.03) | 0.801 (± 0.00) | 0.902 (± 0.03) |
| 0.30 | 0.478 (± 0.16) | 0.819 (± 0.04) | **0.922** (± 0.00) | 0.789 (± 0.03) | 0.789 (± 0.01) | 0.872 (± 0.01) | 0.880 (± 0.01) | 0.770 (± 0.05) | 0.736 (± 0.01) | 0.890 (± 0.00) |
| 0.40 | 0.431 (± 0.11) | 0.807 (± 0.02) | **0.902** (± 0.01) | 0.775 (± 0.01) | 0.762 (± 0.01) | 0.835 (± 0.02) | 0.865 (± 0.02) | 0.749 (± 0.03) | 0.685 (± 0.05) | 0.869 (± 0.02) |
| 0.50 | 0.450 (± 0.09) | 0.758 (± 0.02) | **0.867** (± 0.02) | 0.722 (± 0.02) | 0.748 (± 0.01) | 0.835 (± 0.03) | 0.827 (± 0.03) | 0.656 (± 0.03) | 0.633 (± 0.04) | 0.850 (± 0.02) |
| 0.60 | 0.436 (± 0.10) | 0.706 (± 0.01) | **0.853** (± 0.05) | 0.663 (± 0.02) | 0.608 (± 0.01) | 0.847 (± 0.03) | 0.780 (± 0.02) | 0.664 (± 0.03) | 0.593 (± 0.04) | 0.836 (± 0.02) |
| 0.70 | 0.337 (± 0.03) | 0.604 (± 0.03) | **0.812** (± 0.03) | 0.585 (± 0.03) | 0.538 (± 0.03) | 0.770 (± 0.09) | 0.729 (± 0.01) | 0.560 (± 0.02) | 0.514 (± 0.02) | 0.765 (± 0.01) |
| 0.80 | 0.411 (± 0.09) | 0.594 (± 0.02) | **0.824** (± 0.08) | 0.540 (± 0.01) | 0.486 (± 0.01) | 0.703 (± 0.04) | 0.671 (± 0.01) | 0.513 (± 0.01) | 0.469 (± 0.02) | 0.742 (± 0.02) |
| 0.90 | 0.392 (± 0.11) | 0.531 (± 0.02) | **0.735** (± 0.07) | 0.473 (± 0.03) | 0.449 (± 0.02) | 0.686 (± 0.06) | 0.600 (± 0.04) | 0.434 (± 0.02) | 0.445 (± 0.04) | 0.683 (± 0.02) |
| 0.99 | 0.304 (± 0.02) | 0.329 (± 0.04) | 0.264 (± 0.01) | 0.303 (± 0.02) | 0.294 (± 0.01) | **0.629** (± 0.00) | 0.312 (± 0.04) | 0.305 (± 0.02) | 0.292 (± 0.02) | 0.561 (± 0.02) |

Table 37: F1 scores for TADPOLE under mechanism *U-MCAR* and varying $\mu$

| $\mu$ | GOODIE | GSPN | FairAC | FP | GNNmi | GCNmf | PCFI | GNNzero | GNNmedian | GNNmim |
|---|---|---|---|---|---|---|---|---|---|---|
| 0.00 | 0.804 (± 0.00) | 0.648 (± 0.01) | 0.790 (± 0.00) | 0.806 (± 0.00) | 0.832 (± 0.02) | 0.786 (± 0.00) | 0.792 (± 0.00) | **0.847** (± 0.03) | 0.847 (± 0.03) | 0.809 (± 0.00) |
| 0.10 | 0.789 (± 0.00) | 0.590 (± 0.00) | 0.795 (± 0.00) | 0.801 (± 0.00) | 0.832 (± 0.00) | 0.809 (± 0.00) | 0.821 (± 0.00) | **0.841** (± 0.03) | 0.837 (± 0.03) | 0.820 (± 0.00) |
| 0.20 | 0.808 (± 0.00) | 0.590 (± 0.00) | 0.803 (± 0.00) | 0.823 (± 0.00) | **0.836** (± 0.00) | 0.779 (± 0.00) | 0.802 (± 0.00) | 0.833 (± 0.03) | 0.827 (± 0.04) | 0.799 (± 0.00) |
| 0.30 | 0.814 (± 0.00) | 0.567 (± 0.00) | 0.791 (± 0.00) | 0.806 (± 0.00) | **0.825** (± 0.00) | 0.757 (± 0.00) | 0.803 (± 0.00) | 0.811 (± 0.03) | 0.813 (± 0.03) | 0.802 (± 0.00) |
| 0.40 | 0.804 (± 0.00) | 0.610 (± 0.00) | **0.831** (± 0.00) | 0.800 (± 0.00) | 0.820 (± 0.00) | 0.794 (± 0.00) | 0.799 (± 0.00) | 0.830 (± 0.01) | 0.819 (± 0.02) | 0.805 (± 0.00) |
| 0.50 | 0.752 (± 0.00) | 0.581 (± 0.00) | 0.813 (± 0.00) | 0.809 (± 0.00) | **0.830** (± 0.00) | 0.799 (± 0.00) | 0.810 (± 0.00) | 0.797 (± 0.03) | 0.790 (± 0.03) | 0.814 (± 0.00) |
| 0.60 | 0.756 (± 0.00) | 0.575 (± 0.00) | 0.808 (± 0.00) | 0.785 (± 0.00) | 0.797 (± 0.00) | 0.722 (± 0.00) | 0.791 (± 0.00) | **0.810** (± 0.05) | 0.771 (± 0.04) | 0.799 (± 0.00) |
| 0.70 | 0.610 (± 0.00) | 0.552 (± 0.00) | 0.795 (± 0.00) | 0.740 (± 0.00) | 0.772 (± 0.00) | 0.729 (± 0.00) | 0.762 (± 0.00) | 0.779 (± 0.04) | 0.767 (± 0.03) | **0.802** (± 0.00) |
| 0.80 | 0.669 (± 0.00) | 0.552 (± 0.00) | **0.804** (± 0.00) | 0.757 (± 0.00) | 0.728 (± 0.00) | 0.669 (± 0.00) | 0.775 (± 0.00) | 0.760 (± 0.05) | 0.736 (± 0.04) | 0.764 (± 0.00) |
| 0.90 | 0.759 (± 0.00) | 0.590 (± 0.00) | 0.241 (± 0.00) | 0.758 (± 0.00) | 0.408 (± 0.00) | 0.608 (± 0.00) | 0.767 (± 0.00) | **0.786** (± 0.02) | 0.704 (± 0.02) | 0.763 (± 0.00) |
| 0.99 | **0.707** (± 0.00) | 0.523 (± 0.00) | 0.241 (± 0.00) | 0.241 (± 0.00) | 0.241 (± 0.00) | 0.353 (± 0.00) | 0.241 (± 0.00) | 0.507 (± 0.22) | 0.241 (± 0.00) | 0.700 (± 0.00) |

Table 38: F1 scores for TADPOLE under mechanism *S-MCAR* and varying $\mu$

| $\mu$ | GOODIE | GSPN | FairAC | FP | GNNmi | GCNmf | PCFI | GNNzero | GNNmedian | GNNmim |
|---|---|---|---|---|---|---|---|---|---|---|
| 0.00 | 0.804 (± 0.00) | 0.648 (± 0.01) | 0.790 (± 0.00) | 0.806 (± 0.00) | 0.832 (± 0.02) | 0.786 (± 0.00) | 0.792 (± 0.00) | **0.847** (± 0.03) | 0.847 (± 0.03) | 0.831 (± 0.04) |
| 0.10 | 0.554 (± 0.00) | 0.542 (± 0.00) | 0.805 (± 0.00) | 0.803 (± 0.00) | 0.815 (± 0.00) | 0.751 (± 0.00) | 0.804 (± 0.00) | **0.848** (± 0.02) | 0.846 (± 0.02) | 0.810 (± 0.00) |
| 0.20 | 0.497 (± 0.00) | 0.486 (± 0.00) | 0.818 (± 0.00) | 0.818 (± 0.00) | 0.811 (± 0.00) | 0.737 (± 0.00) | 0.814 (± 0.00) | **0.846** (± 0.02) | 0.845 (± 0.03) | 0.794 (± 0.00) |
| 0.30 | 0.523 (± 0.00) | 0.502 (± 0.00) | 0.775 (± 0.00) | 0.799 (± 0.00) | 0.825 (± 0.00) | 0.777 (± 0.00) | 0.818 (± 0.00) | 0.837 (± 0.02) | **0.838** (± 0.02) | 0.775 (± 0.00) |
| 0.40 | 0.482 (± 0.00) | 0.581 (± 0.00) | 0.800 (± 0.00) | 0.797 (± 0.00) | 0.794 (± 0.00) | 0.719 (± 0.00) | 0.784 (± 0.00) | 0.820 (± 0.03) | **0.823** (± 0.04) | 0.790 (± 0.00) |
| 0.50 | 0.501 (± 0.00) | 0.523 (± 0.00) | 0.757 (± 0.00) | 0.777 (± 0.00) | 0.769 (± 0.00) | 0.739 (± 0.00) | 0.798 (± 0.00) | **0.803** (± 0.02) | 0.797 (± 0.03) | 0.795 (± 0.00) |
| 0.60 | 0.539 (± 0.00) | 0.498 (± 0.00) | 0.802 (± 0.00) | 0.769 (± 0.00) | 0.734 (± 0.00) | 0.693 (± 0.00) | 0.804 (± 0.00) | 0.804 (± 0.05) | 0.799 (± 0.04) | **0.816** (± 0.00) |
| 0.70 | 0.480 (± 0.00) | 0.453 (± 0.00) | 0.748 (± 0.00) | 0.719 (± 0.00) | 0.738 (± 0.00) | 0.642 (± 0.00) | 0.752 (± 0.00) | 0.784 (± 0.03) | 0.777 (± 0.05) | **0.795** (± 0.00) |
| 0.80 | 0.502 (± 0.00) | 0.422 (± 0.00) | 0.689 (± 0.00) | 0.736 (± 0.00) | 0.703 (± 0.00) | 0.555 (± 0.00) | 0.730 (± 0.00) | 0.739 (± 0.02) | 0.740 (± 0.06) | **0.812** (± 0.00) |
| 0.90 | 0.377 (± 0.00) | 0.280 (± 0.00) | 0.503 (± 0.00) | 0.680 (± 0.00) | 0.650 (± 0.00) | 0.420 (± 0.00) | 0.739 (± 0.00) | 0.662 (± 0.07) | 0.557 (± 0.06) | **0.742** (± 0.00) |
| 0.99 | 0.272 (± 0.00) | 0.249 (± 0.00) | 0.241 (± 0.00) | **0.384** (± 0.00) | 0.241 (± 0.00) | 0.241 (± 0.00) | 0.241 (± 0.00) | 0.323 (± 0.05) | 0.241 (± 0.00) | 0.370 (± 0.00) |

Table 39: F1 scores for TADPOLE under mechanism *CD-MCAR* and varying $\mu$

| $\mu$ | GOODIE | GSPN | FairAC | FP | GNNmi | GCNmf | PCFI | GNNzero | GNNmedian | GNNmim |
|---|---|---|---|---|---|---|---|---|---|---|
| 0.00 | 0.804 (± 0.00) | 0.648 (± 0.01) | 0.790 (± 0.00) | 0.806 (± 0.00) | 0.832 (± 0.02) | 0.786 (± 0.00) | 0.792 (± 0.00) | **0.847** (± 0.03) | 0.847 (± 0.03) | 0.831 (± 0.04) |
| 0.10 | 0.786 (± 0.00) | 0.550 (± 0.00) | 0.765 (± 0.00) | 0.760 (± 0.00) | 0.793 (± 0.00) | 0.789 (± 0.00) | 0.785 (± 0.00) | 0.809 (± 0.03) | 0.809 (± 0.03) | **0.815** (± 0.00) |
| 0.20 | 0.785 (± 0.00) | 0.462 (± 0.00) | 0.758 (± 0.00) | 0.777 (± 0.00) | 0.786 (± 0.00) | 0.763 (± 0.00) | 0.804 (± 0.00) | **0.810** (± 0.04) | 0.810 (± 0.04) | 0.806 (± 0.00) |
| 0.30 | 0.654 (± 0.00) | 0.517 (± 0.00) | 0.766 (± 0.00) | 0.788 (± 0.00) | 0.784 (± 0.00) | 0.779 (± 0.00) | 0.782 (± 0.00) | **0.802** (± 0.04) | 0.802 (± 0.04) | 0.800 (± 0.00) |
| 0.40 | 0.685 (± 0.00) | 0.550 (± 0.00) | 0.780 (± 0.00) | 0.764 (± 0.00) | 0.780 (± 0.00) | 0.779 (± 0.00) | 0.774 (± 0.00) | **0.795** (± 0.03) | 0.795 (± 0.03) | 0.780 (± 0.00) |
| 0.50 | 0.778 (± 0.00) | 0.558 (± 0.00) | 0.700 (± 0.00) | 0.728 (± 0.00) | 0.776 (± 0.00) | 0.746 (± 0.00) | 0.731 (± 0.00) | 0.773 (± 0.04) | 0.773 (± 0.04) | **0.785** (± 0.00) |
| 0.60 | **0.783** (± 0.00) | 0.508 (± 0.00) | 0.731 (± 0.00) | 0.708 (± 0.00) | 0.729 (± 0.00) | 0.760 (± 0.00) | 0.714 (± 0.00) | 0.767 (± 0.03) | 0.767 (± 0.03) | 0.745 (± 0.00) |
| 0.70 | 0.725 (± 0.00) | 0.545 (± 0.00) | 0.684 (± 0.00) | 0.638 (± 0.00) | 0.663 (± 0.00) | 0.704 (± 0.00) | 0.710 (± 0.00) | **0.739** (± 0.03) | 0.739 (± 0.03) | 0.722 (± 0.00) |
| 0.80 | **0.656** (± 0.00) | 0.442 (± 0.00) | 0.576 (± 0.00) | 0.391 (± 0.00) | 0.442 (± 0.00) | 0.543 (± 0.00) | 0.419 (± 0.00) | 0.643 (± 0.04) | 0.643 (± 0.04) | 0.615 (± 0.00) |
| 0.90 | **0.704** (± 0.00) | 0.419 (± 0.00) | 0.241 (± 0.00) | 0.348 (± 0.00) | 0.361 (± 0.00) | 0.337 (± 0.00) | 0.292 (± 0.00) | 0.327 (± 0.03) | 0.327 (± 0.03) | 0.409 (± 0.00) |
| 0.99 | 0.687 (± 0.00) | 0.402 (± 0.00) | 0.241 (± 0.00) | 0.348 (± 0.00) | 0.361 (± 0.00) | 0.337 (± 0.00) | 0.292 (± 0.00) | **0.730** (± 0.03) | 0.567 (± 0.17) | 0.409 (± 0.00) |

Table 40: F1 scores for TADPOLE under mechanism *FD-MNAR* and varying $\mu$

| $\mu$ | GOODIE | GSPN | FairAC | FP | GNNmi | GCNmf | PCFI | GNNzero | GNNmedian | GNNmim |
|---|---|---|---|---|---|---|---|---|---|---|
| 0.00 | 0.804 (± 0.00) | 0.648 (± 0.01) | 0.790 (± 0.00) | 0.806 (± 0.00) | 0.832 (± 0.02) | 0.786 (± 0.00) | 0.792 (± 0.00) | 0.846 (± 0.03) | **0.849** (± 0.03) | 0.831 (± 0.04) |
| 0.10 | 0.546 (± 0.07) | 0.643 (± 0.01) | 0.801 (± 0.01) | 0.797 (± 0.01) | 0.822 (± 0.02) | 0.830 (± 0.04) | 0.838 (± 0.03) | 0.841 (± 0.03) | 0.842 (± 0.03) | **0.846** (± 0.04) |
| 0.20 | 0.531 (± 0.11) | 0.624 (± 0.05) | 0.793 (± 0.04) | **0.836** (± 0.01) | 0.810 (± 0.01) | 0.832 (± 0.02) | 0.827 (± 0.01) | 0.832 (± 0.03) | 0.817 (± 0.03) | 0.796 (± 0.00) |
| 0.30 | 0.573 (± 0.12) | 0.580 (± 0.04) | 0.804 (± 0.05) | 0.811 (± 0.03) | 0.806 (± 0.04) | 0.829 (± 0.04) | **0.831** (± 0.02) | 0.827 (± 0.03) | 0.802 (± 0.03) | 0.828 (± 0.03) |
| 0.40 | 0.562 (± 0.09) | 0.615 (± 0.03) | 0.751 (± 0.03) | 0.803 (± 0.01) | 0.793 (± 0.04) | **0.811** (± 0.02) | 0.802 (± 0.03) | 0.806 (± 0.03) | 0.803 (± 0.03) | 0.781 (± 0.02) |
| 0.50 | 0.673 (± 0.04) | 0.646 (± 0.07) | 0.793 (± 0.02) | 0.789 (± 0.02) | 0.796 (± 0.05) | 0.780 (± 0.01) | **0.815** (± 0.03) | 0.809 (± 0.04) | 0.805 (± 0.04) | 0.784 (± 0.03) |
| 0.60 | 0.529 (± 0.09) | 0.633 (± 0.06) | 0.722 (± 0.07) | 0.805 (± 0.04) | 0.785 (± 0.05) | 0.758 (± 0.04) | **0.810** (± 0.03) | 0.803 (± 0.04) | 0.792 (± 0.04) | 0.795 (± 0.03) |
| 0.70 | 0.634 (± 0.05) | 0.571 (± 0.04) | **0.804** (± 0.03) | 0.795 (± 0.04) | 0.746 (± 0.06) | 0.720 (± 0.06) | 0.795 (± 0.05) | 0.776 (± 0.05) | 0.748 (± 0.05) | 0.780 (± 0.03) |
| 0.80 | 0.378 (± 0.10) | 0.590 (± 0.06) | 0.612 (± 0.14) | 0.785 (± 0.02) | 0.692 (± 0.05) | 0.708 (± 0.02) | **0.797** (± 0.03) | 0.776 (± 0.04) | 0.720 (± 0.05) | 0.765 (± 0.00) |
| 0.90 | 0.309 (± 0.10) | 0.597 (± 0.01) | 0.241 (± 0.00) | 0.771 (± 0.03) | 0.663 (± 0.05) | 0.719 (± 0.01) | **0.787** (± 0.02) | 0.779 (± 0.03) | 0.703 (± 0.05) | 0.777 (± 0.06) |
| 0.99 | 0.241 (± 0.00) | 0.600 (± 0.05) | 0.241 (± 0.00) | 0.736 (± 0.03) | 0.241 (± 0.00) | 0.584 (± 0.05) | 0.241 (± 0.00) | 0.733 (± 0.01) | 0.241 (± 0.00) | **0.794** (± 0.04) |

Table 41: F1 scores for TADPOLE under mechanism *CD-MNAR* and varying $\mu$

| $\mu$ | GOODIE | GSPN | FairAC | FP | GNNmi | GCNmf | PCFI | GNNzero | GNNmedian | GNNmim |
|---|---|---|---|---|---|---|---|---|---|---|
| 0.00 | 0.804 (± 0.00) | 0.648 (± 0.01) | 0.790 (± 0.00) | 0.806 (± 0.00) | 0.832 (± 0.02) | 0.786 (± 0.00) | 0.792 (± 0.00) | **0.847** (± 0.03) | 0.847 (± 0.03) | 0.809 (± 0.00) |
| 0.10 | 0.553 (± 0.06) | 0.534 (± 0.09) | 0.793 (± 0.05) | 0.813 (± 0.03) | 0.829 (± 0.04) | 0.792 (± 0.03) | 0.806 (± 0.03) | **0.842** (± 0.02) | 0.826 (± 0.04) | 0.803 (± 0.01) |
| 0.20 | 0.485 (± 0.06) | 0.515 (± 0.04) | 0.804 (± 0.03) | 0.812 (± 0.03) | 0.832 (± 0.03) | 0.810 (± 0.02) | 0.806 (± 0.02) | **0.849** (± 0.01) | 0.826 (± 0.04) | 0.815 (± 0.02) |
| 0.30 | 0.441 (± 0.02) | 0.584 (± 0.06) | 0.805 (± 0.03) | 0.785 (± 0.02) | 0.811 (± 0.03) | 0.786 (± 0.02) | 0.812 (± 0.02) | **0.828** (± 0.03) | 0.813 (± 0.04) | 0.827 (± 0.03) |
| 0.40 | 0.502 (± 0.07) | 0.671 (± 0.03) | 0.828 (± 0.01) | 0.818 (± 0.03) | 0.808 (± 0.02) | 0.793 (± 0.02) | 0.814 (± 0.03) | 0.824 (± 0.02) | 0.826 (± 0.03) | **0.830** (± 0.01) |
| 0.50 | 0.448 (± 0.02) | 0.621 (± 0.04) | 0.784 (± 0.02) | 0.804 (± 0.04) | 0.799 (± 0.04) | 0.756 (± 0.03) | 0.803 (± 0.05) | 0.819 (± 0.02) | 0.800 (± 0.04) | **0.828** (± 0.04) |
| 0.60 | 0.457 (± 0.01) | 0.529 (± 0.03) | 0.791 (± 0.01) | 0.781 (± 0.03) | 0.803 (± 0.03) | 0.710 (± 0.07) | 0.797 (± 0.03) | **0.823** (± 0.04) | 0.783 (± 0.03) | 0.792 (± 0.03) |
| 0.70 | 0.485 (± 0.07) | 0.590 (± 0.09) | 0.639 (± 0.29) | 0.797 (± 0.05) | 0.787 (± 0.04) | 0.710 (± 0.05) | **0.822** (± 0.02) | 0.813 (± 0.03) | 0.784 (± 0.07) | 0.818 (± 0.01) |
| 0.80 | 0.376 (± 0.10) | 0.605 (± 0.04) | 0.434 (± 0.27) | 0.785 (± 0.05) | 0.767 (± 0.09) | 0.744 (± 0.01) | 0.798 (± 0.05) | **0.819** (± 0.04) | 0.776 (± 0.04) | 0.800 (± 0.02) |
| 0.90 | 0.362 (± 0.09) | 0.563 (± 0.03) | 0.241 (± 0.00) | 0.788 (± 0.01) | 0.730 (± 0.08) | 0.689 (± 0.05) | 0.776 (± 0.06) | 0.771 (± 0.06) | 0.704 (± 0.05) | **0.803** (± 0.05) |
| 0.99 | 0.324 (± 0.12) | 0.547 (± 0.08) | 0.241 (± 0.00) | 0.255 (± 0.02) | 0.241 (± 0.00) | 0.348 (± 0.05) | 0.241 (± 0.00) | 0.558 (± 0.15) | 0.241 (± 0.00) | **0.652** (± 0.04) |

## G   COMPLETE RESULT TABLES – R2 REGIME

This appendix complements the analysis of Research Question 3 (Section 4). It reports the complete set of results for the R2 regime, where training and test data are subject to different missingness mechanisms. We include both numerical tables (F1-score mean ± std over 5 runs) and extended visualizations across all models and datasets.

### G.1   NUMERICAL RESULTS

Table 42 reports the full F1-scores for all models, datasets, and shift configurations considered in the R2 regime.

Table 42: F1 (mean ± std over 5 runs). Setup: **R2** missingness distribution shift, where training data are subject to either *FD-MNAR* or *CD-MNAR*, while test data have either no missingness, 25% or 50% of *U-MCAR*

| Task | Train mech. | $\mu$ Test | GOODIE | GSPN | FairAC | GCNmf | PCFI | FP | GNNmi | GNNzero | GNNmedian | GNNmim |
|---|---|---|---|---|---|---|---|---|---|---|---|---|
| SYNTHETIC | *FD-MNAR* | 0 | 0.50 (± 0.15) | 0.68 (± 0.01) | 0.69 (± 0.05) | 0.81 (± 0.01) | 0.79 (± 0.02) | 0.80 (± 0.01) | 0.80 (± 0.01) | 0.81 (± 0.02) | 0.80 (± 0.02) | **0.82 (± 0.01)** |
| | *FD-MNAR* | 0.25 | 0.47 (± 0.13) | 0.64 (± 0.03) | 0.69 (± 0.04) | 0.74 (± 0.04) | 0.75 (± 0.03) | 0.76 (± 0.03) | 0.75 (± 0.03) | 0.76 (± 0.01) | 0.76 (± 0.02) | **0.77 (± 0.03)** |
| | *FD-MNAR* | 0.50 | 0.47 (± 0.13) | 0.64 (± 0.02) | 0.65 (± 0.04) | 0.71 (± 0.03) | 0.73 (± 0.02) | 0.71 (± 0.02) | 0.74 (± 0.02) | 0.71 (± 0.03) | 0.72 (± 0.04) | **0.73 (± 0.02)** |
| | *CD-MNAR* | 0 | 0.71 (± 0.07) | 0.70 (± 0.03) | 0.70 (± 0.05) | 0.80 (± 0.04) | 0.81 (± 0.02) | 0.80 (± 0.02) | 0.78 (± 0.02) | 0.82 (± 0.02) | 0.76 (± 0.02) | **0.85 (± 0.04)** |
| | *CD-MNAR* | 0.25 | 0.66 (± 0.05) | 0.68 (± 0.05) | 0.68 (± 0.03) | 0.75 (± 0.06) | 0.78 (± 0.04) | 0.77 (± 0.04) | 0.77 (± 0.02) | 0.78 (± 0.03) | 0.72 (± 0.03) | **0.80 (± 0.03)** |
| | *CD-MNAR* | 0.50 | 0.56 (± 0.10) | 0.64 (± 0.04) | 0.65 (± 0.01) | 0.73 (± 0.02) | 0.72 (± 0.03) | 0.72 (± 0.05) | 0.72 (± 0.01) | 0.72 (± 0.04) | 0.70 (± 0.01) | **0.75 (± 0.03)** |
| AIR | *FD-MNAR* | 0 | 0.50 (± 0.14) | 0.33 (± 0.04) | 0.66 (± 0.07) | 0.83 (± 0.05) | **0.88 (± 0.01)** | 0.86 (± 0.03) | 0.86 (± 0.03) | 0.85 (± 0.01) | 0.84 (± 0.03) | 0.87 (± 0.02) |
| | *FD-MNAR* | 0.25 | 0.51 (± 0.12) | 0.42 (± 0.04) | 0.65 (± 0.08) | 0.68 (± 0.05) | 0.83 (± 0.05) | 0.81 (± 0.02) | 0.81 (± 0.01) | 0.83 (± 0.01) | 0.80 (± 0.02) | **0.85 (± 0.01)** |
| | *FD-MNAR* | 0.50 | 0.52 (± 0.11) | 0.55 (± 0.03) | 0.70 (± 0.03) | 0.79 (± 0.03) | 0.71 (± 0.03) | **0.80 (± 0.07)** | 0.79 (± 0.06) | 0.78 (± 0.04) | 0.78 (± 0.01) | 0.80 (± 0.05) |
| | *CD-MNAR* | 0 | 0.56 (± 0.16) | 0.35 (± 0.02) | 0.65 (± 0.08) | 0.60 (± 0.20) | **0.88 (± 0.01)** | 0.71 (± 0.07) | 0.86 (± 0.06) | 0.83 (± 0.07) | 0.82 (± 0.03) | 0.85 (± 0.00) |
| | *CD-MNAR* | 0.25 | 0.56 (± 0.16) | 0.45 (± 0.50) | 0.70 (± 0.05) | 0.70 (± 0.05) | 0.84 (± 0.05) | 0.75 (± 0.05) | 0.84 (± 0.04) | 0.80 (± 0.05) | 0.79 (± 0.03) | **0.84 (± 0.06)** |
| | *CD-MNAR* | 0.50 | 0.62 (± 0.04) | 0.47 (± 0.04) | 0.68 (± 0.07) | 0.70 (± 0.02) | **0.80 (± 0.05)** | 0.72 (± 0.03) | 0.76 (± 0.05) | 0.76 (± 0.01) | 0.74 (± 0.03) | 0.76 (± 0.02) |
| ELECTRIC | *FD-MNAR* | 0 | 0.45 (± 0.11) | 0.67 (± 0.11) | **0.92 (± 0.02)** | 0.88 (± 0.12) | 0.69 (± 0.00) | 0.76 (± 0.03) | 0.80 (± 0.02) | 0.83 (± 0.05) | 0.79 (± 0.01) | 0.92 (± 0.01) |
| | *FD-MNAR* | 0.25 | 0.53 (± 0.10) | 0.68 (± 0.06) | **0.89 (± 0.00)** | 0.80 (± 0.02) | 0.73 (± 0.03) | 0.69 (± 0.03) | 0.74 (± 0.02) | 0.76 (± 0.03) | 0.73 (± 0.04) | 0.87 (± 0.01) |
| | *FD-MNAR* | 0.50 | 0.50 (± 0.10) | 0.68 (± 0.01) | **0.90 (± 0.02)** | 0.83 (± 0.01) | 0.75 (± 0.03) | 0.62 (± 0.02) | 0.66 (± 0.03) | 0.68 (± 0.02) | 0.66 (± 0.02) | 0.82 (± 0.02) |
| | *CD-MNAR* | 0 | 0.52 (± 0.10) | 0.78 (± 0.04) | 0.92 (± 0.02) | 0.86 (± 0.02) | 0.88 (± 0.01) | 0.83 (± 0.05) | 0.81 (± 0.01) | 0.81 (± 0.01) | 0.79 (± 0.02) | **0.94 (± 0.00)** |
| | *CD-MNAR* | 0.25 | 0.50 (± 0.10) | 0.78 (± 0.01) | **0.88 (± 0.01)** | 0.86 (± 0.02) | 0.85 (± 0.02) | 0.74 (± 0.04) | 0.73 (± 0.03) | 0.72 (± 0.01) | 0.73 (± 0.02) | 0.85 (± 0.03) |
| | *CD-MNAR* | 0.50 | 0.49 (± 0.12) | 0.70 (± 0.02) | **0.87 (± 0.02)** | 0.82 (± 0.03) | 0.81 (± 0.00) | 0.66 (± 0.01) | 0.70 (± 0.03) | 0.65 (± 0.02) | 0.68 (± 0.02) | 0.83 (± 0.02) |
| TADPOLE | *FD-MNAR* | 0 | 0.52 (± 0.07) | 0.53 (± 0.00) | 0.75 (± 0.03) | 0.74 (± 0.05) | 0.79 (± 0.00) | 0.77 (± 0.00) | 0.76 (± 0.01) | 0.79 (± 0.01) | 0.77 (± 0.02) | **0.83 (± 0.02)** |
| | *FD-MNAR* | 0.25 | 0.48 (± 0.03) | 0.48 (± 0.02) | 0.77 (± 0.01) | 0.73 (± 0.01) | **0.82 (± 0.02)** | 0.78 (± 0.03) | 0.76 (± 0.03) | 0.78 (± 0.03) | 0.74 (± 0.03) | 0.81 (± 0.01) |
| | *FD-MNAR* | 0.50 | 0.48 (± 0.04) | 0.53 (± 0.02) | 0.79 (± 0.02) | 0.71 (± 0.04) | 0.78 (± 0.02) | 0.74 (± 0.02) | 0.73 (± 0.03) | 0.74 (± 0.04) | 0.71 (± 0.02) | **0.82 (± 0.03)** |
| | *CD-MNAR* | 0 | 0.60 (± 0.02) | 0.26 (± 0.02) | 0.79 (± 0.05) | 0.75 (± 0.04) | **0.80 (± 0.04)** | 0.80 (± 0.03) | 0.79 (± 0.05) | 0.79 (± 0.04) | 0.75 (± 0.04) | 0.79 (± 0.06) |
| | *CD-MNAR* | 0.25 | 0.47 (± 0.09) | 0.52 (± 0.02) | 0.82 (± 0.05) | 0.78 (± 0.01) | **0.80 (± 0.04)** | **0.80 (± 0.04)** | 0.77 (± 0.04) | 0.78 (± 0.04) | 0.73 (± 0.06) | 0.75 (± 0.03) |
| | *CD-MNAR* | 0.50 | 0.49 (± 0.07) | 0.62 (± 0.05) | 0.81 (± 0.03) | 0.75 (± 0.00) | 0.79 (± 0.01) | **0.82 (± 0.02)** | 0.76 (± 0.03) | 0.76 (± 0.05) | 0.73 (± 0.06) | 0.74 (± 0.02) |

### G.2   EXTENDED VISUALIZATIONS

In addition to Figure 3 in the main paper, Figures 6 and 8 report the full results for all models under both training mechanisms.

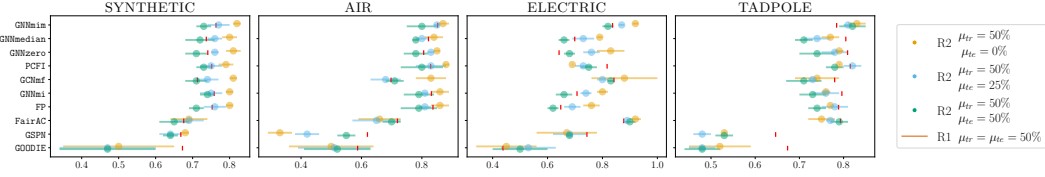

Figure 6: Full results for all models trained with *FD-MNAR* at $\mu_{\text{tr}} = 50\%$, tested on *U-MCAR* with $\mu_{\text{te}} \in \{0\%, 25\%, 50\%\}$. Each panel corresponds to one dataset; each row to one model. Reported values are mean ± std over 5 runs.

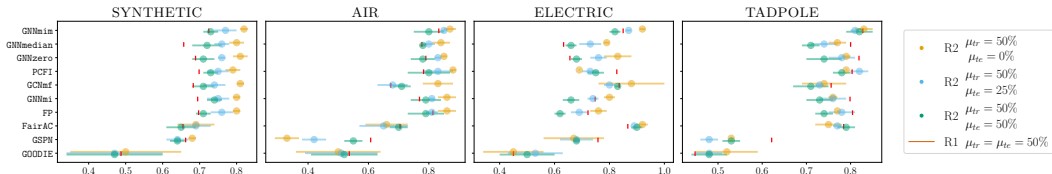

Figure 7: Full results for all models trained with *CD-MNAR* at $\mu_{\text{tr}} = 50\%$, tested on *U-MCAR* with $\mu_{\text{te}} \in \{0\%, 25\%, 50\%\}$. Same layout as Figure 6.

## H  INDUCTIVE SYNTHETIC SETTING

In addition to the transductive experiments reported in the main paper, we also ran a set of experiments in an inductive setting to demonstrate that our model, `GNNmim`, is not restricted to transductive scenarios. As shown in Figure 8, `GNNmim` remains competitive with all other baselines even under this inductive setup.

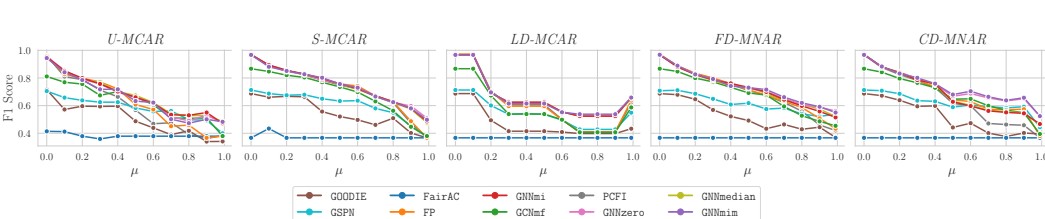

Figure 8: Performance of `GNNmim` and all competitors in an inductive setting. The synthetic dataset is constructed so that test nodes form a separate graph component and are never connected to training nodes, ensuring that no message can propagate between the two sets during training. Despite this strictly inductive setup, `GNNmim` remains competitive with all baselines.

Table 43: F1 scores for  under mechanism *CDMNAR* and varying $\mu$

| $\mu$ | GOODIE | GSPN | FairAC | FP | GNNmi | GCNmf | PCFI | GNNzero | GNNmedian | GNNmim |
|---|---|---|---|---|---|---|---|---|---|---|
| 0.00 | 0.687 (± 0.166) | 0.713 (± 0.045) | 0.367 (± 0.000) | **0.972 (± 0.011)** | 0.968 (± 0.011) | 0.867 (± 0.023) | 0.970 (± 0.011) | 0.968 (± 0.011) | 0.968 (± 0.011) | 0.967 (± 0.011) |
| 0.10 | 0.672 (± 0.167) | 0.708 (± 0.022) | 0.367 (± 0.000) | 0.880 (± 0.014) | 0.881 (± 0.014) | 0.842 (± 0.010) | 0.876 (± 0.011) | 0.875 (± 0.018) | 0.878 (± 0.020) | **0.883 (± 0.020)** |
| 0.20 | 0.639 (± 0.151) | 0.686 (± 0.048) | 0.367 (± 0.000) | 0.836 (± 0.015) | 0.838 (± 0.022) | 0.796 (± 0.026) | 0.825 (± 0.018) | 0.840 (± 0.022) | **0.842 (± 0.020)** | 0.832 (± 0.019) |
| 0.30 | 0.595 (± 0.122) | 0.636 (± 0.031) | 0.367 (± 0.000) | 0.785 (± 0.020) | 0.785 (± 0.034) | 0.765 (± 0.036) | 0.782 (± 0.023) | 0.796 (± 0.029) | 0.793 (± 0.026) | **0.801 (± 0.020)** |
| 0.40 | 0.598 (± 0.119) | 0.631 (± 0.043) | 0.367 (± 0.000) | 0.734 (± 0.019) | 0.758 (± 0.024) | 0.729 (± 0.021) | 0.731 (± 0.008) | 0.754 (± 0.017) | 0.750 (± 0.023) | **0.759 (± 0.017)** |
| 0.50 | 0.442 (± 0.092) | 0.589 (± 0.029) | 0.367 (± 0.000) | 0.643 (± 0.036) | 0.628 (± 0.040) | 0.647 (± 0.041) | 0.616 (± 0.029) | 0.668 (± 0.023) | 0.632 (± 0.030) | **0.680 (± 0.018)** |
| 0.60 | 0.473 (± 0.063) | 0.605 (± 0.034) | 0.367 (± 0.000) | 0.629 (± 0.031) | 0.597 (± 0.029) | 0.649 (± 0.041) | 0.600 (± 0.052) | 0.687 (± 0.013) | 0.602 (± 0.033) | **0.704 (± 0.021)** |
| 0.70 | 0.401 (± 0.070) | 0.592 (± 0.024) | 0.367 (± 0.000) | 0.574 (± 0.016) | 0.562 (± 0.007) | 0.599 (± 0.064) | 0.471 (± 0.041) | 0.656 (± 0.023) | 0.566 (± 0.018) | **0.664 (± 0.027)** |
| 0.80 | 0.377 (± 0.012) | 0.584 (± 0.026) | 0.367 (± 0.000) | 0.571 (± 0.026) | 0.551 (± 0.020) | 0.567 (± 0.044) | 0.463 (± 0.069) | 0.634 (± 0.025) | 0.557 (± 0.016) | **0.638 (± 0.028)** |
| 0.90 | 0.402 (± 0.062) | 0.592 (± 0.031) | 0.367 (± 0.000) | 0.574 (± 0.048) | 0.544 (± 0.020) | 0.548 (± 0.052) | 0.458 (± 0.046) | 0.650 (± 0.033) | 0.547 (± 0.028) | **0.657 (± 0.020)** |
| 0.99 | 0.395 (± 0.052) | 0.444 (± 0.060) | 0.367 (± 0.000) | 0.380 (± 0.022) | 0.467 (± 0.020) | 0.395 (± 0.035) | 0.367 (± 0.000) | **0.524 (± 0.045)** | 0.464 (± 0.013) | **0.524 (± 0.045)** |

Table 44: F1 scores for  under mechanism *FDMNAR* and varying $\mu$

| $\mu$ | GOODIE | GSPN | FairAC | FP | GNNmi | GCNmf | PCFI | GNNzero | GNNmedian | GNNmim |
|---|---|---|---|---|---|---|---|---|---|---|
| 0.00 | 0.687 (± 0.166) | 0.708 (± 0.045) | 0.367 (± 0.000) | **0.972 (± 0.011)** | 0.967 (± 0.011) | 0.867 (± 0.022) | 0.968 (± 0.013) | 0.967 (± 0.011) | 0.967 (± 0.011) | 0.968 (± 0.011) |
| 0.10 | 0.679 (± 0.166) | 0.711 (± 0.012) | 0.367 (± 0.000) | 0.888 (± 0.013) | 0.879 (± 0.024) | 0.847 (± 0.013) | 0.885 (± 0.014) | 0.882 (± 0.022) | 0.886 (± 0.020) | **0.889 (± 0.017)** |
| 0.20 | 0.646 (± 0.154) | 0.686 (± 0.033) | 0.367 (± 0.000) | **0.834 (± 0.024)** | 0.825 (± 0.024) | 0.799 (± 0.016) | 0.832 (± 0.026) | 0.830 (± 0.022) | 0.825 (± 0.025) | 0.826 (± 0.028) |
| 0.30 | 0.569 (± 0.133) | 0.649 (± 0.013) | 0.367 (± 0.000) | **0.800 (± 0.042)** | 0.786 (± 0.034) | 0.772 (± 0.028) | 0.796 (± 0.025) | 0.789 (± 0.036) | 0.782 (± 0.032) | 0.793 (± 0.036) |
| 0.40 | 0.522 (± 0.134) | 0.608 (± 0.037) | 0.367 (± 0.000) | 0.759 (± 0.021) | **0.761 (± 0.027)** | 0.732 (± 0.032) | 0.753 (± 0.026) | 0.757 (± 0.032) | 0.743 (± 0.028) | 0.742 (± 0.032) |
| 0.50 | 0.492 (± 0.135) | 0.618 (± 0.008) | 0.367 (± 0.000) | 0.714 (± 0.016) | 0.731 (± 0.015) | 0.692 (± 0.027) | 0.710 (± 0.028) | 0.724 (± 0.017) | **0.736 (± 0.018)** | 0.730 (± 0.019) |
| 0.60 | 0.433 (± 0.084) | 0.575 (± 0.025) | 0.367 (± 0.000) | 0.675 (± 0.031) | 0.699 (± 0.032) | 0.676 (± 0.022) | 0.674 (± 0.039) | 0.702 (± 0.030) | 0.687 (± 0.027) | **0.716 (± 0.031)** |
| 0.70 | 0.464 (± 0.090) | 0.582 (± 0.020) | 0.367 (± 0.000) | 0.630 (± 0.031) | 0.643 (± 0.037) | 0.594 (± 0.040) | 0.623 (± 0.035) | 0.651 (± 0.037) | 0.635 (± 0.033) | **0.661 (± 0.019)** |
| 0.80 | 0.429 (± 0.065) | 0.540 (± 0.009) | 0.367 (± 0.000) | 0.586 (± 0.021) | 0.598 (± 0.027) | 0.527 (± 0.053) | 0.560 (± 0.030) | 0.607 (± 0.029) | 0.609 (± 0.019) | **0.620 (± 0.024)** |
| 0.90 | 0.444 (± 0.082) | 0.522 (± 0.034) | 0.367 (± 0.000) | 0.508 (± 0.105) | 0.558 (± 0.049) | 0.486 (± 0.061) | 0.460 (± 0.129) | 0.589 (± 0.042) | 0.575 (± 0.044) | **0.592 (± 0.023)** |
| 0.99 | 0.370 (± 0.005) | 0.538 (± 0.041) | 0.367 (± 0.000) | 0.433 (± 0.093) | 0.515 (± 0.035) | 0.454 (± 0.076) | 0.420 (± 0.105) | **0.561 (± 0.036)** | 0.521 (± 0.040) | 0.550 (± 0.040) |

Table 45: F1 scores for under mechanism *LDMCAR* and varying $\mu$

| $\mu$ | GOODIE | GSPN | FairAC | FP | GNNmi | GCNmf | PCFI | GNNzero | GNNmedian | GNNmim |
|---|---|---|---|---|---|---|---|---|---|---|
| 0.00 | 0.687 (± 0.166) | 0.713 (± 0.045) | 0.367 (± 0.000) | **0.972 (± 0.011)** | 0.968 (± 0.011) | 0.867 (± 0.023) | 0.970 (± 0.011) | 0.968 (± 0.011) | 0.968 (± 0.011) | 0.967 (± 0.011) |
| 0.10 | 0.687 (± 0.166) | 0.713 (± 0.045) | 0.367 (± 0.000) | **0.972 (± 0.011)** | 0.968 (± 0.011) | 0.867 (± 0.023) | 0.970 (± 0.011) | 0.968 (± 0.011) | 0.968 (± 0.011) | 0.967 (± 0.011) |
| 0.20 | 0.494 (± 0.117) | 0.601 (± 0.039) | 0.367 (± 0.000) | 0.701 (± 0.023) | 0.692 (± 0.031) | 0.673 (± 0.036) | **0.705 (± 0.019)** | 0.692 (± 0.031) | 0.692 (± 0.031) | 0.696 (± 0.029) |
| 0.30 | 0.415 (± 0.076) | 0.537 (± 0.032) | 0.367 (± 0.000) | 0.596 (± 0.010) | **0.624 (± 0.010)** | 0.539 (± 0.028) | 0.606 (± 0.006) | **0.624 (± 0.010)** | **0.624 (± 0.010)** | 0.615 (± 0.011) |
| 0.40 | 0.415 (± 0.076) | 0.543 (± 0.037) | 0.367 (± 0.000) | 0.596 (± 0.010) | **0.624 (± 0.010)** | 0.539 (± 0.028) | 0.606 (± 0.006) | **0.624 (± 0.010)** | **0.624 (± 0.010)** | 0.615 (± 0.011) |
| 0.50 | 0.415 (± 0.076) | 0.537 (± 0.032) | 0.367 (± 0.000) | 0.596 (± 0.010) | **0.624 (± 0.010)** | 0.539 (± 0.028) | 0.606 (± 0.006) | **0.624 (± 0.010)** | **0.624 (± 0.010)** | 0.615 (± 0.011) |
| 0.60 | 0.409 (± 0.053) | 0.495 (± 0.044) | 0.367 (± 0.000) | 0.497 (± 0.015) | **0.555 (± 0.019)** | 0.498 (± 0.022) | 0.501 (± 0.022) | **0.555 (± 0.019)** | **0.555 (± 0.019)** | 0.552 (± 0.027) |
| 0.70 | 0.398 (± 0.037) | 0.428 (± 0.030) | 0.367 (± 0.000) | 0.410 (± 0.027) | 0.524 (± 0.044) | 0.407 (± 0.051) | 0.407 (± 0.025) | 0.524 (± 0.044) | 0.524 (± 0.044) | **0.538 (± 0.023)** |
| 0.80 | 0.398 (± 0.037) | 0.428 (± 0.030) | 0.367 (± 0.000) | 0.410 (± 0.027) | 0.524 (± 0.044) | 0.407 (± 0.051) | 0.407 (± 0.025) | 0.524 (± 0.044) | 0.524 (± 0.044) | **0.538 (± 0.023)** |
| 0.90 | 0.398 (± 0.037) | 0.428 (± 0.030) | 0.367 (± 0.000) | 0.410 (± 0.027) | 0.524 (± 0.044) | 0.407 (± 0.051) | 0.407 (± 0.025) | 0.524 (± 0.044) | 0.524 (± 0.044) | **0.538 (± 0.023)** |
| 0.99 | 0.433 (± 0.069) | 0.549 (± 0.024) | 0.367 (± 0.000) | 0.637 (± 0.036) | 0.659 (± 0.029) | 0.587 (± 0.031) | 0.623 (± 0.027) | **0.660 (± 0.025)** | 0.652 (± 0.025) | 0.658 (± 0.023) |

Table 46: F1 scores for under mechanism *SMCAR* and varying $\mu$

| $\mu$ | GOODIE | GSPN | FairAC | FP | GNNmi | GCNmf | PCFI | GNNzero | GNNmedian | GNNmim |
|---|---|---|---|---|---|---|---|---|---|---|
| 0.00 | 0.687 (± 0.166) | 0.713 (± 0.045) | 0.367 (± 0.000) | **0.972 (± 0.011)** | 0.968 (± 0.011) | 0.867 (± 0.023) | 0.970 (± 0.011) | 0.968 (± 0.011) | 0.968 (± 0.011) | 0.967 (± 0.011) |
| 0.10 | 0.661 (± 0.148) | 0.687 (± 0.013) | 0.434 (± 0.133) | 0.887 (± 0.012) | **0.894 (± 0.016)** | 0.847 (± 0.025) | 0.891 (± 0.018) | **0.894 (± 0.017)** | 0.890 (± 0.021) | 0.881 (± 0.018) |
| 0.20 | 0.667 (± 0.157) | 0.675 (± 0.036) | 0.367 (± 0.000) | 0.850 (± 0.017) | 0.855 (± 0.027) | 0.820 (± 0.030) | **0.856 (± 0.025)** | 0.847 (± 0.018) | 0.851 (± 0.027) | 0.851 (± 0.028) |
| 0.30 | 0.664 (± 0.155) | 0.679 (± 0.034) | 0.367 (± 0.000) | **0.830 (± 0.016)** | 0.829 (± 0.032) | 0.804 (± 0.028) | 0.822 (± 0.025) | 0.828 (± 0.032) | 0.824 (± 0.034) | 0.827 (± 0.038) |
| 0.40 | 0.557 (± 0.152) | 0.650 (± 0.029) | 0.367 (± 0.000) | 0.785 (± 0.030) | 0.796 (± 0.035) | 0.769 (± 0.018) | 0.785 (± 0.029) | 0.785 (± 0.043) | 0.790 (± 0.039) | **0.802 (± 0.032)** |
| 0.50 | 0.521 (± 0.152) | 0.633 (± 0.045) | 0.367 (± 0.000) | 0.757 (± 0.029) | 0.758 (± 0.018) | 0.735 (± 0.019) | 0.748 (± 0.030) | **0.760 (± 0.018)** | 0.755 (± 0.019) | 0.756 (± 0.009) |
| 0.60 | 0.497 (± 0.135) | 0.636 (± 0.058) | 0.367 (± 0.000) | **0.742 (± 0.030)** | 0.722 (± 0.034) | 0.698 (± 0.021) | 0.723 (± 0.038) | 0.724 (± 0.039) | 0.716 (± 0.031) | 0.730 (± 0.027) |
| 0.70 | 0.461 (± 0.125) | 0.580 (± 0.062) | 0.367 (± 0.000) | 0.670 (± 0.018) | 0.671 (± 0.029) | 0.631 (± 0.036) | 0.666 (± 0.038) | **0.673 (± 0.030)** | 0.672 (± 0.028) | 0.666 (± 0.035) |
| 0.80 | 0.509 (± 0.121) | 0.549 (± 0.071) | 0.367 (± 0.000) | 0.628 (± 0.053) | **0.629 (± 0.025)** | 0.563 (± 0.070) | 0.621 (± 0.044) | 0.623 (± 0.013) | 0.622 (± 0.025) | 0.625 (± 0.037) |
| 0.90 | 0.402 (± 0.071) | 0.455 (± 0.068) | 0.367 (± 0.000) | 0.487 (± 0.070) | 0.580 (± 0.043) | 0.447 (± 0.060) | 0.474 (± 0.092) | **0.597 (± 0.026)** | 0.575 (± 0.039) | 0.580 (± 0.027) |
| 0.99 | 0.367 (± 0.000) | 0.372 (± 0.010) | 0.367 (± 0.000) | 0.367 (± 0.000) | 0.486 (± 0.027) | 0.380 (± 0.019) | 0.367 (± 0.000) | **0.509 (± 0.038)** | 0.476 (± 0.024) | 0.498 (± 0.031) |

Table 47: F1 scores for under mechanism *UMCAR* and varying $\mu$

| $\mu$ | GOODIE | GSPN | FairAC | FP | GNNmi | GCNmf | PCFI | GNNzero | GNNmedian | GNNmim |
|---|---|---|---|---|---|---|---|---|---|---|
| 0.00 | 0.715 (± 0.096) | 0.705 (± 0.033) | 0.414 (± 0.055) | 0.960 (± 0.009) | 0.953 (± 0.006) | 0.811 (± 0.030) | **0.960 (± 0.009)** | 0.953 (± 0.006) | 0.953 (± 0.006) | 0.944 (± 0.017) |
| 0.10 | 0.572 (± 0.137) | 0.658 (± 0.031) | 0.412 (± 0.057) | 0.827 (± 0.050) | 0.851 (± 0.043) | 0.769 (± 0.112) | 0.810 (± 0.034) | **0.855 (± 0.044)** | 0.846 (± 0.047) | 0.841 (± 0.051) |
| 0.20 | 0.596 (± 0.165) | 0.638 (± 0.025) | 0.379 (± 0.000) | 0.798 (± 0.033) | **0.799 (± 0.020)** | 0.756 (± 0.032) | 0.788 (± 0.027) | 0.790 (± 0.028) | 0.788 (± 0.021) | 0.785 (± 0.021) |
| 0.30 | 0.594 (± 0.145) | 0.625 (± 0.014) | 0.359 (± 0.040) | **0.771 (± 0.037)** | 0.757 (± 0.046) | 0.674 (± 0.133) | 0.712 (± 0.045) | 0.758 (± 0.049) | **0.771 (± 0.042)** | 0.718 (± 0.047) |
| 0.40 | 0.596 (± 0.132) | 0.625 (± 0.006) | 0.379 (± 0.000) | **0.721 (± 0.055)** | 0.702 (± 0.044) | 0.702 (± 0.055) | 0.664 (± 0.080) | 0.697 (± 0.049) | 0.701 (± 0.048) | 0.718 (± 0.029) |
| 0.50 | 0.487 (± 0.113) | 0.583 (± 0.040) | 0.379 (± 0.000) | 0.608 (± 0.067) | 0.660 (± 0.027) | 0.664 (± 0.053) | 0.568 (± 0.074) | 0.659 (± 0.021) | **0.674 (± 0.022)** | 0.633 (± 0.035) |
| 0.60 | 0.439 (± 0.118) | 0.558 (± 0.034) | 0.379 (± 0.000) | 0.572 (± 0.077) | 0.617 (± 0.038) | 0.606 (± 0.081) | 0.469 (± 0.102) | 0.617 (± 0.038) | **0.622 (± 0.039)** | **0.622 (± 0.062)** |
| 0.70 | 0.390 (± 0.074) | **0.561 (± 0.019)** | 0.379 (± 0.000) | 0.451 (± 0.092) | 0.534 (± 0.073) | 0.511 (± 0.095) | 0.476 (± 0.118) | 0.518 (± 0.076) | 0.541 (± 0.089) | 0.502 (± 0.092) |
| 0.80 | 0.418 (± 0.123) | 0.499 (± 0.029) | 0.379 (± 0.000) | 0.459 (± 0.074) | **0.530 (± 0.060)** | 0.508 (± 0.088) | 0.392 (± 0.087) | 0.490 (± 0.059) | 0.528 (± 0.044) | 0.473 (± 0.052) |
| 0.90 | 0.340 (± 0.048) | 0.493 (± 0.022) | 0.379 (± 0.000) | 0.367 (± 0.046) | **0.550 (± 0.139)** | 0.511 (± 0.082) | 0.362 (± 0.041) | 0.532 (± 0.134) | 0.529 (± 0.131) | 0.501 (± 0.122) |
| 0.99 | 0.341 (± 0.045) | 0.400 (± 0.025) | 0.379 (± 0.000) | 0.379 (± 0.000) | 0.472 (± 0.022) | 0.380 (± 0.003) | 0.384 (± 0.011) | 0.476 (± 0.038) | **0.485 (± 0.018)** | 0.483 (± 0.033) |

# I   GAIN USING MIM WITH COMPETITORS

Tables 48 through 51 report the performance gain observed when all competitor models described in the main paper are equipped with the MIM mask, mirroring the setup used for `GNNmim`. Consistently, basic imputation methods that replace missing features with a constant, such as `GNNmi` and `GNNmedian`, show a positive and comparable performance increase when supplied with the same mask. This suggests that the improvement comes from the model's ability to selectively ignore the padded or imputed feature values indicated by the mask.

Table 48: F1 gain from using mask on SYNTHETIC under mechanism *U-MCAR*

| $\mu$ | FairAC | FP | GCNmf | GNNmedian | GNNmi | GOODIE | GSPN | PCFI | GNNzero |
|---|---|---|---|---|---|---|---|---|---|
| 0.00 | -0.087 | -0.016 | -0.145 | 0.002 | 0.003 | -0.256 | -0.094 | -0.020 | 0.005 |
| 0.10 | -0.094 | -0.022 | -0.065 | 0.006 | 0.005 | -0.253 | -0.080 | -0.004 | 0.001 |
| 0.20 | -0.102 | -0.013 | -0.005 | 0.002 | 0.004 | -0.215 | -0.052 | -0.001 | 0.008 |
| 0.30 | -0.078 | 0.002 | -0.021 | 0.012 | 0.014 | -0.198 | -0.068 | -0.008 | 0.015 |
| 0.40 | -0.082 | 0.008 | -0.022 | 0.012 | 0.07 | -0.223 | -0.075 | 0.006 | 0.025 |
| 0.50 | 0.011 | -0.006 | -0.010 | 0.005 | 0.09 | -0.268 | -0.079 | -0.018 | 0.007 |
| 0.60 | -0.025 | -0.004 | -0.029 | 0.004 | 0.013 | -0.346 | -0.072 | -0.001 | 0.000 |
| 0.70 | 0.013 | 0.001 | -0.044 | 0.005 | 0.004 | -0.321 | -0.008 | 0.006 | 0.006 |
| 0.80 | -0.070 | -0.008 | 0.009 | 0.002 | 0.015 | -0.429 | 0.015 | -0.014 | 0.039 |
| 0.90 | -0.020 | -0.017 | -0.011 | 0.011 | 0.014 | -0.346 | 0.053 | 0.001 | 0.001 |
| 0.99 | 0.052 | -0.007 | 0.056 | -0.020 | -0.013 | -0.422 | 0.024 | -0.011 | -0.013 |

Table 49: F1 gain from using mask on SYNTHETIC under mechanism *S-MCAR*

| $\mu$ | FairAC | FP | GCNmf | GNNmedian | GNNmi | GOODIE | GSPN | PCFI | GNNzero |
|---|---|---|---|---|---|---|---|---|---|
| 0.00 | -0.080 | -0.016 | -0.145 | 0.002 | 0.003 | -0.256 | -0.091 | 0.05 | 0.005 |
| 0.10 | 0.013 | 0.001 | -0.077 | 0.03 | 0.04 | -0.211 | 0.005 | -0.11 | -0.011 |
| 0.20 | -0.018 | -0.039 | -0.086 | 0.003 | 0.007 | -0.245 | -0.019 | -0.026 | 0.031 |
| 0.30 | 0.000 | -0.026 | -0.083 | 0.006 | 0.015 | -0.234 | -0.013 | -0.015 | 0.016 |
| 0.40 | 0.010 | -0.034 | -0.012 | 0.002 | 0.019 | -0.185 | -0.014 | -0.018 | 0.024 |
| 0.50 | -0.062 | -0.048 | 0.005 | 0.006 | 0.016 | -0.207 | -0.036 | -0.039 | 0.033 |
| 0.60 | -0.045 | -0.028 | -0.038 | 0.018 | 0.032 | -0.161 | 0.001 | -0.026 | 0.038 |
| 0.70 | 0.009 | -0.007 | -0.025 | 0.011 | 0.025 | -0.153 | -0.015 | -0.033 | 0.064 |
| 0.80 | 0.010 | -0.011 | -0.046 | 0.011 | 0.02 | -0.136 | -0.002 | 0.004 | 0.029 |
| 0.90 | -0.045 | 0.003 | -0.018 | 0.002 | -0.002 | -0.071 | 0.043 | -0.000 | -0.019 |
| 0.99 | 0.128 | -0.024 | 0.074 | 0.002 | -0.015 | 0.048 | 0.033 | -0.025 | -0.011 |

Table 50: F1 gain from using mask on SYNTHETIC under mechanism *LD-MCAR*

| $\mu$ | FairAC | FP | GCNmf | GNNmedian | GNNmi | GOODIE | GSPN | PCFI | GNNzero |
|---|---|---|---|---|---|---|---|---|---|
| 0.00 | -0.073 | -0.016 | -0.145 | 0.002 | 0.003 | -0.256 | -0.094 | -0.020 | 0.005 |
| 0.10 | -0.047 | 0.104 | -0.012 | 0.026 | 0.095 | -0.222 | -0.014 | 0.097 | -0.08 |
| 0.20 | -0.105 | -0.078 | -0.081 | 0.004 | 0.075 | -0.251 | -0.092 | -0.067 | 0.081 |
| 0.30 | -0.106 | -0.119 | -0.106 | 0.015 | 0.101 | -0.331 | -0.091 | -0.118 | 0.133 |
| 0.40 | 0.014 | -0.044 | -0.049 | 0.015 | 0.039 | -0.337 | -0.054 | -0.033 | 0.098 |
| 0.50 | 0.080 | -0.002 | -0.004 | 0.015 | 0.002 | -0.362 | 0.027 | 0.003 | 0.077 |
| 0.60 | -0.079 | -0.073 | -0.068 | 0.004 | 0.081 | -0.386 | -0.046 | -0.069 | 0.139 |
| 0.70 | -0.111 | -0.084 | -0.034 | 0.001 | 0.070 | -0.423 | -0.039 | -0.060 | 0.139 |
| 0.80 | 0.001 | -0.084 | -0.074 | 0.001 | 0.085 | -0.422 | -0.056 | -0.086 | 0.130 |
| 0.90 | -0.067 | -0.090 | -0.066 | 0.001 | 0.096 | -0.439 | 0.023 | -0.072 | 0.143 |
| 0.99 | 0.046 | 0.037 | -0.054 | 0.007 | 0.014 | -0.359 | 0.025 | 0.039 | 0.020 |

Table 51: F1 gain from using mask on SYNTHETIC under mechanism *FD-MNAR*

| $\mu$ | FairAC | FP | GCNmf | GNNmedian | GNNmi | GOODIE | GSPN | PCFIGNNzero | |
|---|---|---|---|---|---|---|---|---|---|
| 0.00 | -0.080 | -0.018 | -0.141 | 0.002 | 0.003 | -0.256 | -0.081 | -0.018 | 0.005 |
| 0.10 | -0.035 | -0.006 | -0.057 | 0.007 | 0.013 | -0.216 | -0.002 | -0.001 | 0.014 |
| 0.20 | 0.018 | 0.015 | 0.024 | 0.06 | 0.005 | -0.193 | -0.012 | -0.009 | -0.005 |
| 0.30 | 0.021 | 0.002 | -0.005 | 0.002 | 0.007 | -0.138 | 0.015 | 0.016 | -0.018 |
| 0.40 | 0.001 | -0.007 | -0.031 | 0.006 | 0.011 | -0.186 | -0.032 | 0.003 | 0.021 |
| 0.50 | -0.025 | -0.011 | -0.020 | 0.008 | 0.013 | -0.208 | -0.009 | -0.007 | 0.022 |
| 0.60 | 0.011 | 0.006 | -0.019 | 0.012 | 0.008 | -0.121 | 0.030 | 0.013 | 0.013 |
| 0.70 | 0.022 | 0.029 | 0.004 | 0.000 | 0.003 | -0.063 | 0.044 | 0.013 | 0.012 |
| 0.80 | 0.010 | 0.013 | -0.010 | 0.002 | 0.001 | -0.006 | -0.017 | 0.033 | 0.020 |
| 0.90 | 0.053 | 0.032 | -0.032 | 0.005 | 0.011 | 0.048 | -0.023 | 0.020 | 0.018 |
| 0.99 | 0.156 | 0.002 | -0.008 | 0.001 | 0.010 | -0.015 | 0.006 | -0.003 | 0.007 |

Table 52: F1 gain from using mask on SYNTHETIC under mechanism *CD-MNAR*

| $\mu$ | FairAC | FP | GCNmf | GNNmedian | GNNmi | GOODIE | GSPN | PCFIGNNzero | |
|---|---|---|---|---|---|---|---|---|---|
| 0.00 | -0.078 | -0.016 | -0.145 | 0.002 | 0.003 | -0.256 | -0.091 | -0.020 | 0.005 |
| 0.10 | -0.025 | -0.002 | -0.060 | 0.004 | 0.010 | -0.239 | -0.019 | -0.005 | 0.001 |
| 0.20 | 0.023 | 0.006 | -0.003 | 0.004 | 0.002 | -0.202 | -0.029 | 0.004 | 0.001 |
| 0.30 | -0.005 | 0.017 | -0.004 | 0.009 | 0.007 | -0.121 | -0.030 | -0.006 | 0.023 |
| 0.40 | -0.045 | 0.017 | -0.015 | 0.014 | 0.017 | 0.005 | -0.024 | 0.021 | 0.020 |
| 0.50 | -0.035 | 0.010 | 0.001 | 0.048 | 0.010 | -0.035 | -0.042 | 0.009 | 0.036 |
| 0.60 | 0.054 | 0.036 | -0.011 | 0.019 | 0.015 | -0.111 | -0.047 | 0.073 | 0.037 |
| 0.70 | 0.038 | 0.051 | 0.001 | 0.025 | 0.028 | -0.064 | 0.031 | 0.072 | 0.026 |
| 0.80 | 0.045 | 0.046 | 0.047 | 0.017 | 0.011 | -0.028 | -0.021 | 0.086 | 0.037 |
| 0.90 | 0.136 | 0.033 | 0.039 | 0.011 | 0.021 | -0.009 | -0.047 | 0.075 | 0.037 |
| 0.99 | 0.098 | -0.041 | 0.057 | 0.017 | 0.015 | -0.050 | 0.044 | 0.013 | 0.018 |

