# OpenReview forum: "Rethinking GNNs and Missing Features: Challenges, Evaluation and a Robust Solution"
_ICLR.cc/2026/Conference — Submitted to ICLR 2026_

### Official Review · Reviewer_Gu9p · 2025-10-26

**Soundness:** 4
**Presentation:** 4
**Contribution:** 4
**Rating:** 6
**Confidence:** 3

**Summary:**

This paper identifies two key challenges with existing datasets to evaluate missing feature-based GNNs: a) High-dimensional but sparse node features where most values are zero, b) Uniform MCAR: a missing mechanism applied to generate incomplete or missing features, which the authors claim to be overly simplistic. Moreover, they identify a more realistic missing setting where the missing feature distribution is different during training and testing.

To establish these challenges, authors first theoretically show that when feature sparsity is high, a large amount of missingness is required to degrade the model quality and produce meaningful loss of information. The author introduces 1 synthetic and 3 real-world datasets to establish a better benchmark.

Then they establish multiple missingness mechanisms for evaluating and benchmarking the methods and datasets. Namely, they introduce label-dependent MCAR, feature-dependent MNAR, and class-dependent MNAR.

Finally, they introduce from their theory, the Missing Indicator Method (NIM) to model missing features in graph machine learning, GNNmin, where they concatenate the missing feature mask as additional input to GNN.

The authors show that this simplistic method performs robustly across all datasets and various missing conditions.

**Strengths:**

Authors have identified a relevant problem for industry-level deployments of GNNs. The Authors have established the significance of their motivation well enough with good theoretical insights. They have proposed a novel evaluation design with a novel introduction of distribution shift between train and test feature missingness. The proposed method is simple and effective.

**Weaknesses:**

A) The proposed method is non-inductive, i.e., it won't be able to handle new nodes in the graph. This, I think, is its major limitation.
B) Overall, I think the positioning of the paper can be improved further if the authors can position the proposed method as an effective baseline, as the paper's core novelty lies in identifying the challenges with the current evaluation setup.  Moreover, this method can supplement existing baselines as well to improve their performance.
C) Comparison with simple baselines like 0 imputation or mean/median imputation will be advisable to further strengthen the claims.
D) Moreover, with datasets like Cora, CiteSeer, etc, what happens if we ignore all zero feature values and apply missingness on non-zero values only? Will that solve the core issue highlighted with these datasets.

**Questions:**

I request the authors to respond to the questions raised in the weakness section.

---

> ### Author Response · Authors · 2025-11-18
>
> We thank the reviewer for the constructive and encouraging feedback, particularly for highlighting the relevance of our motivation, the strength of our theoretical insights, and the simplicity and effectiveness of our proposed evaluation design and model. We address the raised weaknesses and questions below.
>
> **Limitation in inductive settings**
>
> The experimental setup we used is transductive, as this is the setting adopted by the vast majority of related work on node classification, including all baselines we compare against. We thank the reviewer for pointing this out, and we have clarified it more explicitly in the revised manuscript at the beginning of Section 5.
> However, there are no intrinsic limitations that prevent our method, GNNmim, from operating in an inductive setting as well. To demonstrate this, we have added a new experiment in Appendix H, where we construct an inductive synthetic dataset, ensuring that test nodes are completely unseen during training. As shown by the reported results, the overall behavior mirrors the transductive setting, with GNNmim remaining competitive relative to all competing methods.
>
> **Experiments with existing models**
>
> We agree that our method serves as an effective baseline, and we have updated the paper to explicitly present it as such. In addition, we have included Appendix I, where we report the performance gains obtained by competitor models when the MIM mask is provided alongside their input features.
> We show that basic imputation methods (GNNmi, GNNmedian, GNNzero) obtain a positive gain when equipped with the MIM mask, as in the case of imputing the missing feature values with a constant as ideally the model learns to ignore such values (as we stated in footnote 1, page 4).
>
>
> **Experiments with mean, zero , median imputation**
>
> We would like to point out that the mean-imputation baseline is already included in our experiments and corresponds to the model denoted as GNNmi. Following your suggestion, we have updated the paper and the experimental section to include two further baselines, GNNzero and GNNmedian, which replace missing features with zeros and feature-wise medians, respectively. As expected, their performance is generally lower than GNNmim, which explicitly encodes the missingness mask.
>
>
> **Missingness on 1-values on Cora/Citeseer/Pubmed**
>
> If we ignore all zero entries and mask only the non-zero values (the “1”s in the BoW vectors), the resulting setting corresponds to a specific MNAR mechanism, since missingness depends directly on the true feature value. In that sense, we agree with the reviewer that this would create a harder benchmark than standard MCAR, and the information loss for the class label is much faster (as a function of the missingness percentage) than what we get in Theorem 2 for MCAR.
> However, this construction is not realistic for BoW-style features. In Cora, CiteSeer, and PubMed, features are binary indicators of word occurrence. These feature vectors will usually be constructed by starting with an all-0 vector, and inserting 1’s for terms that appear in the document. While it is possible that this process leads to ‘false 0’s’, i.e., terms whose occurrence has not been recognized due to parsing or spelling errors, there is really no mechanism that leads to an explicit ‘?’ in this data. Our goal is to study practically meaningful missingness mechanisms, such as arising from sensor dropouts, reporting bias, or feature-dependent data acquisition. That said we ran an experiment applying missingness only on 1-features in the Cora dataset. In spite of the apparently harder task, the results obtained are similar to the “easy” setting presented in Figure 4 (top left plot). This similarity likely stems from the fact that even when a fairly high percentage of  the 1-values are masked, the aggregate of the remaining data from all neighbors still contains a sufficient signal for an accurate prediction (helped by a high feature and label homophily). In this experiment GNNmim obtains stable performance even when features are 100% missing, because, by masking only the 1-features, the M matrix, essentially contains the same or similar information as the original features.

---

> > ### Author Response · Authors · 2025-11-24
> >
> > Dear Reviewer,
> >
> > Thank you again for your helpful feedback. We would kindly like to ask whether our responses have satisfactorily addressed your concerns, or if any points remain unclear.
> > We would be happy to provide further explanation if needed.
> >
> > Thank you for your time and consideration.

---

### Official Review · Reviewer_TVVf · 2025-10-29

**Soundness:** 3
**Presentation:** 3
**Contribution:** 3
**Rating:** 6
**Confidence:** 3

**Summary:**

This paper primarily addresses the issue of handling missing node features in the practical application of Graph Neural Networks and challenges the evaluation methods used in existing research. It ultimately proposes a more challenging benchmark and a simple yet effective model.

**Strengths:**

1.Theoretical proof: The paper reveals the limitations of traditional sparse datasets when evaluating the robustness of missing features, making the evaluation process more scientifically grounded.

2.Introduction of new datasets: New datasets with dense, raw, and semantically meaningful features are introduced, providing more challenging and representative benchmarks for future research.

3.Proposing a more robust model: GNNmim is a simple yet effective new approach that consistently outperforms complex existing specialized models across various datasets and missing mechanisms.

**Weaknesses:**

1.The proposed synthetic/real-world datasets have relatively few features, reflecting the nature of the original measurements, but this also limits the direct generalizability of the findings in high-dimensional sparse feature scenarios.

2.GNNmim uses zero-padding as a placeholder for missing values, which is an arbitrary replacement rather than a semantically meaningful feature attribution. More theoretical analysis is needed to justify the appropriateness of this approach. Additionally, the effectiveness of GNNmim depends on the accurate construction of the missing indicator matrix M. If the missing information contains errors or uncertainties, model performance may be affected.

3.While the proposed MNAR generation mechanisms (FD-MNAR, CD-MNAR) are more realistic than MCAR, they are still synthetic and may not capture all the complex non-random missing patterns encountered in real-world scenarios.

**Questions:**

Please refer to Weaknesses part.

---

> ### Author Response · Authors · 2025-11-18
>
> We thank the reviewer for the constructive comments and for highlighting the strengths of our theoretical analysis, benchmark design, and the effectiveness of GNNmim. We address the noted weaknesses and questions below.
>
> **Generalizability**
>
> We agree that our synthetic and real-world datasets have relatively few features; this directly reflects the nature of the underlying measurements (e.g., physical or clinical variables), which are typically low-dimensional. Importantly, this design choice is intentional: our theory shows that in high-dimensional sparse settings (e.g., BoW/TF-IDF), missingness has almost no effect on the information content, making robustness evaluations unreliable. This is precisely why existing benchmarks fail to differentiate methods. Our goal is not to cover all feature regimes, but to focus on those where missingness is genuinely challenging and meaningful, i.e., dense, semantically meaningful features. That said, generalization to higher-dimensional settings is still supported in two ways:
> (1) Empirical evidence on a larger synthetic graph (Appendix E) shows consistent trends when both the number of nodes and features are increased.
> (2) GNNmim itself is agnostic to feature dimensionality and has the same computational complexity as the base GNN; nothing prevents its use in sparse/high-dimensional regimes.
> We have clarified these points more explicitly in the revised paper, at the end of Section 3.
>
>
> **Theoretical analysis of MIM mask**
>
> (1) The main element of the GNNmim design is the inclusion of the missing indicator matrix as part of the input, thus enabling the use of missingness data, especially in informative NMAR settings. As clarified in the footnote on page 4 (that the reviewer also seems to refer to), the zero-padding is a somewhat arbitrary choice that would not matter at all if the model successfully learned to ignore whichever constant is used to fill in missing values. Since this ideal outcome is difficult to achieve in practice, the use of  zeros is a practically motivated choice: zeros will behave more ‘neutral’ than other constants, e.g. minimizing the impact on the computed gradients by the fill-in values.  To analyze this point more carefully, we added an ablation in Appendix I in which we replace zeros with mean and median imputation (columns 4 and 5 of Tables from 48 to 52). Across all missingness settings, we observe a positive and consistent performance gain when the missingness mask is used, regardless of which value is imputed (but mostly a small advantage for zero padding). GNNmim is intended as a minimal proof-of-concept model demonstrating the usefulness of the missing indicator method in the GNN context. More sophisticated solutions with stronger theoretical guarantees are the subject of future work.
>
> (2) In our setting, the missingness indicator matrix M is directly and deterministically constructed from the observed dataset. Missing values are explicitly marked in the raw data (with a dedicated symbol “?”, see Theorem 2), so the mask M is uniquely defined and contains no uncertainty. Therefore, GNNmim’s effectiveness does not rely on learning or inferring M; the mask simply encodes which entries are observed and which are not. To avoid ambiguity, we have now clarified in the revised version (Section 2)  that (i) missing entries are explicitly identified in the provided datasets, and (ii) the mask M is deterministically constructed from these identifiers, with no estimation or uncertainty involved.
>
> **More realistic MNAR scenarios**
>
> We fully agree that our MNAR mechanisms capture only a subset of the possible non-random missingness patterns that may arise in practice. Exploring more realistic and graph-specific mechanisms, where missingness depends also/primarily on the graph structure itself, is indeed an important direction for future work. At the same time, we note that ultimately realistic MNAR behavior can only be evaluated on truly incomplete real-world graph datasets, which are currently missing in the literature. We have clarified these aspects in the revised Conclusion and Future Work section.

---

> > ### Author Response · Authors · 2025-11-24
> >
> > Dear Reviewer,
> >
> > We sincerely appreciate your review and the insights you provided. We just wished to check whether our responses have resolved your comments, or if you believe additional clarification could still strengthen the paper.
> > Please let us know if any further detail would be useful.
> >
> > Thank you for your time and consideration

---

> > ### Comment · Reviewer_TVVf · 2025-11-26
> > **Official Comment by Reviewer TVVf**
> >
> > Thank you very much for the detailed response and clarifications. They have addressed most of my concerns and also highlighted several interesting directions for future work. After carefully reconsidering the overall quality of the paper, I have decided to maintain my current positive score.

---

### Official Review · Reviewer_EzuY · 2025-10-31

**Soundness:** 2
**Presentation:** 2
**Contribution:** 2
**Rating:** 2
**Confidence:** 3

**Summary:**

The paper revisits learning on graphs with missing node features and argues that common benchmarks and MCAR-only masking protocols obscure the true difficulty of the problem due to extreme feature sparsity and simplistic missingness. It contributes (i) an information-theoretic analysis showing sparsity dampens the impact of additional missingness, (ii) one synthetic and three real-world, denser-feature datasets, (iii) more realistic MCAR/MNAR protocols with train–test shifts, and (iv) GNNmim, a missing-indicator GNN that treats the mask as input rather than imputing features. Experiments show GNNmim is consistently robust across datasets and mechanisms.

**Strengths:**

1. The motivation to investigate why existing benchmarks and missingness assumptions fail to reflect real-world scenarios is well justified.
2. The proposed GNNmim framework is simple yet effective, showing consistent robustness across various missing-feature settings.

**Weaknesses:**

1. The information-theoretic results (e.g., ∆ bounds/data-processing) are important but scattered; adding “takeaway” remarks after each theorem and clarifying practical conditions (e.g., bandwidth/feature distributions) would help adoption.
2. More direct comparisons and discussion versus imputation-free conditional models and density-oriented approaches are needed.
3. The experimental studies are not strong enough since they are conducted on limited benchmark datasets and lack evaluations on larger or more realistic scenarios that can demonstrate the practical effectiveness and robustness of the proposed approach.
4. The significance of this work in advancing or motivating future research directions in graph neural networks is not clearly established.

**Questions:**

1. I am curious about, how would you recommend practitioners detect and handle mask-distribution shift at test time when using GNNmim, and can a simple calibration or reweighting mitigate the R2 drop you observe?
2. Can you quantify when MIM beats imputation (e.g., under which MNAR strengths or graph homophily levels), and provide a small decision guide for method choice?
3. Could you report ablation on the mask channel (mask only, features only, both) to isolate the gain from explicit missingness modeling?

---

> ### Author Response · Authors · 2025-11-18
>
> We thank the reviewer for the thoughtful feedback and for highlighting the relevance of our motivation and the effectiveness of GNNmim under different missing-feature settings. We address the raised concerns and questions below.
>
> **Takeaway of theorems**
>
> We thank the reviewer for this suggestion. In the revised version of the paper, we now include short “Intuition” remarks immediately after both Theorem 1 and Theorem 2 to make their practical implications explicit.
>
> Intuition for Theorem 1
>
> If the data is feature-MAR and label-MAR, then no further assumptions or modeling efforts about the missingness mechanism are needed: in all cases the likelihood function (or loss function) reduces to the form of equation (6), which does not contain the missingness mechanism $P_\lambda$ any longer.
>
> Intuition for Theorem 2
>
> 1) Under label-MAR, the missingness pattern carries no predictive information, and predicting the label from incomplete data is at least as hard as prediction from the complete data.
> 2) When node features are extremely sparse (e.g., BoW/TF-IDF), the information loss induced by missingness is provably negligible unless missingness is extremely high. As a result, existing sparse benchmarks inherently make all methods appear robust, preventing meaningful comparison.
>
>
> **More comparisons**
>
>  We have included in our experimental section  all existing GNN methods specifically designed to handle missing node features that we are aware of. If the reviewer has specific additional works in mind that  we have missed, we would be grateful for  references and we will try to include them in the revised version.
>
> **Limited benchmarks**
>
> We agree that strong empirical validation is essential. For studying missing node features, suitable datasets must satisfy simultaneously: (i) dense, moderately sized, semantically meaningful features (sparse/BoW features make missingness almost irrelevant, as shown in our theory), (ii) non-trivial accuracy on complete data, and (iii) feature–structure complementarity (Coupette et al., 2025). Despite extensive searching, we found no real-world large-scale graphs satisfying all these requirements, a limitation highlighted also by recent benchmark critiques in GNN research (Bechler-Speicher et al., 2025). We would be happy to include any datasets the reviewer is aware of that meet these criteria.
> Regarding scalability, GNNmim has the same computational complexity as the underlying GNN, as it only appends the missingness mask. If the backbone scales to millions of nodes, so does GNNmim. Additional runtime experiments (Appendix E) confirm this. Moreover, experiments on a larger synthetic graph (Appendix E) show that increasing graph size and degree does not degrade GNNmim relative to baselines.
> We have clarified these points more explicitly in the revised paper, at the end of Section 3.
>
> **Significance not clear**
>
> The core ambition of our work is precisely to redefine how research on GNNs with missing features should move forward. Our paper highlights that progress in this area has been limited by the evaluation itself: existing benchmarks rely on sparse, non-informative features and overly benign missingness mechanisms. These choices make current results difficult to interpret and prevent a real understanding of robustness. By providing new datasets, realistic missingness protocols, and a clear theoretical framing, we establish a foundation for more meaningful research directions. In this context, GNNmim serves as a proof of concept showing that, when evaluation is done correctly, a simple assumption-free model outperforms more complex ones.
> We have clarified the aim and scope of our work at the end of the introduction in the revised version of the paper.
>
> **Mask Distribution Shift**
>
> In practice, a mask-distribution shift will also lead to a shift in the distribution of the observed data (the $\tilde{X}$ in our Theorem 2), and thus would be detectable by statistical tests (assuming a sufficient amount of data to test on). How to handle a detected shift is a much more difficult question, however. Data distribution shifts are a generally unsolved problem in machine learning, and the special case of shifts in the missingness mechanism are unlikely to have special properties that make this case more amenable to simple solutions than the general case. In case a distribution shift has been detected, a general suggestion can be to revert to methods that in theory are more robust with respect to the underlying missingness mechanism, as discussed in the following point.

---

> ### Author Response · Authors · 2025-11-18
>
> **When is MIM helpful?**
>
> Our theory provides qualitative guidance on when GNNmim is preferable:
> (1) Theoretical decision guide.
> • MNAR, no train–test shift: MIM/GNNmim is theoretically preferred, as it makes no assumptions about the missingness process, and can exploit dependencies between missingness and class label.
> • MAR, no shift: MIM and GCNmf are both theoretically justified t.
> • MAR, with shift: GCNmf is theoretically safer, as it does not encode  the specific missingness pattern of the training data .
> • MNAR, with shift: no method has theoretical guarantees.
> Notably, simple imputations (mean/median/zero) have no theoretical justification in any regime; their appeal is purely practical when missingness is negligible.
> (2) Empirical evidence.
> Despite the decision structure above, our experiments show that GNNmim remains competitive even in regimes where theory offers it no advantage, including under train–test distribution shifts.
> (3) Role of homophily.
> Homophily does not affect our theoretical analysis; the key determinant is the missingness mechanism (MAR vs. MNAR and alignment across train/test), not structural properties of the graph.
>
> **Ablation**
>
> The feature-only baseline (i.e., GNN with zero imputation) is now  included among the main baselines. In most cases, 0-imputation performs on a par with mean imputation, and is less accurate than GNNmim. To address the question about explicit missingness modeling, we additionally ran a mask-only variant of GNNmim. As expected, the missingness mask alone carries almost no predictive signal, since it encodes only the presence/absence of features while discarding all feature information. For completeness, we report below the F1 scores obtained on the synthetic dataset under varying missingness levels, showing the limited predictive power of this variant.
>
> ### F1 scores of the mask-only variant (synthetic dataset)
>
> | μ    | U-MCAR          | S-MCAR          | LD-MCAR         | FD-MNAR         | CD-MNAR         |
> |------|------------------|------------------|------------------|------------------|------------------|
> | 0.00 | 0.338 ± 0.000 | 0.338 ± 0.000 | 0.338 ± 0.000 | 0.338 ± 0.000 | 0.338 ± 0.000 |
> | 0.10 | 0.410 ± 0.066 | 0.345 ± 0.014 | 0.338 ± 0.000 | 0.413 ± 0.065 | 0.403 ± 0.044 |
> | 0.20 | 0.413 ± 0.053 | 0.364 ± 0.033 | 0.338 ± 0.000 | 0.482 ± 0.063 | 0.532 ± 0.049 |
> | 0.30 | 0.507 ± 0.037 | 0.371 ± 0.068 | 0.338 ± 0.000 | 0.520 ± 0.041 | 0.573 ± 0.032 |
> | 0.40 | 0.510 ± 0.063 | 0.397 ± 0.080 | 0.338 ± 0.000 | 0.529 ± 0.073 | 0.594 ± 0.029 |
> | 0.50 | 0.533 ± 0.046 | 0.466 ± 0.082 | 0.338 ± 0.000 | 0.571 ± 0.018 | 0.622 ± 0.038 |
> | 0.60 | 0.521 ± 0.046 | 0.459 ± 0.063 | 0.346 ± 0.019 | 0.567 ± 0.042 | 0.661 ± 0.035 |
> | 0.70 | 0.549 ± 0.040 | 0.403 ± 0.051 | 0.419 ± 0.055 | 0.572 ± 0.031 | 0.626 ± 0.029 |
> | 0.80 | 0.565 ± 0.044 | 0.449 ± 0.060 | 0.413 ± 0.048 | 0.591 ± 0.028 | 0.589 ± 0.029 |
> | 0.90 | 0.553 ± 0.051 | 0.479 ± 0.041 | 0.413 ± 0.048 | 0.551 ± 0.049 | 0.591 ± 0.028 |
> | 0.99 | 0.522 ± 0.034 | 0.502 ± 0.028 | 0.528 ± 0.032 | 0.528 ± 0.026 | 0.532 ± 0.031 |

---

> > ### Author Response · Authors · 2025-11-24
> >
> > Dear Reviewer,
> >
> > Thank you once again for the constructive comments. We would like to politely ask whether our rebuttal has fully addressed your concerns, or if any points remain unclear.
> > We remain available for any further clarification.
> >
> > Thank you for your time and consideration.

---

> > ### Comment · Reviewer_EzuY · 2025-11-25
> >
> > Thanks to the authors for providing the requested clarifications. Some of my concerns have been addressed, and I have accordingly updated my score in a more positive direction. However, I still believe that the positioning of the paper can be strengthened. In particular, it would be valuable to more clearly frame the proposed method as an effective baseline that naturally arises from the paper’s main contribution, namely the identification and analysis of the limitations of the current evaluation setup. It may also help to explicitly discuss how this method can serve as a complementary component to existing baselines, potentially improving their performance. I encourage the authors to incorporate these points and the above clarifications into the revised version of the manuscript for the benefit of future readers.

---

> > > ### Author Response · Authors · 2025-11-25
> > >
> > > We thank the reviewer for the thoughtful follow-up and for the constructive suggestions, which helped us strengthen the positioning of the paper.
> > >
> > > As suggested, we have revised the manuscript to more clearly articulate how our GNNmim emerges naturally from the analysis of the limitations of the current evaluation setup. In particular, we added a dedicated explanation at the end of Section 1 (Introduction) that explicitly frames GNNmim as an effective baseline motivated directly by our analysis.
> > >
> > > Regarding the reviewer’s suggestion to discuss how our masking mechanism can complement existing baselines, we have previously incorporated a new experiment in Appendix I in order to answer to reviewer Gu9p. As shown in the new table, all imputation methods benefit from the inclusion of the mask. This finding aligns with the intuition formalized in Footnote 1 (page 4): ideally, a model should be capable of ignoring features that the mask designates as missing, regardless of the specific imputed value.
> > >
> > > We hope these additions improve the clarity and impact of the paper.

---

### Official Review · Reviewer_iRLs · 2025-11-03

**Soundness:** 4
**Presentation:** 4
**Contribution:** 4
**Rating:** 8
**Confidence:** 4

**Summary:**

This paper presents the case that existing GNN evaluations with missing features and sparse datasets are fundamentally flawed. The paper then introduces a benchmark with dense datasets and reasonable missing feature patterns e.g. MNAR. Additionally, they introduce a simple GNN model that concats mask for missing features, and empirically show that on this new benchmark, this simple/baseline model outperforms most existing GNNs.

**Strengths:**

S1: The main premise is quite interesting, and having myself evaluated GNN models on sparse datasets, I can clearly see how it could have been flawed. The experiments and theoretical analysis (sparsity and information loss connection) make argument intuitive.



S2: Contributes datasets that elicits the flaws of earlier benchmarks by showing that even a simple GNN (with mask for missing features) is competitive.

S3. Paper is well written, easy to follow, and claims are supported by corresponding experiments.

**Weaknesses:**

W1. The graph scale of real-world dataset is quite small. Most real datasets currently have millions of nodes. How do results generalize to large scale graphs with dense features? Also will GNNmim remain competitive in such graphs?
Further the benchmark consists of 3 graphs – they may not be representative of real graphs. How non-trivial it is to expand the benchmark?

W2. The paper emphasizes on GNNmim as a new method, however, the main contribution seems to be the benchmark developed to assess GNNs appropriately.

W3. The paper could benefit from discussion on what makes GNNmim effective.What it learns, or exploits that is present in dense graphs.

**Questions:**

Q1. Do you think the wordings for GNNmim could be changed from consistently outperformance to is competitive?
Q2. Given your results, I’d be curious to see how even more simpler baseline like zero-imputation GNN perform? Will it be comparable to GNNmim?

---

> ### Author Response · Authors · 2025-11-18
>
> We thank the reviewer for the constructive comments and for the positive remarks regarding our main premise, analysis, and benchmark. We address the raised questions and concerns below.
>
> **Small datasets**
>
> We agree that larger real-world graphs are desirable. For studying missing node features, suitable datasets must satisfy simultaneously: (i) dense, low-dimensional features, since sparse high-dimensional ones (e.g., BoW/TF-IDF) make missingness almost inconsequential (as shown in our theory); (ii)  GNNs achieve high accuracy with complete features; ensuring the task is meaningful; (iii) feature–structure complementarity and separability (Coupette et al., 2025). Despite extensive searching, we did not find real-world large-scale datasets that meet all these requirements simultaneously. This aligns with recent observations about the scarcity of suitable GNN benchmarks (Bechler-Speicher et al., 2025). We emphasize that this is not a limitation of our work, but a structural limitation of current publicly available datasets. We would be delighted to incorporate any dataset suggestions from the reviewer that meet the above criteria.
>
> **Will GNNmim remain competitive on large graphs?**
>
> GNNmim has the same computational complexity as the underlying GNN, since it only concatenates the missingness mask to the input features. If the backbone scales to millions of nodes, GNNmim scales identically. Additional runtime experiments (now included in Appendix E of the revised paper) confirm scalability. A potential scale-related factor is higher average node degree. Preliminary results on our larger synthetic dataset (Appendix E) indicate that increased degree does not harm GNNmim relative to competitors.
>
> **Benchmark extensibility**
>
> Expanding the benchmark is conceptually straightforward but practically non-trivial. The difficulty is not architectural, but rather the scarcity of datasets satisfying all three properties listed above. Our work aims also to highlight the urgent need for more realistic datasets tailored to missing-feature studies, a gap increasingly acknowledged in recent position papers on graph learning benchmarks. We are actively exploring additional datasets and welcome any pointers from the community.
>
> We have clarified these points more explicitly in the revised paper, at the end of Section 3.
>
>
> **Contribution non clear**
>
> We thank the reviewer for the insightful observation. Indeed, our primary objective is not to propose a new architecture per se, but to rethink how robustness to missing node features is evaluated. As shown in Sec. 3–4, much of the perceived progress in this area stems from evaluation artifacts: (i) standard benchmarks rely on extremely sparse features, which largely neutralize the effect of missingness, and (ii) missingness mechanisms are almost always benign (uniform MCAR), failing to reflect realistic conditions.
> Our contribution addresses both issues by introducing dense, semantically meaningful datasets and more realistic missingness protocols, including MNAR and distribution-shift scenarios. Within this improved evaluation setup, GNNmim is intentionally simple: it shows that, once artifacts are removed, a lightweight, assumption-free model can outperform more complex approaches. Thus, the broader contribution lies in establishing a principled and realistic evaluation framework, with GNNmim serving as a clear baseline within it.
> We have clarified the aim and scope of our work at the end of the introduction in the revised version of the paper.
>
>
> **Effectiveness of GNNmim**
>
> We thank the reviewer for the insightful comments.
> Assumption-free design. A key theoretical strength of GNNmim, compared to alternative methods, is that it is assumption-free, i.e., it does not require specifying or estimating any particular missingness mechanism (MCAR, MAR, or MNAR). This design choice allows GNNmim to generalize across a wide range of real-world scenarios without depending on unverifiable assumptions about how data become missing. Unlike other methods, it can learn to exploit observations that certain values are missing, which in MNAR settings is potentially informative for the prediction task at hand.
>
> **Comparison with zero-imputation**
>
> Q1: We agree with the reviewer that “is competitive” is a more appropriate and precise wording, since in some scenarios the performance gap is not large enough to justify “consistently outperforms.” We have revised the manuscript accordingly and replaced all occurrences of “outperforms” with “is competitive”.
>
> Q2: We thank the reviewer for the suggestion. We have updated the paper and the experimental section to include the additional baselines GNNzero and GNNmedian, which replace missing features with zeros and feature-wise medians, respectively. As expected, their performance is generally lower than GNNmim.

---

> > ### Comment · Reviewer_iRLs · 2025-11-21
> >
> > First of all, I'd like to thank the authors for their comprehensive response to my comments and for a strong rebuttal to some of the points I raised. With respect to datasets, my main concern was about how representative the chosen datasets are of the real graphs. However, I agree that the paper also raises exactly the point that the literature lacks graphs with dense features. Here are a couple of ways to obtain graphs with dense node features (i) A dataset with 17 features which I think fulfills the criteria https://pytorch-geometric.readthedocs.io/en/2.5.3/generated/torch_geometric.datasets.DGraphFin.html (ii) Use any large scale graph with attributed nodes, use an ecoder like BERT to get dense features. Please DO NOT include these in experiments; this is just for discussion/reference.
> > I also thank the author for including new baselines GNNzero and GNNmedian, which makes the paper stronger in my opinion.  Thank you for acknowledging the writing edits, and also positioning the contribution bettter in Intro. Responses address all of my concerns.

---

> > > ### Author Response · Authors · 2025-11-22
> > >
> > > Thank you very much for the follow-up and for the constructive feedback throughout the review process. We appreciate the suggestions regarding datasets with dense node features and will certainly consider them in future work. We're also glad to hear that the additions and revisions addressed your concerns.
> > >
> > > Thank you again for the helpful discussion.
> > >
> > > Best regards,
> > >
> > > The authors

---

### Author Response · Authors · 2025-12-02

Dear Area Chair,

Following the recent OpenReview rollback, we would like to provide a short factual summary of the discussion and of the clarifications introduced during the rebuttal period,
The reviewers appreciated the motivation behind our work, the clarity of the theoretical analysis, and the simplicity and effectiveness of the proposed baseline model.

To further strengthen the paper, and following their suggestions, we introduced several improvements and additions:
- a clearer explanation of the characteristics that a dataset must satisfy to be suitable for studying missing node features and for evaluating GNNs more generally, aligning our discussion with recent works highlighting the scarcity of appropriate GNN benchmarks;
- clearer intuition remarks after Theorem 1 and Theorem 2, highlighting their practical implications;
- a more explicit positioning of GNNmim as an assumption-free baseline that naturally emerges from our analysis of current evaluation limitations;
- additional baselines (GNNzero, GNNmedian) and ablations (mask-only, features-only), addressing requests from multiple reviewers;
- a new inductive-setting experiment (Appendix H), directly responding to concerns about non-inductive behavior;
- clarification that the missingness mask is deterministically constructed from the raw dataset and contains no uncertainty;
- expanded explanations in Sections 1 and 3, improving clarity around motivation, contributions, and benchmark design;
- a new experiment (Appendix I) showing that existing basic imputation baselines benefit from incorporating the missingness mask, as suggested by the reviewers.

Immediately before the rollback:
- Two reviewers with positive initial scores indicated that their questions had been addressed.
- The only reviewer who had initially given a negative score explicitly stated that several concerns had been resolved.
- The remaining positive reviewer did not further engage during the discussion, but we provided additional results (Appendix H) addressing the main limitation they identified and showing that GNNmim remains competitive in inductive settings.

Thank you for our time and consideration

The Authors

---

### Meta-Review · Area_Chair_Cgas · 2026-01-07

**Summary:**

There are several major concerns:

The empirical evaluation remains insufficiently convincing for real-world deployment, as it relies on a relatively small benchmark (one synthetic and three modest-sized real-world graphs). This limited scale raises concerns about representativeness and makes it difficult to assess the method’s scalability and generalization to large graphs or realistic missingness scenarios (Reviewers iRLs, EzuY, TVVf).

Reviewers also noted that the initial experimental section lacked several essential controls and baseline comparisons (e.g., simple imputation methods, mask-only and feature-only ablations). This omission made it challenging to confidently attribute the observed improvements to the proposed method itself, limiting the strength and comprehensiveness of the evaluation (Reviewers iRLs, EzuY, Gu9p).

The contribution of the paper in motivating future research directions was not clearly established. For example, Reviewer iRLs explicitly mentioned that the main contribution appeared to be the new benchmark developed rather than the proposed GNNmim method itself. Similar concerns were echoed by other reviewers, indicating a broader issue regarding the significance and novelty of the proposed methodology (Reviewers iRLs, EzuY, Gu9p).

Finally, reviewers identified additional weaknesses, including incomplete positioning relative to related work, limited clarity connecting theoretical results to practical implications, overly simplified missingness protocols, and unresolved concerns about generalization and inductive applicability, further weakening the overall strength of this submission (Reviewers EzuY, TVVf, Gu9p).

**Reviewer Concerns:**

The authors partially addressed concerns regarding missing baselines and ablation studies by adding experiments primarily on a synthetic dataset; however, these improvements remain limited due to insufficient validation on realistic benchmarks.

However, three major concerns are still outstanding:
1. The empirical evaluation remains weak and not representative of real-world deployments, as it continues to rely on a small number of relatively small graphs, leaving scalability and generalization largely untested.
3. The contributions of this work are still not clearly discussed.
2. Broader concerns such as simplified missingness protocols, incomplete positioning within related literature, and unresolved issues regarding inductive applicability and generalization were not convincingly addressed.

**Reviewer Scores:**

I believe Reviewer EzuY, who initially gave a low score (2), would likely maintain their score or at most slightly increase it to 4, since their fundamental concerns remain largely unresolved despite the authors' clarifications. For Reviewers iRLs, TVVf, and Gu9p, who provided higher initial scores, even though their numeric evaluations were positive, the major concerns they raised regarding empirical evaluation, representativeness, and methodological novelty were significant and remain insufficiently addressed.

---

### Decision · Program_Chairs · 2026-01-26

Reject